# Adaptive Bandit Algorithms for Contextual Matching Markets

**Shiyun Lin** [1]   **Simon Mauras** [2]   **Vianney Perchet** [3]   **Nadav Merlis** [4]

## Abstract

We study bandit learning in matching markets, where players and arms constitute the two market sides, and the players' utilities are linear in the arm contexts. In each round, new arms arrive with observable contexts. Then, the algorithm matches them to players, aiming to minimize each player's regret against a *stable matching benchmark*. This contextual structure creates significant complexity: subtle context shifts can slightly alter one player's utility while completely reconfiguring the underlying benchmark, causing large regret spikes for others. We address this in two settings: *stochastic* contexts, drawn from a latent distribution, and *adversarial* contexts, which may be arbitrary. For the stochastic case, we introduce a novel minimum preference gap to capture learning difficulty and provide a fully adaptive algorithm with an instance-dependent poly-logarithmic regret upper bound. We also establish matching instance-independent regret upper and lower bounds under a mild distributional assumption. For the adversarial setting, we propose a tractable regret notion that remains valid under arbitrary contexts and achieves an instance-independent sublinear regret bound via an adaptive algorithm.

## 1. Introduction

Two-sided matching markets provide a foundational theoretical framework for modeling and analyzing coordination problems across a wide range of economic and social domains. Classic applications include college admissions (matching students with schools) (Gale & Shapley, 1962), labor markets (assigning jobs to workers) (Kelso Jr & Crawford, 1982; Mine et al., 2013), medical residency place-

ment (pairing doctors with hospitals) (Roth, 1984), and ride-sharing (matching passengers with drivers) (Shi et al., 2023). In these markets, participants on each side hold preferences over the other. For example, jobs rank workers by skill, while workers rank jobs based on compensation or location. The central objective is to achieve a *stable matching*, where no participant pair would mutually prefer to match with each other over their current assignments. Given full preference lists, the Gale-Shapley Deferred Acceptance algorithm (Gale & Shapley, 1962) efficiently finds a stable matching that is optimal for one side of the market.

However, in online marketplaces like crowdsourcing and gig platforms, worker preferences over jobs are often not known *a priori* and must instead be learned through repeated matchings during task delegation. Recent work has thus modeled this scenario using multi-player multi-armed bandits (Das & Kamenica, 2005; Liu et al., 2020; Basu et al., 2021; Li et al., 2022; Kong & Li, 2023; Zhang & Fang, 2024; Lin et al., 2026), casting the two sides of the market as players and arms. This framework enables the development of learning algorithms that integrate Gale-Shapley dynamics with bandit-feedback techniques.

Most existing work models the market as a stochastic multi-armed bandit, assuming noisy feedback from a single, *static* preference profile. However, this assumption is often unrealistic in real-world applications. In labor markets, for example, player preferences shift with external factors such as project cycles, difficulty, and reward structures. To capture such *dynamic preferences*, a canonical approach uses contextual bandits (Chu et al., 2011; Abbasi-Yadkori et al., 2011), where the matching utility for a player-arm pair is modeled as the inner product of the player's latent preference vector and the arm's observed feature (or context) vector. The platform observes noisy utility feedback after each match and must learn the underlying preference vectors to rank arms effectively with the contextual information in each round, aiming for per-step stability or approximate stability in the long run.

Consider an online freelance platform. While a worker's preference type – defined by stable valuations for attributes like pay and complexity – remains fixed, the realized features of a specific job (e.g., its budget, deadline, and required skills) are naturally stochastic. In practice, these

[1]Center for Statistical Science, School of Mathematical Sciences, Peking University [2]INRIA, FairPlay Joint Team [3]CREST, ENSAE, IP Paris; Criteo AI lab, FairPlay Joint Team [4]Technion - Israel Institute of Technology. Correspondence to: Shiyun Lin <shiyunlin@stu.pku.edu.cn>.

*Proceedings of the 43rd International Conference on Machine Learning*, Seoul, South Korea. PMLR 306, 2026. Copyright 2026 by the author(s).

features can be modeled as independent draws from a fixed but unknown distribution associated with the job's category (such as graphic design or copywriting). This motivates our first setting, in which the contextual feature vector of each arm is generated *stochastically* from a category-specific distribution in every matching round. In contrast, other realistic matching environments involve non-stationary or even strategically manipulated contexts. On a competitive procurement platform, for instance, clients may adaptively adjust a project's described terms, complexity, or evaluation metrics – either due to market shifts or to strategically exploit contractors' learned bidding behaviors. To capture such adversarial dynamics, we also analyze a setting in which the sequence of context vectors can be *arbitrary and potentially adversarial*.

Adaptive algorithms in bandits (Wu et al., 2018; Hao et al., 2020) dynamically balance exploration and exploitation based on observed data, improving both statistical efficiency and practical deployment. In two-sided matching markets, this adaptivity is crucial for automatically adjusting to changing user distributions, enhancing performance in non-stationary environments. We therefore seek to develop *adaptive* bandit algorithms for contextual matching markets.

## 1.1. Main Contributions

**Novel Metric for Characterizing Problem Difficulty** To characterize problem difficulty, we introduce novel metrics tailored to each setting. For the stochastic environment, we propose a *minimum preference gap*, a relaxed version of prior gap definitions that offers better realism and practicality. For the adversarial environment, we define a tractable form of *regret* measured against a player-optimal stable matching benchmark – employing its exact form in the large-gap regime and an approximation in the small-gap regime. Together, these metrics form a unified analytical framework that enables us to derive meaningful theoretical guarantees for our proposed algorithms.

**Fully Adaptive Algorithms** We develop separate algorithms for the stochastic and adversarial settings. Both employ an *adaptive phase transition* scheme, alternating between refining parameter estimates and performing (approximately) stable matchings. This adaptive design allows handling *arbitrary context covariance structures* in the stochastic case and *arbitrary context sequences* in the adversarial case. For the stochastic setting, we add a batched design that maintains and shrinks a candidate minimum preference gap toward the true value, enabling operation without prior knowledge of the true gap with respect to the distribution.

**Theoretical Regret Guarantees** We establish non-asymptotic regret upper bounds for our proposed algorithms under minimal assumptions, achieving instance-dependent poly-logarithmic rate in the stochastic setting and instance-

independent sublinear rate in the adversarial setting. For the stochastic setting, under the additional condition of a strictly positive minimum eigenvalue for the context covariance matrix, we provide a logarithmic regret guarantee. In the asymptotic regime of the stochastic setting, we also derive instance-independent regret bounds, proving tight $\tilde{\Theta}\left(T^{2/3}\right)$ upper and lower bounds, where $T$ is the time horizon. Notably, this result contrasts with prior work on stochastic bandit learning in matching markets with static preferences, where instance-independent guarantees are only possible for approximation benchmark.

## 1.2. Related Work

**Bandit Learning in Matching Markets** Das & Kamenica (2005) initiated the study of bandit learning in matching markets by introducing the problem and proposing empirical solutions. Liu et al. (2020) later provided a formal theoretical formulation and established initial guarantees on stable regret. Subsequent research has expanded this line of work, aiming to improve regret bounds (Sankararaman et al., 2021; Kong & Li, 2023) and explore various problem variants, including decentralized matching (Liu et al., 2021; Basu et al., 2021), many-to-one markets (Wang et al., 2022), settings with preference indifference (Kong et al., 2025; Lin et al., 2026), markets with one-sided preferences (Lin, 2026), and models with transferable utilities (Jagadeesan et al., 2023). For a comprehensive survey, please refer to Li et al. (2025).

Despite extensive work on static preferences, research on dynamic preference structures remains limited. Recent approaches model non-stationarity via constraints on the total number of preference changes (Muthirayan et al., 2023) or on per-round utility drift (Ghosh et al., 2024). In contextual matching markets, previous work assumes prior knowledge of the minimum preference gap and covariance structure (Li et al., 2022) or studies decentralized settings with fixed and known number of environments (Parikh et al., 2024). In contrast, we develop *fully adaptive*, centralized algorithms for both stochastic and adversarial settings, requiring *no prior knowledge* of the environment.

**Contextual Combinatorial Bandits** Contextual combinatorial bandits (CCB) extend the multi-armed bandit framework to sequential decision-making where an agent, given contextual information, must select a *set of items* (a "super arm") to maximize an aggregate reward. Under semi-bandit feedback, Qin et al. (2014), Wen et al. (2015), and Takemura et al. (2021) incorporate linear reward assumption into the model, while Chen et al. (2018) studies volatile arms with submodular rewards. In contrast, Li et al. (2024) examines the full-bandit setting, and Zierahn et al. (2023) considers nonstochastic contexts.

We study contextual bandits in matching markets, where the

focus is on forming bilateral matches between players and arms. We receive reward feedback for every matched pair and consider both stochastic and adversarial contexts. A key innovation is our *player-optimal stable regret* metric, which ensures fairness for each player rather than optimizing for a single aggregate benchmark as in CCB, introducing unique algorithmic challenges.

## 2. Problem Formulation

In this section, we formulate the problem setting of contextual bandits in matching markets over a finite horizon $T$. For a positive integer $n$, we use $[n]$ to represent $\{1, 2, \cdots, n\}$.

Consider a market with $N$ players and $K$ arms. We define the set of players as $\mathcal{N} = \{p_1, p_2, \cdots, p_N\}$ and the set of arms as $\mathcal{K} = \{a_1, a_2, \cdots, a_K\}$. Following previous work (Liu et al., 2020; 2021; Sankararaman et al., 2021; Kong & Li, 2023; Lin et al., 2026), we assume $N \leq K$ to ensure that each player has a chance of being matched.

### 2.1. Preference Structure

**Fixed and known preferences of arms over players** We assume each arm $a_j$ has a fixed, strict, and known preference ordering $P_{a_j}$ over players. That is, if $p_i \succ_{a_j} p_{i'}$ in $P_{a_j}$, arm $a_j$ strictly prefers player $p_i$ to player $p_{i'}$. This knowledge of arm-side preferences is a mild and common assumption, consistent with prior work in bandit learning in matching markets (Liu et al., 2020; Kong & Li, 2023; Lin et al., 2026), and reflects settings where arms (e.g., jobs or schools) have well-established rankings over players (e.g., candidates or applicants) through prior evaluations such as recruitment interviews and entrance examinations.

**Dynamic and unknown preferences of players over arms** In contrast, players' preferences over arms are dynamic, uncertain, and must be learned over time. We model these preferences via a linear contextual utility model. At each time step $t$, the utility of player $p_i$ for arm $a_j$ is given by

$$\boldsymbol{U}_{i,j}(t) = \theta_i^\top \mathrm{x}_j(t), \tag{1}$$

where $\theta_i \in \mathbb{R}^d$ is an unknown *preference parameter* intrinsic to player $p_i$ and fixed across all $T$ rounds, and $\mathrm{x}_j(t) \in \mathbb{R}^d$ is the *contextual feature vector* of arm $a_j$ at time $t$, revealed before the matching decision. The true utility is not directly observed; instead, upon matching player $p_i$ to arm $a_j$ at time $t$, the platform receives a noisy reward

$$y_{i,j}(t) = \boldsymbol{U}_{i,j}(t) + \epsilon_{i,j}(t), \tag{2}$$

with $\epsilon_{i,j}(t)$ being i.i.d. noise with mean 0. And if a player is not matched at time $t$, the received reward $y_i(t)$ is set to be 0. With all player-arm pairs, we could construct a utility matrix $\boldsymbol{U}(t)$ encoding the preferences of players over arms. This formulation captures settings such as online labor

markets, where workers (players) have latent preferences over jobs (arms) that can only be inferred through repeated interactions. In such markets, the ranking of a worker over available jobs is revealed only when job tasks are posted, while the platform can access the fixed preferences of jobs over workers through prior screening or interviews.

For the valuation model, we make the following assumptions throughout the paper.

**Assumption 1.** Contexts are bounded: $\|\mathrm{x}_j(t)\|_2 \leq B_x$, $\forall j \in [K]$ almost surely.

**Assumption 2.** Let $\{F_t\}_{t=0}^\infty$ be a filtration such that $\mathrm{x}_{\mu_t(i)}(t)$ is $F_{t-1}$ measurable and $\epsilon_t$ is $F_t$-measurable, and $\epsilon_t$ is conditionally $R$-subgaussian for some $R \geq 0$.

**Assumption 3.** The preference parameters are bounded: $\|\theta_i\|_2 \leq B_\theta, \forall i \in [N]$.

Denote $B_y := 2B_\theta B_x$. Following conventional linear contextual bandits literature, we assume $B_y \leq 1$.

### 2.2. Matching and Stability

Stability is a key concept to characterize the matching in the market, which ensures there is no *justified envy* in the market. Formally, a matching $\mu_t$ is stable at time $t$ if no player and arm has an incentive to abandon their current partner according to the utility matrix $\boldsymbol{U}(t)$ and the preference profile $(P_a)_{a \in \mathcal{K}}$.

**Definition 1** (Stability). A matching $\mu_t$ is stable at time $t$ if there is no *blocking pair* $(p_i, a_j)$ such that $p_i \succ_{a_j} \mu_t(a_j)$ and $\boldsymbol{U}_{i,j}(t) > \boldsymbol{U}_{i,\mu_t(p_i)}(t)$.

Stable matchings may not be unique. Let $\mathcal{S}_t := \{\mu : \mu \text{ is a stable matching w.r.t. } \boldsymbol{U}(t) \text{ and } (P_a)_{a \in \mathcal{K}}\}$ be the set of all stable matchings with respect to $\boldsymbol{U}(t)$ and $(P_a)_{a \in \mathcal{K}}$. Given complete preferences, the Gale-Shapley (GS) algorithm (Gale & Shapley, 1962) finds a stable matching after repeated proposals from one side of the market to the other. When the preferences for both sides are strict, due to the lattice structure of the problem, the matching returned by the GS algorithm is always optimal within $\mathcal{S}_t$ for each member of the proposing side (Knuth, 1997). Let $\mu_t^* = \{(i, \mu_t^*(i)) : i \in [N]\} \in \mathcal{S}_t$ be the players' most preferred stable matching. That is to say, $\boldsymbol{U}_{i,\mu_t^*(i)}(t) \geq \boldsymbol{U}_{i,\mu(i)}(t)$ for any $\mu \in \mathcal{S}_t, i \in [N]$. We define $\boldsymbol{U}_i^*(t) := \boldsymbol{U}_{i,\mu_t^*(i)}(t) = \max_{\mu \in \mathcal{S}_t} \boldsymbol{U}_{i,\mu_t(i)}(t)$ as the player's *optimal stable share* at time $t$.

### 2.3. Regret Definition

The performance of an algorithm is measured by the *player-optimal stable regret*. For a player $p_i \in \mathcal{N}$, this regret is defined at time $T$ as the sum, over all rounds, of the difference in reward between being matched by the time-varying

player-optimal stable policy $\mu_t^*$ and by the algorithm's implemented policy $\mu_t$:

$$Reg_i(T) = \sum_{t=1}^{T} \boldsymbol{U}_i^*(t) - \mathbb{E}\left[\sum_{t=1}^{T} y_i(t)\right], \quad (3)$$

where $y_i(t) = y_{i,\mu_t(i)}(t)$ is the observed reward of player $p_i$ under the matching $\mu_t$ at time $t$, with the expectation over algorithmic randomness and noise.

The goal of the problem is to design an online matching algorithm that minimizes the regret for every player by learning the unknown preference parameters $\{\theta_i\}$ while ensuring stable or near-stable matchings over time.

## 3. Contextual Matching Markets with Stochastic Contexts

In this section, we consider the setting where the context feature vector $\mathrm{x}_j(t)$ of each arm $a_j$ is drawn i.i.d. from a fixed but unknown distribution $\mathcal{D}_{\mathrm{x}_j}$ at every round $t$.

Given the realized contexts $\mathrm{x}_j$, for $j \in [K]$, define the *minimum difference* between any two utilities for player $p_i$ as[1] $\delta_{\min}^{(i)} = \min_{j,j' \in [K]} |(\theta_i)^\top (\mathrm{x}_j - \mathrm{x}_{j'})|$, and the minimum difference for the utility matrix as $\delta_{\min} = \min_{i \in [N]} \delta_{\min}^{(i)}$.

The difficulty of the learning and matching problem is directly tied to the minimum difference: a larger difference simplifies the task by making rankings more easily distinguishable. To formalize this measure of problem hardness, we define the following minimum preference gap.

**Definition 2** (Minimum Preference Gap). Given the ground-truth context distribution $(\mathcal{D}_{\mathrm{x}_j})_{j \in [K]}$ and preference parameters $(\theta_i)_{i \in [N]}$, the *minimum preference gap* $\Delta_{\min}$ is defined as the supremum of all gaps for which the event $\{\delta_{\min} \geq \Delta\}$ is sufficiently likely – specifically, the probability of the event is greater than $1 - \frac{\log T}{T\Delta^2}$. That is,

$$\Delta_{\min} := \sup\left\{\Delta : \mathbb{P}\left(\delta_{\min} \geq \Delta\right) \geq 1 - \frac{\log T}{T\Delta^2}\right\}. \quad (4)$$

The definition of the minimum preference gap is motivated by the exploration-exploitation trade-off. To estimate a utility within a precision $\Delta$ and with confidence $1 - \frac{1}{T}$, we require $\mathcal{O}\left(\log T/\Delta^2\right)$ exploration samples. Conversely, for a given $\Delta$, if $\mathbb{P}\left(\delta_{\min} \geq \Delta\right) > 1 - \zeta$ for some threshold $\zeta > 0$, then during exploitation there is at most an $\mathcal{O}(\zeta)$ probability of making a ranking error due to indistinguishability, leading to an $\mathcal{O}(\zeta T)$ regret contribution. Balancing the exploration regret, $\mathcal{O}\left(\log T/\Delta^2\right)$, against the exploitation regret, $\mathcal{O}(\zeta T)$, by setting these terms equal yields the optimal threshold $\zeta = \frac{\log T}{T\Delta^2}$, which directly informs our definition. To aid interpretability, we provide an illustrative example in Appendix I.

*Remark* 3.1. Our minimum preference gap $\Delta_{\min}$ generalizes the stricter sub-optimality gap $\tilde{\Delta}$ used in prior work (Li et al., 2022), which relies on a deterministic condition: a uniform lower bound for $\delta_{\min}$ over all $T$ rounds. This could be viewed as $\mathbb{P}\left(\delta_{\min} \geq \tilde{\Delta}\right) = 1$. In contrast, our framework adopts a probabilistic soft threshold. It is clear that $\Delta_{\min} \geq \tilde{\Delta}$, and thus any regret upper bound established using our condition $\Delta_{\min}$ immediately provides a valid guarantee for the more restrictive $\tilde{\Delta}$.

On the other hand, unlike the deterministic lower bounds in Parikh et al. (2024), which constrain the environment's behavior and entirely rule out worst-case realizations, our probabilistic minimum preference gap captures the intrinsic difficulty of the distribution, permits occasional ties, and provides stronger guarantees when worst-case scenarios arise with only small probability.

### 3.1. Batched Adaptive Regret-Balancing Algorithm

We present Batched Adaptive Regret-Balancing (BARB, Algorithm 1) for contextual bandits in matching markets with stochastic contexts. BARB operates in batches. In batch $k$, it maintains a candidate gap $\Delta_k$ – initialized with the input $\Delta_1$ and shrunk across batches (Line 21) – to iteratively approximate the true minimum preference gap $\Delta_{\min}$.

Each batch executes two adaptive phases: an *exploration phase* (Line 8 - 10), where we use maximum cardinality matchings (Line 9) to collect samples to refine parameter estimates, and an *exploitation phase*, where the Gale-Shapley algorithm is deployed when rankings are distinguishable (Line 13); otherwise, evidence is gathered to advance to the next batch (Line 14 - 16).

Take $\mathcal{G}_k^{(i)}$ as the set of rounds in which player $p_i$ explores, i.e., $p_i$ is involved in the maximum cardinality matching in those rounds in $\mathcal{G}_k^{(i)}$. Let $i_s = \mu_s(i)$ be the arm matched to player $p_i$ under the implemented matching $\mu_s$ at time $s$. In time $t_k$ of the exploration phase of the $k$-th batch, we use the following update rule for the Gram matrix $\boldsymbol{V}_i^{(t_k)}$ and the preference parameter $\hat{\theta}_i^{(t_k)}$:[2]

$$\boldsymbol{V}_i^{(t_k)} = \lambda \boldsymbol{I} + \sum_{s < t_k, s \in \mathcal{G}_k^{(i)}} \mathrm{x}_{i_s} \mathrm{x}_{i_s}^\top,$$

$$\hat{\theta}_i^{(t_k)} = \left(\boldsymbol{V}_i^{(t_k)}\right)^{-1} \sum_{s < t_k, s \in \mathcal{G}_k^{(i)}} \mathrm{x}_{i_s} y_{i,i_s}. \quad (5)$$

Given the ridge parameter $\lambda$, let $\delta = T^{-2}$, let the input parameter $\eta = R\sqrt{d \log\left(\frac{1+TB_x^2/\lambda}{\delta}\right)} + \lambda^{1/2} B_\theta$. Classical result on ridge regression in the bandit framework (Abbasi-

---

[1]Without causing ambiguity, we omit the time index in the following.

[2]Without ambiguity, we omit the time index $s$ for the observed context and the reward.

**Algorithm 1** Batched Adaptive Regret-Balancing (BARB), abstract version

**Input:** Time horizon $T$, preference profile $(P_a)_{a \in \mathcal{K}}$, context dimension $d$, parameter $\eta$, candidate gap $\Delta_1$, ridge penalty parameter $\lambda$.

1: $t \leftarrow 1$.
2: **for** $k = 1, 2, \cdots$ **do**
3:      $\xi_k \leftarrow \frac{\Delta_k}{\eta}$; $\boldsymbol{V}_i^{(1)} \leftarrow \lambda \boldsymbol{I}_{d \times d}, \hat{\theta}_i^{(1)} \leftarrow \boldsymbol{0}_d, \forall i \in [N]$.
4:      $N_k \leftarrow 0$. {Exploitation rounds with overlap CIs.}
5:      **for** $t_k = 1, 2, \cdots$ **do**
6:          Observe context $\mathrm{x}_j(t_k)$ for arm $a_j, \forall j \in [K]$.
7:          **if** $\exists i, j$, s.t. $\|x_j(t_k)\|_{\left(\boldsymbol{V}_i^{(t_k)}\right)^{-1}} > \xi_k$ **then**
8:              Form a bipartite graph $G_{t_k}$ such that $\forall (i, j) \in G_{t_k}$ has $\|\mathrm{x}_j(t_k)\|_{\left(\boldsymbol{V}_i^{(t_k)}\right)^{-1}} > \xi_k$.
9:              $\mu_{t_k} \leftarrow$ **Maximum Cardinality Matching** on $G_{t_k}$.
10:             Update $\boldsymbol{V}_i^{(t_k+1)}$ and $\hat{\theta}^{(t_k+1)}$ with Eq.(5).
11:          **else**
12:             $\hat{\boldsymbol{U}}_{i,j}(t_k) \leftarrow \left(\hat{\theta}_i^{(t_k)}\right)^\top \mathrm{x}_j(t_k)$.
13:             $\mu_{t_k} \leftarrow$ **Deferred Acceptance** with $\hat{\boldsymbol{U}}(t_k)$.
14:             **if** $\exists i$, overlapping utility CIs exist **then**
15:                 $N_k \leftarrow N_k + 1$.
16:             **end if**
17:          **end if**
18:          $t \leftarrow t + 1$.
19:          **if** $t > T$ **then** Stop and return. **end if**
20:          **if** $N_k > \frac{3 \log T}{16 \Delta_k^2}$ **then**
21:             $\Delta_{k+1} \leftarrow \frac{\Delta_k}{\sqrt{2}}$, enter the next batch.
22:          **end if**
23:      **end for**
24: **end for**

Yadkori et al., 2011) shows that $\|\theta_i - \hat{\theta}_i^{(t_k)}\|_{\boldsymbol{V}_i^{(t_k)}} \leq \eta$ with high probability. On the other hand, the relationship $|\boldsymbol{U}_{i,j} - \hat{\boldsymbol{U}}_{i,j}| \leq \|\theta_i - \hat{\theta}_i^{(t_k)}\|_{\boldsymbol{V}_i^{(t_k)}} \|\mathrm{x}_j(t_k)\|_{\left(\boldsymbol{V}_i^{(t_k)}\right)^{-1}}$ suggests that having $\|\mathrm{x}_j(t_k)\|_{\left(\boldsymbol{V}_i^{(t_k)}\right)^{-1}} \leq \frac{\Delta_k}{\eta} := \xi_k$ suffices to ensure the estimated utility $\boldsymbol{U}$ is of precision $\Delta_k$ with high probability. This observation motivates the design of the adaptive criterion (Line 7) to transit between exploration and exploitation within each batch.

During the exploitation phase of batch $k$, for every player $p_i$, we construct confidence intervals with radius $\Delta_k$ for the estimated utility of each arm and track overlaps between intervals of distinct arms using a counter $N_k$. An excessive overlap indicates that $\Delta_k$ may be too large relative to $\Delta_{\min}$. Consequently, when $N_k$ exceeds a specified threshold (Line 20), the algorithm moves to batch $k + 1$.

## 3.2. Theoretical Analysis

We now present the regret upper bound for each player by following the BARB algorithm.

**Theorem 3.1** (Upper Bound for Stochastic Contexts). *Assume that the environment follows Assumption 1, 2 and 3, and the contexts are stochastic. Following the BARB algorithm, the player-optimal stable regret for $p_i \in \mathcal{N}$ is*

$$Reg_i(T) = \mathcal{O}\left(\frac{\log^2 T}{\Delta_{\min}^2} + \frac{\log T}{\Delta_{\min}^2}\right) = \mathcal{O}\left(\frac{\log^2 T}{\Delta_{\min}^2}\right). \quad (6)$$

We provide the proof sketch for Theorem 3.1 as follows, while the detailed proof is deferred to Appendix B.

*Proof Sketch.* In the exploration phase of the $k$-th batch, from the contruction of graph $G_{t_k}$, we know that $\|\mathrm{x}_{i_{t_k}}(t_k)\|_{\left(\boldsymbol{V}_i^{(t_k)}\right)^{-1}} > \xi_k$, while by elliptical potential lemma (Carpentier et al., 2020), we have $\sum_{t_k \in \mathcal{G}_k^{(i)}} \|\mathrm{x}_{i_{t_k}}(t_k)\|_{\left(\boldsymbol{V}_i^{(t_k)}\right)^{-1}} \leq \sqrt{|\mathcal{G}_k^{(i)}| d \log\left(\frac{|\mathcal{G}_k^{(i)}| + d\lambda}{d\lambda}\right)}$ and hence $|\mathcal{G}_k^{(i)}| \leq \frac{d \log\left(\frac{T+d\lambda}{d\lambda}\right)}{\xi_k^2}$. Let $\mathcal{G}_k$ be the set of rounds that any player explores, we have $\mathcal{G}_k = \cup_i \mathcal{G}_k^{(i)}$ and hence $|\mathcal{G}_k| \leq \sum_i |\mathcal{G}_k^{(i)}| \leq \frac{Nd \log\left(\frac{T+d\lambda}{d\lambda}\right)}{\xi_k^2}$. Since $\xi_k = \frac{\Delta_k}{\eta}$ with $\eta = \mathcal{O}\left(\sqrt{\log T}\right)$, we have that the exploration regret is no larger than $B_y |\mathcal{G}_k| = \mathcal{O}\left(\frac{\log^2 T}{\Delta_k^2}\right)$.

In the exploitation phase of the $k$-th batch, we incur no regret when the confidence intervals for the utilities are non-overlapped by calling the Gale-Shapley algorithm. On the other hand, by the stopping threshold, we have at most $N_k$ rounds with overlapped confidence intervals in the $k$-th exploitation phase, by the specified threshold for $N_k$ (Line 20), the exploitation regret for the $k$-th batch is $\mathcal{O}(\frac{\log T}{\Delta_k^2})$.

Finally, with multiplicative Chernoff bound (Mitzenmacher & Upfal, 2017), we could stop exploration at the first batch when $\Delta_k < \Delta_{\min}/4$ with high probability. By the doubling trick applied in the algorithm design, the regret of the $k$-th batch dominates the sum of the regret of the first $k - 1$ batches, and hence the final regret is as shown in Eq.(6). $\square$

The regret upper bound in Theorem 3.1 holds for a fully adaptive algorithm. The algorithm's strength is its minimal assumption: contexts need only be stochastic, with no further requirements on the underlying distribution. Under the additional structural assumption that the context covariance matrix has a uniformly bounded minimum eigenvalue, we can eliminate a $\log T$ factor from the regret bound. This refined bound then explicitly reflects the problem's hardness by inversely relating to this eigenvalue lower bound.

**Assumption 4.** Let $\mathcal{D}_\mathrm{x}$ be the joint distribution of contexts for all $K$ arms. Define the context covariance matrix as

$V = \mathbb{E}\left[XX^\top \mid X \in \mathcal{D}_x\right]$. Then there exists a deterministic constant $\tilde{\lambda}$ such that for all $X \in \mathcal{D}_x$ we have the minimum eigenvalue of the covariance matrix $\lambda_{\min}(V) \geq \tilde{\lambda} > 0$.

Assumption 4 ensures that, under the distribution $\mathcal{D}_x$, the data spans all dimensions of the feature space, thereby avoiding multicolinearity or near-degeneracy. When Assumption 4 holds, it is possible to design a batched explore-then-commit (batched-ETC) algorithm that retains the regret-balancing principle of BARB (Algorithm 1) without requiring adaptive exploration. The resulting algorithm achieves an alternative regret upper bound of $\mathcal{O}(\frac{\log T}{\tilde{\lambda}^2 \Delta_{\min}^2})$. The algorithmic details and the regret analysis are deferred to Appendix C.

### 3.3. Matching Asymptotic Regret Bounds

In this subsection, we demonstrate that when the distribution of the minimum difference $\delta_{\min}$ satisfies a mild regularity condition, the minimum preference gap definition (Definition 2) yields an asymptotic regret upper bound of $\tilde{\mathcal{O}}(T^{2/3})$ as $T \to \infty$. Furthermore, we construct an instance that satisfies this regularity condition and attains a matching lower bound of $\Omega(T^{2/3})$, establishing the tightness of the asymptotic regret rate. These matching regret bounds also justify the rationality of the minimum preference gap definition.

We first present the following assumption that ensures the cumulative distribution function (CDF) of $\delta_{\min}$ is dominated by a linear function when it is close enough to 0.

**Assumption 5.** Under the definition of minimum difference $\delta_{\min}$, denote the CDF of $\delta_{\min}$ as $F(\Delta)$, for some constant $\Delta_0$ sufficiently close to 0, and for some constant $c > 0$, we have, $\forall \Delta \leq \Delta_0, F(\Delta) \leq c\Delta$.

Assumption 5 requires that the probability that the minimum utility difference falls below $t$ grows at most linearly in $t$, ensuring that we do not have excessively high probability for small gaps. This assumption is typically satisfied when the minimum difference $\delta_{\min}$ has a bounded probability density function (PDF) in a neighbourhood of zero.

The following theorem shows the asymptotic regret upper bound under Assumption 5.

**Theorem 3.2.** *Under Assumption 5, when $T$ is sufficiently large, the regret of the BARB algorithm satisfies $Reg_i(T) = \tilde{\mathcal{O}}(T^{2/3})$, $\forall i \in [N]$.*

The proof of Theorem 3.2 can be found in Appendix D. We now provide a matching minimax regret lower bound in the following theorem.

**Theorem 3.3.** *Let $N = K = 3$. For any policy $\pi$, there exists an instance satisfying Assumption 5 such that for some constants $c, \Delta_0 > 0$, when $\Delta \leq \Delta_0$, the CDF of $\delta_{\min}$ obeys*

$F_T(\Delta) \leq c\Delta$ *for all $T \geq 1$.[3] Then at least one player suffers regret $\Omega(T^{2/3})$, i.e., there exists $i \in [N]$ such that $Reg_i(T) \geq f(c)T^{2/3}$, where $f(c)$ depends only on $c$.*

The proof idea is to construct two instances with carefully designed preference parameters and context distributions, such that their resulting utility matrices differ in only a single entry. Insufficient sampling of this critical entry leads to high regret for one player in the first instance, while sampling it too frequently incurs high *social regret* (the sum of regrets across all players) in the second instance. This inherent trade-off yields the stated regret lower bound. The details can be found in Appendix E.

## 4. Contextual Matching Markets with Adversarial Contexts

In contrast to the stochastic setting, we now analyze worst-case scenarios in which no assumptions are placed on context generation, i.e., contexts may even be chosen by an adaptive adversary. This adversarial model offers robustness guarantees against non-stationary environments and strategic manipulation, ensuring algorithm performance even when stochastic assumptions fail to hold.

In the stochastic setting (Section 3), the regret bound scales as $1/\Delta_{\min}^2$, implying the problem becomes harder as $\Delta_{\min}$ shrinks. However, $\Delta_{\min}$ is defined with respect to a stationary context distribution, which no longer exists in the adversarial case. Here, an adversary can select arbitrary contexts, potentially forcing the minimum utility difference $\delta_{\min}$ arbitrarily small for arbitrarily many rounds. This makes the problem intractable under the usual regret measure. To restore tractability, we first introduce a refined, tractable regret definition suitable for adversarial contexts. We then develop an adaptive algorithm for this setting and establish a corresponding regret upper bound.

### 4.1. Tractable Regret with Adversarial Contexts

The inverse scaling of regret with the minimum preference gap represents a fundamental limit in bandit learning for matching markets, an unavoidable phenomenon demonstrated in Example 2 in Liu et al. (2020). Consequently, when the minimum gap is small or the preference profile admits ties, standard regret guarantees become vacuous. The underlying issue is that a market with ties lacks a single, universally utility-maximizing stable matching, creating a dilemma where uncertainty in one player-arm pair can propagate regret to other players. To address this challenge, Lin et al. (2026) introduced an approximation regret framework

---

[3]The context distribution and hence the distribution of $\delta_{\min}$ may depend on $T$. Yet, the condition holds uniformly over $T$, and so the guarantees of Theorem 3.2 holds starting from some $T_0 > 0$.

in stochastic multi-armed bandits for static markets. In our setting – contextual bandits for dynamic markets with adversarial contexts – we may encounter infinitely many rounds with arbitrarily small preference gaps, making meaningful regret guarantees impossible under the standard stable regret definition. Therefore, we adapt this idea and propose a regret definition that smoothly interpolates between rounds with large and small utility differences.

Before presenting the refined regret definition, we first introduce some notation about $\varepsilon$-stable matching and approximation oracle.

**Definition 3** ($\varepsilon$-Stability)**.** Given $\varepsilon \geq 0$, a matching $\mu_t$ is $\varepsilon$-stable at time $t$ if there is no $\varepsilon$-*blocking pair* $(p_i, a_j)$ such that $p_i \succ_{a_j} \mu_t(a_j)$ and $\boldsymbol{U}_{i,j}(t) > \boldsymbol{U}_{i,\mu_t(p_i)}(t) + \varepsilon$.

The notion of $\varepsilon$-stability is a relaxation of stability, where setting $\varepsilon = 0$ makes it equivalent to stability (Definition 1). In general, $\varepsilon$-stable matching is not unique. We define

$$\mathcal{S}_t^\varepsilon := \{\mu : \mu \text{ is an } \varepsilon\text{-stable matching w.r.t. } \boldsymbol{U}(t) \text{ and } (P_a)_{a \in \mathcal{K}}\},$$

and we call $a_j$ a *valid $\varepsilon$-stable match* for player $p_i$ at time $t$ if there exists an $\varepsilon$-stable matching matches $p_i$ with $a_j$, and it is the *optimal $\varepsilon$-stable match* of player $p_i$ at time $t$ if it is the most preferred valid $\varepsilon$-stable match, i.e., there exists a matching $\mu_t^\varepsilon \in \mathcal{S}_t^\varepsilon$ such that $\mu_t^\varepsilon(i) = a_j$ and $\boldsymbol{U}_{i,\mu_t^\varepsilon(i)}(t) = \max_{\mu \in \mathcal{S}_t^\varepsilon} \boldsymbol{U}_{i,\mu(i)}(t)$. We say $\boldsymbol{U}_{i,\mu_t^\varepsilon(i)}(t)$ is the *$\varepsilon$-optimal stable share* for player $p_i$ at time $t$, denoted as $\boldsymbol{U}_i^\varepsilon(t)$.

Since no single $\varepsilon$-stable matching maximizes all players' utilities simultaneously, Lin et al. (2026) proposed a utility approximation under bounded instability, ensuring each player achieves a certain satisfaction level in the presence of (statistical) ties in a static market. We adapt this to the contextual setting: when the minimum utility difference $\delta_{\min}$ is large – making preference rankings and the unique player-optimal stable matching easy to identify – we use the optimal stable share $\boldsymbol{U}_i^*(t)$ as the benchmark. Conversely, when $\delta_{\min}$ is small, we adopt an $\alpha$-approximation of $\boldsymbol{U}_i^\varepsilon(t)$, where $\alpha \in (0, 1]$ is an approximation ratio. Formally, given predetermined parameters $\Delta$ and $\varepsilon$ with $\varepsilon < \Delta$ – typically set as $\varepsilon = \Delta/2$ for simplicity – we define the player's *$\alpha$-approximate $\Delta$-optimal stable regret* as follows:

$$Reg_i^{\alpha,\Delta}(T) = \sum_{t=1}^T \left[ \boldsymbol{U}_i^*(t)\mathbb{1}\{\delta_{\min}(t) > \Delta\} \right. \tag{7}$$
$$\left. + \alpha\boldsymbol{U}_i^\varepsilon(t)\mathbb{1}\{\delta_{\min}(t) \leq \Delta\} \right] - \mathbb{E}\left[\sum_{t=1}^T y_i(t)\right].$$

We assume access to an offline approximation oracle that, when given players' utility matrix $\boldsymbol{U}$, the uncertainty set with radius $\gamma$, this oracle returns a matching whose expected utility is guaranteed to be at least an $\alpha$-fraction of $\boldsymbol{U}_i^\varepsilon$, with an additive error bounded by $2\gamma + \varepsilon$. For instance, Lin et al. (2026) constructed an oracle with an approximation

---

**Algorithm 2** Adaptive Explore-Choose Oracle (AdECO), abstract version

**Input:** Time horizon $T$, preference profile $(P_a)_{a \in \mathcal{K}}$, context dimension $d$, parameter $\Delta$, $\varepsilon$, and $\eta$, ridge penalty parameter $\lambda$.

1: $\xi \leftarrow \frac{\Delta-\varepsilon}{4\eta}$; $\boldsymbol{V}_i^{(1)} \leftarrow \lambda\boldsymbol{I}_{d \times d}, \hat{\theta}_i^{(1)} \leftarrow \boldsymbol{0}_d, \forall i \in [N]$.
2: **for** $t = 1, 2, \cdots, T$ **do**
3:      Observe context $\mathrm{x}_j(t)$ for arm $a_j$, $\forall j \in [K]$.
4:      **if** $\exists i, j$, s.t. $\|\mathrm{x}_j(t)\|_{\left(\boldsymbol{V}_i^{(t)}\right)^{-1}} > \xi$ **then**
5:          Form a bipartite graph $G_t$ such that $\forall (i, j) \in G_t$ has $\|\mathrm{x}_j(t)\|_{\left(\boldsymbol{V}_i^{(t)}\right)^{-1}} > \xi$.
6:          $\mu_t \leftarrow$ **Maximum Cardinality Matching** on $G_t$.
7:          Update $\boldsymbol{V}_i^{(t+1)}$ and $\hat{\theta}^{(t+1)}$ according to Eq.(8).
8:      **else**
9:          $\hat{\boldsymbol{U}}_{i,j}(t) \leftarrow \left(\hat{\theta}_i^{(t)}\right)^\top \mathrm{x}_j(t)$.
10:         **if** Distance between any two CIs $\geq \varepsilon$ **then**
11:             $\mu_t \leftarrow$ **Deferred Acceptance** with $\hat{\boldsymbol{U}}(t)$.
12:         **else**
13:             $\mu_t \leftarrow \alpha$**-Approximation Oracle** with $\hat{\boldsymbol{U}}(t)$ and instability tolerance $\frac{\Delta+\varepsilon}{2}$.
14:         **end if**
15:      **end if**
16: **end for**

---

ratio of $1/\lfloor \log_2 N + 2 \rfloor$. For more details, please refer to Appendix F. In our online learning algorithm, we call this offline oracle with estimated utilities and constructed confidence intervals to approach the benchmark utility when $\delta_{\min}$ is small.

### 4.2. Adaptive Explore-Choose Oracle Algorithm

We propose the Adaptive Explore-Choose Oracle Algorithm (AdECO) for contextual bandits in matching markets with adversarial contexts. AdECO's exploration scheme follows the same adaptive principle as BARB: When $\|\mathrm{x}_j(t)\|_{\left(\boldsymbol{V}_i^{(t)}\right)^{-1}}$ is greater than the threshold $\xi$, we perform exploration through maximum cardinality matching on a bipartite graph $\mathcal{G}_t$, and update the estimation with the contextual information and the collected rewards (Line 4 - 7). Otherwise, the algorithm exploits by choosing between executing the Gale-Shapley algorithm (Line 11) or calling the $\alpha$-approximation oracle (Line 13), based on whether the confidence intervals for the utilities are sufficiently separated.

The setting of the parameter $\eta$ follows the same idea as that employed in the stochastic contexts case. Take $\mathcal{G}^{(i)}$ as the set of rounds in which player $p_i$ explores, i.e., $p_i$ is involved in the maximum cardinality matching in those rounds $s \in \mathcal{G}^{(i)}$. In time $t$ of the exploration phase, we use

the following update rule for the Gram matrix $V_i^{(t)}$ and the preference parameter $\hat{\theta}_i^{(t)}$:

$$
\begin{aligned}
V_i^{(t)} &= \lambda I + \sum_{s<t, s\in\mathcal{G}^{(i)}} x_{i_s} x_{i_s}^\top, \\
\hat{\theta}_i^{(t)} &= \left(V_i^{(t)}\right)^{-1} \sum_{s<t, s\in\mathcal{G}^{(i)}} x_{i_s} y_{i,i_s}.
\end{aligned}
\tag{8}
$$

### 4.3. Theoretical Analysis

We now present the regret upper bound for each player by following the AdECO algorithm.

**Theorem 4.1** (Upper Bound for Adversarial Contexts). *Following the AdECO algorithm with parameter $\Delta$ and $\varepsilon$, the $\alpha$-approximate $\Delta$-optimal stable regret for $p_i \in \mathcal{N}$ is*

$$
Reg_i^{\alpha,\Delta}(T) = \mathcal{O}\left(\frac{Nd\log^2 T}{(\Delta-\varepsilon)^2} + \frac{\Delta+\varepsilon}{2}\cdot T\right).
\tag{9}
$$

$\Delta$ *is an algorithmic input, and the above regret bound holds uniformly for any choice of $\Delta$. By setting $\Delta = \mathcal{O}(T^{-1/3})$, $\varepsilon = \Delta/2$, we have the regret upper bound as $\mathcal{O}(T^{2/3})$.*

We provide the proof sketch for Theorem 4.1 as follows, while the detailed proof is deferred to Appendix G.

*Proof Sketch.* Similar to the proof in Theorem 3.1, in the exploration phase, by elliptical potential lemma (Carpentier et al., 2020), we have that $|\mathcal{G}| \leq \frac{Nd\log\left(\frac{T+d\lambda}{d\lambda}\right)}{\xi^2} = \frac{16\eta^2 Nd\log\left(\frac{T+d}{d}\right)}{(\Delta-\varepsilon)^2} = \mathcal{O}\left(\frac{Nd\log^2 T}{(\Delta-\varepsilon)^2}\right)$.

In every exploitation round, if we call the Gale-Shapley algorithm, we incur 0 or negative regret since it returns the unique player-optimal stable matching when we have correct estimation of the preference ranking, while if we call the $\alpha$-approximation oracle with $\hat{U}(t)$ and instability tolerance $\frac{\Delta+\varepsilon}{2}$, we are in the scenario where $\delta_{\min}$ is small and we incur $\frac{\Delta+\varepsilon}{2}$ approximation regret.

Therefore, the final regret is $\mathcal{O}\left(\frac{Nd\log^2 T}{(\Delta-\varepsilon)^2} + \frac{\Delta+\varepsilon}{2}\cdot T\right)$. $\square$

*Remark* 4.1. The $\alpha$-independent regret in Theorem 4.1 consists of exploration and exploitation terms. In the exploration phase with small $\delta_{\min}$, the benchmark is $\alpha U_i^\varepsilon(t)$, making the regret scales linearly with $\alpha$. Since $\alpha \leq 1$, we adopt the looser upper bound to keep exploration regret independent of $\alpha$. In the exploitation phase, the approximation oracle's guarantee (Definition 4) bounds the per-round loss in small-gap regimes by $(\Delta + \varepsilon)/2$, which is likewise independent of $\alpha$.

*Remark* 4.2. Inspired by the adversarial context results, we could call back the stochastic context case, if the distribution of $\delta_{\min}$ satisfies Assumption 5, then by the asymptotic regret upper bound (Theorem 3.2), we know that by setting a

maximum exploration time of $\mathcal{O}(T^{\frac{2}{3}+\zeta})$ for some small $\zeta$, it suffices to finish exploration. And if the exploration could not stop until $\mathcal{O}(T^{\frac{2}{3}+\zeta})$ rounds, it is probably that the distribution of $\delta_{\min}$ is of poor condition, i.e., the probability of having small enough $\delta_{\min}$ is rather high. In this case, we could switch between the GS oracle and the approximation oracle as we propose in the adversarial context case, and gain a meaningful regret guarantee with respect to the $\alpha$-approximate $\Delta$-optimal stable regret.

## 5. Numerical Experiments

In this section, we empirically validate the convergence of BARB (stochastic contexts) and AdECO (adversarial contexts). For the stochastic setting, we also benchmark performance against Batched-ETC (Appendix C) and ETC (Li et al., 2022).

Each experiment runs for $T = 200k$ rounds, and results are averaged over 20 independent trials. Besides the averaged cumulative regret, we also plot the standard error, defined as the standard deviation divided by $\sqrt{20}$.

Due to space constraints, we present results for the stochastic setting with a large minimum preference gap $\Delta_{\min}$ and a small minimum eigenvalue of the context covariance matrix; additional comparisons are provided in Appendix H.

We evaluate player-optimal stable regret in a market with $N = 4$ players and $K = 4$ arms. Contexts have dimension $d = 3$. Each arm's preference over players is a random permutation of $\{1, \cdots, N\}$. For preference and context generation, we first sample two orthonormal vectors $v, z \in \mathbb{R}^d$. We set $\theta_i = v$ for all $i \in [N]$, ensuring $\|\theta_i\| = 1$. Next, we define equally spaced values $\gamma_j = \frac{j-1}{K-1}$ for $j \in [K]$ and set $\tilde{x}_j = \gamma_j v + \sqrt{1-\gamma_j^2} z$. At each time $t$, the $k$-th entry of the context vector $x_j(t)$ is independently drawn from a Gaussian distribution with mean $\tilde{x}_{j,k}$ and variance 0.01. The ground-truth utility for player $p_i$ and arm $a_j$ at time $t$ is $U_{i,j}(t) = \theta_i^\top x_j(t)$. When a match occurs, a reward is sampled from the Gaussian distribution with mean $U_{i,j}(t)$ and variance 0.05. Then we would truncate it so that the absolute value of the observed reward is bounded by $B_y = 1$.

The ridge penalty parameter is set to $\lambda = 1$ for all algorithms. For ETC, the exploration length is $h = 5000$; for Batched-ETC, the initial exploration length is $T_1 = 100$; and for BARB, the initial candidate gap is $\Delta_1 = 0.5$.

We present the player-optimal stable regret for each player in Figure 1. The results demonstrate that BARB outperforms both ETC and Batched-ETC in settings where the minimum eigenvalue $\lambda_{\min}$ of the context covariance matrix is small. In Appendix H, we provide a complementary example where $\lambda_{\min}$ is large, using the same algorithmic

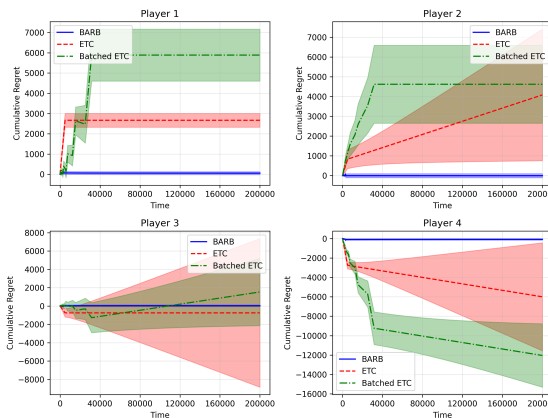

*Figure 1.* Experimental Comparison of BARB, Batched-ETC, and ETC in a market with 4 players and 4 arms.

parameters. There, BARB achieves performance comparable to ETC. Since the covariance structure of contexts is typically unknown in practice, the fully adaptive design of BARB and the fact that its regret upper bound does not depend on the covariance structure of the context distribution offer more robust performance across varying problem instances than the parameter-dependent ETC.

# 6. Extension in Decentralized Contextual Matching Markets

Algorithm 1 and 2 operate in centralized contextual matching markets, where a platform coordinates all players. In practice, however, decentralized markets where each player acts independently are more relevant. To accommodate decentralization within our existing framework, we introduce the following assumption for environment identification.

**Assumption 6.** Let $\mathcal{E}$ denote the set of environments. For any two distinct environments $e, e' \in \mathcal{E}$, the top-$N$ ranked arms for each player $p_i$ are distinct; that is, the ranking vectors $\nu_i^e$ and $v\nu_i^{e'}$ differ in at least one of the first $N$ entries.

Here, an environment is defined by its preference ranking: two contexts with the same ranking belong to the same environment. This assumption enables environment identification using only individual players' preference rankings, without direct communication. While it also appears in Parikh et al. (2024), our requirement is weaker: they require each player to know the exact number of possible environments, whereas we allow it to be unknown. Furthermore, Parikh et al. (2024) shows that this assumption is necessary by providing an example where sublinear regret is impossible without it.

Additionally, each player $p_i$ maintains two market-sharable flags: $F_i$ indicates whether the Mahalanobis norm $\|x_j\|_{V_i^{-1}}$ for any arm $a_j$ falls below a given threshold, and $F_i'$ indi-

cates whether the confidence intervals are disjoint in the current exploitation round.

For simplicity, we focus on the stochastic contexts setting. In each round, when multiple players propose to the same arm, only the highest-ranked player in that arm's preference profile receives a reward; the others receive zero.

The algorithm is presented from player $p_i$'s perspective. The candidate gap $\Delta_k$ is public, so $\xi_k$ can be inferred locally. At round $t_k$, after observing each arm $a_j$'s context, $p_i$ computes $\|x_j(t_k)\|_{V_i^{-1}}$ and compares it to $\xi_k$. If any $x_j$ yields a norm exceeding the threshold, $F_i$ is set to True.

If any player has $F_i = \text{True}$, the system explores. In that case, $p_i$ proposes to the arm with the largest $\|x_j(t_k)\|_{V_i^{-1}}$ among those exceeding $\xi_k$. If none exists, $p_i$ proposes nothing that round. Updates for $V_i$ and $\hat{\theta}_i$ follow Eq. (5).

If all players have $F_i = \text{False}$, the system exploits. Player $p_i$ computes the estimated utilities and confidence intervals for every arm. If all CIs are disjoint, set $F_i' = \text{False}$; otherwise, set $F_i' = \text{True}$. If any player has $F_i' = \text{True}$, the counter $N_{i,k} \leftarrow N_{i,k} + 1$. This counter is maintained locally but is identical across players. The player proposes to an arbitrary arm. Once $N_{i,k}$ exceeds the BARB threshold, the algorithm moves to the next batch, shrinking $\Delta_k$ by a factor of $\sqrt{2}$. If instead all players have $F_i' = \text{False}$, the player identifies the environment from the current estimated utilities and proposes to the highest-ranked arm that has not yet rejected $p_i$ in that environment.

With the above implementation, the exploration regret matches that of BARB, while the exploitation regret incurs an additional $N^2 E$ term, where $E$ is the number of environments occuring over the horizon. The algorithm does not require knowledge of $E$, and this quantity is used only for theoretical analysis. Since this additional term is independent of $T$, the overall regret order remains unchanged.

# 7. Conclusions and Discussions

This paper studies contextual bandits in matching markets under a linear utility model, with the objective of minimizing individual *player-optimal stable regret*. The problem is complex: minor context shifts can drastically reconfigure the stable matching benchmark, causing disproportionate regret. We address this for both *stochastic and adversarial contexts*, introducing a novel *minimum preference gap* measure for the former and a tractable regret notion for the latter. We develop fully adaptive algorithms with theoretical guarantees for both settings. For the stochastic case, we also provide matched instance-independent regret bounds. While the proposed framework assumes linear contextual utilities, extending it to accommodate markets with more complex structures would be valuable.

## Acknowledgement

This research was supported by Israel Science Foundation research grant (ISF's No. 4118/25) and the Maimonides Fund's Future Scientists Center. Vianney Perchet's research was supported in part by the French National Research Agency (ANR) in the framework of the PEPR IA FOUNDRY project (ANR-23-PEIA-0003) and through the grant DOOM ANR-23-CE23-0002. It was also funded by the European Union (ERC, Ocean, 101071601). Views and opinions expressed are however those of the author(s) only and do not necessarily reflect those of the European Union or the European Research Council Executive Agency. Neither the European Union nor the granting authority can be held responsible for them.

## Impact Statement

This paper presents work whose goal is to advance the field of Machine Learning. There are many potential societal consequences of our work, none of which we feel must be specifically highlighted here.

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

# A. Batched Adaptive Regret-Balancing Algorithm

The full version of Batched Adaptive Regret-Balancing (BARB) algorithm is presented in Algorithm 3.

---

**Algorithm 3** Batched Adaptive Regret-Balancing (BARB), full version

---

**Input:** Time horizon $T$, preference profile $(P_a)_{a \in \mathcal{K}}$, context dimension $d$, parameter $\eta$, initial candidate gap $\Delta_1$, ridge penalty parameter $\lambda$.

1: $t \leftarrow 1$.
2: **for** $k = 1, 2, \cdots$ **do**
3:      $\xi_k \leftarrow \frac{\Delta_k}{\eta}$.
4:      $\boldsymbol{V}_i^{(1)} \leftarrow \lambda \boldsymbol{I}_{d \times d}, \hat{\theta}_i^{(1)} \leftarrow \boldsymbol{0}_d, \forall i \in [N]$.
5:      $N_k \leftarrow 0$. {Number of rounds with overlapping CIs during exploitation.}
6:      **for** $t_k = 1, 2, \cdots$ **do**
7:          Observe context $\mathrm{x}_j(t_k)$ for every arm $a_j, \forall j \in [K]$.
8:          **if** $\exists i \in [N], j \in [K]$, s.t. $\|\mathrm{x}_j(t_k)\|_{(\boldsymbol{V}_i^{(t_k)})^{-1}} > \xi_k$ **then** {Exploration}
9:              Collect all $(i,j)$ such that $\|\mathrm{x}_j(t_k)\|_{(\boldsymbol{V}_i^{(t_k)})^{-1}} > \xi_k$, form a bipartite graph $G_{t_k}$.
10:             Find a **maximum cardinality matching** $\mu^c$ on $G_{t_k}$, let $M_{t_k}^c := \{i \in [N] : i \text{ is matched in } \mu^c\}$ match

$$\mu_{t_k}(i) \leftarrow \begin{cases} \mu^c(i) & \text{if } i \in M_{t_k}^c, \\ \emptyset & \text{otherwise.} \end{cases}$$

11:             Update $\boldsymbol{V}_i^{(t_k+1)}$ and $\hat{\theta}_i^{(t_k+1)}, \forall i \in M_{t_k}^c$ according to Eq.(5).
12:             $\boldsymbol{V}_i^{(t_k)} \leftarrow \boldsymbol{V}_i^{(t_k-1)}$ and $\hat{\theta}_i^{(t_k)} \leftarrow \hat{\theta}_i^{(t_k-1)}, \forall i \in [N] \setminus M_{t_k}^c$.
13:          **else**{Exploitation}
14:             $\hat{\theta}_i^{(t_k)} \leftarrow \hat{\theta}_i^{(t_k-1)}, \forall i \in [N]$.
15:             $\hat{U}_{i,j}(t_k) = (\hat{\theta}_i^{(t_k)})^\top \mathrm{x}_j(t_k), F_i \leftarrow$ False.
16:             $\mu_{t_k}(i) \leftarrow \mu^{GS}(i), \forall i \in [N]$. (Input $\hat{\boldsymbol{U}}(t_k)$ and $(P_a)_{a \in \mathcal{K}}$ for **Gale-Shapley**)
17:             **for** $i = 1, 2, \cdots, N$ **do**
18:                 $\hat{\boldsymbol{U}}_{i,\cdot}^{\text{sort}}(t_k) \leftarrow \text{Sort}\left(\hat{\boldsymbol{U}}_{i,\cdot}(t_k), \text{decreasing}\right)$
19:                 $\Delta_{i,\min} \leftarrow \min\left\{\hat{\boldsymbol{U}}_{i,j}^{\text{sort}}(t_k) - \hat{\boldsymbol{U}}_{i,j+1}^{\text{sort}}(t_k)\right\}, j \in [N]$
20:                 **if** $\Delta_{i,\min} > 2\Delta_k$ **then**
21:                     $F_i \leftarrow$ True
22:                 **end if**
23:             **end for**
24:             **if** $\exists i \in [N]$ s.t. $F_i ==$ False **then** {CIs are overlapped}
25:                 $N_k \leftarrow N_k + 1$.
26:             **end if**
27:          **end if**
28:          $t \leftarrow t + 1$.
29:          **if** $t > T$ **then**
30:             Stop and return.
31:          **end if**
32:          **if** $N_k > \frac{3 \log T}{16 \Delta_k^2}$ **then** {Exploitation regret exceeds exploration regret}
33:             Break the loop and enter the next batch, set $\Delta_{k+1} \leftarrow \frac{\Delta_k}{\sqrt{2}}$,
34:          **end if**
35:      **end for**
36: **end for**

---

# B. Proof of Theorem 3.1

*Proof.* Denote $k^*$ as the largest batch number of Algorithm 1. That is to say, the counter $N_k$ would not meet the pre-defined threshold in batch $k^*$. In batch $k$, given $\Delta_k$, for the exploration phase, denote the estimation indices for player $p_i$ as $\mathcal{G}_k^{(i)} = \left\{ t_{k_1}^{(i)}, \cdots, t_{k_{|\mathcal{G}_k^{(i)}|}}^{(i)} \right\}$. Notice that for $t_k \in \mathcal{G}_k^{(i)}$, we have $\|\mathbf{x}_j(t_k)\|_{\left(\mathbf{V}_i^{(t_k)}\right)^{-1}} > \xi_k = \frac{\Delta_k}{\eta}$ for $j = \mu_{t_k}(i)$, and we only use the contexts and matching reward from $\mathcal{G}_k^{(i)}$ to update the Gram matrix $\mathbf{V}_i$ and the estimation of the preference parameter $\hat{\theta}_i$. Take $\mathcal{G}_k$ as the set of rounds in which the market explores, we have $\mathcal{G}_k = \cup_{i \in [N]} \mathcal{G}_k^{(i)}$. Define $\mathcal{F}_k = \left\{ \exists t_k \notin \mathcal{G}_k, i \in [N], j \in [K] : |\hat{\mathbf{U}}_{i,j}(t) - \mathbf{U}_{i,j}(t)| > \Delta_k \right\}$ as the bad event that some preference of players over arms is not estimated well during the exploitation phase. When $k > k^*$, we define $\mathbb{P}(\mathcal{F}_k) = 0$. For the exploitation phase, using $\Delta_k$ as the width for the confidence interval constructed for the estimated utility, let $\mathcal{F}_d^{(t)} := \left\{ \hat{\mathbf{U}}_{i,j}^{\text{sort}}(t) - \hat{\mathbf{U}}_{i,j+1}^{\text{sort}}(t) > 2\Delta_k, \forall j \in [N], \forall i \in [N] \right\} \cap \left\{ \hat{\mathbf{U}}_{i,N}^{\text{sort}}(t) - \hat{\mathbf{U}}_{i,j}^{\text{sort}}(t) > 2\Delta_k, \forall j \in [N+1, K], \forall i \in [N] \right\}$ as the event that the first $(N+1)$-ranked arms have disjoint confidence intervals for any player $p_i$ at time $t$. Denote $T_k'$ as the total length of the $k$-th batch. Denote $\mathcal{F}_{\Delta_{k^*}} := \left\{ \Delta_{k^*} \geq \frac{\Delta_{\min}}{4\sqrt{2}} \right\}$ as the event that the algorithm stops at an early batch.

The optimal stable regret for each player $p_i$ by following BARB (Algorithm 1) satisfies

$$Reg_i(T) = \mathbb{E}\left[\sum_{t=1}^{T} \left(\boldsymbol{U}_i^*(t) - \boldsymbol{U}_{i,\mu_t(i)}(t)\right)\right] \tag{10}$$

$$= \mathbb{E}\left[\sum_{t=1}^{T} \left(\boldsymbol{U}_i^*(t) - \boldsymbol{U}_{i,\mu_t(i)}(t)\right) \mathbb{1}\{\forall k \in [T] : \neg\mathcal{F}_k\}\right] + \mathbb{E}\left[\sum_{t=1}^{T} \left(\boldsymbol{U}_i^*(t) - \boldsymbol{U}_{i,\mu_t(i)}(t)\right) \mathbb{1}\{\exists k \in [T] : \mathcal{F}_k\}\right]$$

$$\leq \mathbb{E}\left[\sum_{t=1}^{T} \left(\boldsymbol{U}_i^*(t) - \boldsymbol{U}_{i,\mu_t(i)}(t)\right) \mathbb{1}\{\forall k \in [T] : \neg\mathcal{F}_k\}\right] + B_y T \mathbb{P}\left(\exists k \in [T] : \mathcal{F}_k\right) \tag{11}$$

$$\leq \mathbb{E}\left[\sum_{t=1}^{T} \left(\boldsymbol{U}_i^*(t) - \boldsymbol{U}_{i,\mu_t(i)}(t)\right) \mathbb{1}\{\forall k \in [T] : \neg\mathcal{F}_k\}\right] + B_y T \cdot \frac{N}{T} \tag{12}$$

$$= \mathbb{E}\left[\sum_{t=1}^{T} \left(\boldsymbol{U}_i^*(t) - \boldsymbol{U}_{i,\mu_t(i)}(t)\right) \mathbb{1}\{\{\forall k \in [k^*] : \neg\mathcal{F}_k\} \cap \mathcal{F}_{\Delta_{k^*}}\}\right]$$

$$\quad + \mathbb{E}\left[\sum_{t=1}^{T} \left(\boldsymbol{U}_i^*(t) - \boldsymbol{U}_{i,\mu_t(i)}(t)\right) \mathbb{1}\{\{\forall k \in [k^*] : \neg\mathcal{F}_k\} \cap \neg\mathcal{F}_{\Delta_{k^*}}\}\right] + B_y N$$

$$\leq \mathbb{E}\left[\sum_{t=1}^{T} \left(\boldsymbol{U}_i^*(t) - \boldsymbol{U}_{i,\mu_t(i)}(t)\right) \mathbb{1}\{\{\forall k \in [k^*] : \neg\mathcal{F}_k\} \cap \mathcal{F}_{\Delta_{k^*}}\}\right]$$

$$\quad + B_y T \mathbb{P}\left(\{\forall k \in [k^*] : \neg\mathcal{F}_k\} \cap \neg\mathcal{F}_{\Delta_{k^*}}\right) + B_y N \tag{13}$$

$$\leq \mathbb{E}\left[\sum_{k=1}^{k^*}\left[\sum_{t_k \in \mathcal{G}_k} \left(\boldsymbol{U}_i^*(t_k) - \boldsymbol{U}_{i,\mu_t(i)}(t_k)\right) + \sum_{t_k \notin \mathcal{G}_k} \left(\boldsymbol{U}_i^*(t_k) - \boldsymbol{U}_{i,\mu_t(i)}(t_k)\right)\right] \mathbb{1}\{\{\forall k \in [k^*] : \neg\mathcal{F}_k\} \cap \mathcal{F}_{\Delta_{k^*}}\}\right]$$

$$\quad + B_y T \cdot \frac{1}{T} + B_y N \tag{14}$$

$$\leq \mathbb{E}\left[\sum_{k=1}^{k^*}\left[B_y |\mathcal{G}_k| \mathbb{1}\{\{\forall k \in [k^*] : \neg\mathcal{F}_k\} \cap \mathcal{F}_{\Delta_{k^*}}\}\right.\right. \tag{15}$$

$$\quad + \sum_{t_k \notin \mathcal{G}_k}\left[\left(\boldsymbol{U}_i^*(t_k) - \boldsymbol{U}_{i,\mu_t(i)}(t_k)\right) \mathbb{1}\left\{\{\forall k \in [k^*] : \neg\mathcal{F}_k\} \cap \mathcal{F}_{\Delta_{k^*}} \cap \mathcal{F}_d^{(t_k)}\right\}\right.$$

$$\quad \left.\left.\left. + \left(\boldsymbol{U}_i^*(t_k) - \boldsymbol{U}_{i,\mu_t(i)}(t_k)\right) \mathbb{1}\left\{\{\forall k \in [k^*] : \neg\mathcal{F}_k\} \cap \mathcal{F}_{\Delta_{k^*}} \cap \neg\mathcal{F}_d^{(t_k)}\right\}\right]\right]\right] + B_y(1 + N)$$

$$\leq \mathbb{E}\left[\sum_{k=1}^{k^*}\left[B_y |\mathcal{G}_k| \mathbb{1}\{\{\forall k \in [k^*] : \neg\mathcal{F}_k\} \cap \mathcal{F}_{\Delta_{k^*}}\}\right.\right.$$

$$\quad \left.\left. + \sum_{t_k \notin \mathcal{G}_k} \left(\boldsymbol{U}_i^*(t_k) - \boldsymbol{U}_{i,\mu_t(i)}(t_k)\right) \mathbb{1}\left\{\{\forall k \in [k^*] : \neg\mathcal{F}_k\} \cap \mathcal{F}_{\Delta_{k^*}} \cap \neg\mathcal{F}_d^{(t_k)}\right\}\right]\right] + B_y(1 + N) \tag{16}$$

$$\leq B_y \mathbb{E}\left[\sum_{k=1}^{k^*} [|\mathcal{G}_k| + N_k] \mathbb{1}\{\{\forall k \in [k^*] : \neg\mathcal{F}_k\} \cap \mathcal{F}_{\Delta_{k^*}}\}\right] + B_y(1 + N) \tag{17}$$

$$\leq B_y \mathbb{E}\left[\sum_{k=1}^{k^*} \left(\frac{\eta^2 N d \log\left(\frac{T+d\lambda}{d\lambda}\right)}{\Delta_k^2} + \frac{3\log T}{16\Delta_k^2}\right) \mathbb{1}\{\{\forall k \in [k^*] : \neg\mathcal{F}_k\} \cap \mathcal{F}_{\Delta_{k^*}}\}\right] + B_y(1 + N) \tag{18}$$

$$\leq 2B_y \mathbb{E}\left[\left(\frac{\eta^2 N d \log\left(\frac{T+d\lambda}{d\lambda}\right)}{\Delta_{k^*}^2} + \frac{3\log T}{16\Delta_{k^*}^2}\right) \mathbb{1}\{\{\forall k \in [k^*] : \neg\mathcal{F}_k\} \cap \mathcal{F}_{\Delta_{k^*}}\}\right] + B_y(1 + N) \tag{19}$$

$$\leq 64 B_y \left(\frac{\eta^2 N d \log\left(\frac{T+d\lambda}{d\lambda}\right)}{\Delta_{\min}^2} + \frac{3\log T}{\Delta_{\min}^2}\right) + B_y(1 + N) = \mathcal{O}\left(\frac{\log^2 T}{\Delta_{\min}^2} + \frac{\log T}{\Delta_{\min}^2}\right) \tag{20}$$

Eq.(10) holds according to the assumption that the noises are i.i.d. with mean 0. Eq.(11), (13), and (15) come from the fact that the expected per-round regret is $\mathbb{E}\left[\boldsymbol{U}_i^*(t) - y_i(t)\right] = \boldsymbol{U}_i^*(t) - \boldsymbol{U}_{i,\mu_t(i)}(t)$ and both terms lie in $[-B_\theta B_x, B_\theta B_x]$ under Assumptions 1 and 3, and hence the expected regret per round is bounded by $2B_\theta B_x = B_y$. Eq.(12) relies on Lemma B.1. Eq.(14) holds according to Lemma B.2 and the fact that $\mathbb{P}\left(\{\forall k \in [k^*] : \neg\mathcal{F}_k\} \cap \neg\mathcal{F}_{\Delta_{k^*}}\right) \leq \mathbb{P}\left(\neg\mathcal{F}_{\Delta_{k^*}}\right)$. Eq. (16) holds because, with Lemma B.3, under the good event $\neg\mathcal{F}_k$ (ensuring all utilities are well-estimated) and the event $\mathcal{F}_d^{(t_k)}$ (ensuring the top $(N+1)$-ranked confidence intervals are disjoint at time $t_k$), the Gale-Shapley subroutine invoked by Algorithm 1 is guaranteed to output the OSS for that round. Eq.(17) follows from the algorithm setting that $N_k$ counts the number of times during exploitation that the top $(N+1)$-ranked CIs are overlapped. Eq.(18) holds according to Lemma B.4 and the algorithm setting that when $N_k > \frac{3\log T}{16\Delta_k^2}$, we would move to the next batch. Eq.(19) holds based on Lemma B.5. Eq.(20) follows from the parameter setting that $\eta = \mathcal{O}\left(\sqrt{\log T}\right)$.

$\square$

**Lemma B.1.** *Let* $\delta = T^{-2}$, $\eta = R\sqrt{d\log\left(\frac{1+TB_x^2/\lambda}{\delta}\right)} + \lambda^{1/2}B_\theta$, *we have* $\mathbb{P}\left(\exists k \in [T] : \mathcal{F}_k\right) \leq \frac{N}{T}$.

*Proof.*

$$\mathbb{P}\left(\exists k \in [T] : \mathcal{F}_k\right) = \mathbb{P}\left(\exists k \in [T], \exists t_k \notin \mathcal{G}_k, i \in [N], j \in [K] : |\hat{\boldsymbol{U}}_{i,j}(t_k) - \boldsymbol{U}_{i,j}(t_k)| > \Delta_k\right)$$

$$\leq \sum_{k=1}^{T}\sum_{i=1}^{N} \mathbb{P}\left(\exists j \in [K], \exists t_k \notin \mathcal{G}_k : |\hat{\boldsymbol{U}}_{i,j}(t_k) - \boldsymbol{U}_{i,j}(t_k)| > \Delta_k\right) \tag{21}$$

$$\leq \sum_{k=1}^{T}\sum_{i=1}^{N} \mathbb{P}\left(\exists j \in [K], \exists t_k \notin \mathcal{G}_k : \|\hat{\theta}_i - \theta_i\|_{\boldsymbol{V}_i^{(t_k)}} \|\mathbf{x}_j(t)\|_{\left(\boldsymbol{V}_i^{(t_k)}\right)^{-1}} > \Delta_k\right) \tag{22}$$

$$\leq \sum_{k=1}^{T}\sum_{i=1}^{N} \mathbb{P}\left(\exists t_k \notin \mathcal{G}_k : \|\hat{\theta}_i - \theta_i\|_{\boldsymbol{V}_i^{(t_k)}} > \eta\right) \tag{23}$$

$$\leq TN\delta = \frac{N}{T}. \tag{24}$$

Eq.(21) follows from the union bound. Eq.(22) holds according to Cauchy-Schwarz inequality. Eq.(23) comes from the fact that for any $t_k \notin \mathcal{G}_k$, we have $\|\mathbf{x}_j(t_k)\|_{\left(\boldsymbol{V}_i^{(t_k)}\right)^{-1}} \leq \xi_k = \frac{\Delta_k}{\eta}$ in Algorithm 1. Eq.(24) holds according to Lemma J.2 (and notice it trivially holds if $k > k^*$). $\square$

**Lemma B.2.** *It holds that* $\mathbb{P}\left(\{\forall k \in [k^*] : \neg\mathcal{F}_k\} \cap \neg\mathcal{F}_{\Delta_{k^*}}\right) \leq \frac{1}{T}$.

*Proof.* Let $k_0$ be the first $k$ for which $\Delta_k < \Delta_{\min}/4$, i.e., $\Delta_{k_0} \geq \Delta_{\min}/4\sqrt{2}$ since $\Delta_{k+1} = \Delta_k/\sqrt{2}$. Since in batch $k_0$, the threshold for determining moving to the next batch is $N_{k_0} \geq \frac{3\log T}{16\Delta_{k_0}^2}$, we have

$$\neg\mathcal{F}_{\Delta_{k^*}} = \left\{\Delta_{k^*} < \frac{\Delta_{\min}}{4\sqrt{2}}\right\} \subseteq \left\{N_{k_0} \geq \frac{3\log T}{16\Delta_{k_0}^2}\right\}.$$

Therefore, to prove the claim, it suffices to show that

$$\mathbb{P}\left(\{\forall k \in [k^*] : \neg\mathcal{F}_k\} \cap \left\{N_{k_0} \geq \frac{3\log T}{16\Delta_{k_0}^2}\right\}\right) \leq \frac{1}{T}.$$

Let $\tilde{N}_k$ be the number of rounds of the $k$-th batch such that $\delta_{\min}^{(t_k)} \leq 4\Delta_k$. Under the event $\neg\mathcal{F}_{k_0}$, when there exists overlap CIs with CI radius $\Delta_{k_0}$ at time $t_{k_0}$, there must exist a player $p_i$ and two arms $a_j$ and $a_{j'}$ with $\hat{\boldsymbol{U}}_{i,j}(t_{k_0}) \geq \hat{\boldsymbol{U}}_{i,j'}(t_{k_0})$, such that

$\hat{U}_{i,j}(t_{k_0}) - \Delta_{k_0} \leq \hat{U}_{i,j'}(t_{k_0}) + \Delta_{k_0}$, and hence $U_{i,j}(t_{k_0}) \leq \hat{U}_{i,j}(t_{k_0}) + \Delta_{k_0} \leq \hat{U}_{i,j'}(t_{k_0}) + 3\Delta_{k_0} \leq U_{i,j'}(t_{k_0}) + 4\Delta_{k_0}$, which implies $\delta_{\min}^{(t_{k_0})} \leq 4\Delta_{k_0}$. Therefore, under the event $\neg \mathcal{F}_{k_0}$, we have $N_{k_0} \leq \tilde{N}_{k_0}$, and hence it suffices to show that

$$\mathbb{P}\left( \{\forall k \in [k^*] : \neg \mathcal{F}_k\} \cap \left\{ \tilde{N}_{k_0} \geq \frac{3\log T}{16\Delta_{k_0}^2} \right\} \right) \leq \mathbb{P}\left( \tilde{N}_{k_0} \geq \frac{3\log T}{16\Delta_{k_0}^2} \right) \leq \frac{1}{T}.$$

As $\Delta_{k_0} < \Delta_{\min}$, and recall the definition of the minimum preference gap $\Delta_{\min}$ (Definition 2), for every round $t_{k_0}$ in batch $k_0$, we have

$$\mathbb{P}\left( \delta_{\min}^{(t_{k_0})} \leq 4\Delta_{k_0} \right) \leq \mathbb{P}\left( \delta_{\min}^{(t_{k_0})} \leq \Delta_{\min} \right) \leq \frac{\log T}{T\Delta_{\min}^2}.$$

Denote the length of the $k$-th batch as $T'_k$, we have

$$\mathbb{E}\left[ \tilde{N}_{k_0} \right] = \mathbb{E}\left[ \sum_{t_{k_0}=1}^{T'_{k_0}} \mathbb{1}\left\{ \delta_{\min}^{(t_{k_0})} \leq 4\Delta_{k_0} \right\} \right] \leq T \cdot \frac{\log T}{T\Delta_{\min}^2} = \frac{\log T}{\Delta_{\min}^2}.$$

Let $\rho$ satisfy $(1+\rho)\mathbb{E}\left[ \tilde{N}_{k_0} \right] = \frac{3\log T}{\Delta_{\min}^2}$, then $\rho\mathbb{E}\left[ \tilde{N}_{k_0} \right] \geq \frac{2\log T}{\Delta_{\min}^2}$ and $\rho \geq 2$. Therefore,

$$\begin{aligned}
\mathbb{P}\left( \tilde{N}_{k_0} \geq \frac{3\log T}{16\Delta_{k_0}^2} \right) &\leq \mathbb{P}\left( \tilde{N}_{k_0} \geq \frac{3\log T}{\Delta_{\min}^2} \right) \\
&= \mathbb{P}\left( \tilde{N}_{k_0} \geq (1+\rho)\mathbb{E}\left[ \tilde{N}_{k_0} \right] \right) \\
&\leq \exp\left( -\frac{\rho^2\mathbb{E}\left[ \tilde{N}_{k_0} \right]}{2+\rho} \right) \qquad\qquad (25) \\
&\leq \exp\left( -\frac{\rho\mathbb{E}\left[ \tilde{N}_{k_0} \right]}{2} \right) \\
&\leq \exp\left( -\frac{\log T}{\Delta_{\min}^2} \right) \\
&\leq \frac{1}{T}. \qquad\qquad\qquad\qquad\qquad (26)
\end{aligned}$$

Eq.(25) holds according to the multiplicative Chernoff bound (Lemma J.3) by noticing that $\tilde{N}_{k_0}$ is the sum of independent Bernoulli trials. Eq.(26) follows from the assumptions that $B_y \leq 1$ so that $\Delta_{\min} \leq 1$. $\qquad\square$

**Lemma B.3.** *In the Gale-Shapley algorithm, at most $N$ proposals are made per player before termination. Consequently, each player's optimal stable match must be among its top $N$ preferred arms.*

*Proof.* Within the Gale-Shapley algorithm, a proposal from a player results in the arm being tentatively matched to that player. To prove the claim, assume by contradiction that more than $N$ distinct arms are proposed to. This would imply that at least $N+1$ players are involved in proposals. However, since there are exactly $N$ players, once $N$ proposals have been made, every player must be tentatively matched. At this point, the algorithm terminates. Therefore, the maximum number of distinct arms that can be proposed to is $N$. Since each player proposes to arms strictly in descending order of preference, it follows that a player's optimal stable match must be among its top $N$ ranked arms. $\qquad\square$

**Lemma B.4.** *In batch $k$, given $\eta$ and the ridge penalty parameter $\lambda$, for a given $\Delta_k$, let $\mathcal{G}_k$ denote the set of rounds in which the system explores. Almost surely, the length of the exploration phase $\mathcal{G}_k$ is bounded as*

$$|\mathcal{G}_k| \leq \frac{\eta^2 N d \log\left( \frac{T+d\lambda}{d\lambda} \right)}{\Delta_k^2}. \qquad\qquad (27)$$

*Proof.* The following proof is adapted from the proof in Tullii et al. (2024, Lemma 1).

Let $\mathcal{G}_k^{(i)} = \{t_k : p_i \in M_{t_k}^c\}$ denote the set of rounds in which player $p_i$ is involved in the maximum cardinality matching, and updates its estimation for the Gram matrix and the preference parameter. For player $p_i$, for all $t_k \in \mathcal{G}_k^{(i)}$, we can write the Gram matrix as

$$\boldsymbol{V}_i^{(t_k)} = \lambda \boldsymbol{I}_d + \sum_{s < t_k, s \in \mathcal{G}_k^{(i)}} \mathrm{x}_{i_s} \mathrm{x}_{i_s}^\top,$$

where $i_s = \mu_s(i)$. By the elliptical potential lemma (Lemma J.1), we have that

$$\sum_{t_k \in \mathcal{G}_k^{(i)}} \|\mathrm{x}_{i_{t_k}}\|_{\left(\boldsymbol{V}_i^{(t_k)}\right)^{-1}} \leq \sqrt{|\mathcal{G}_k^{(i)}| d \log\left(\frac{|\mathcal{G}_k^{(i)}| + d\lambda}{d\lambda}\right)}.$$

Since for all $t_k \in \mathcal{G}_k^{(i)}$, we have that $\|\mathrm{x}_{i_{t_k}}\|_{\left(\boldsymbol{V}_i^{(t_k)}\right)^{-1}} \geq \xi_k = \frac{\Delta_k}{\eta}$, this implies that

$$|\mathcal{G}_k^{(i)}| \cdot \frac{\Delta_k}{\eta} \leq \sqrt{|\mathcal{G}_k^{(i)}| d \log\left(\frac{|\mathcal{G}_k^{(i)}| + d\lambda}{d\lambda}\right)}.$$

Now, almost surely, $|\mathcal{G}_k^{(i)}| \leq T$. Using this bound and reorganizing the inequality leads to

$$|\mathcal{G}_k^{(i)}| \leq \frac{\eta^2 d \log\left(\frac{T + d\lambda}{d\lambda}\right)}{\Delta_k^2}.$$

When the system explores at time $t_k$, at least one player $p_i$ and one arm $a_j$ triggers the condition that $\|\mathrm{x}_j(t_k)\|_{\left(\boldsymbol{V}_i^{(t_k)}\right)^{-1}} > \xi_k$ and is involved in the maximum cardinality matching. Therefore, we have $\mathcal{G}_k = \cup_{i \in [N]} \mathcal{G}_k^{(i)}$, and hence

$$|\mathcal{G}_k| \leq \sum_{i \in [N]} |\mathcal{G}_k^{(i)}| \leq \frac{\eta^2 N d \log\left(\frac{T + d\lambda}{d\lambda}\right)}{\Delta_k^2}.$$

$\square$

**Lemma B.5.** *For any* $n > 1$, $\sum_{k=1}^{n-1} \frac{1}{\Delta_k^2} \leq \frac{1}{\Delta_n^2}$.

*Proof.* From the setting of Algorithm 1, we know that $\Delta_{k+1} = \frac{\Delta_k}{\sqrt{2}}$. Therefore,

$$\sum_{k=1}^{n-1} \frac{1}{\Delta_k^2} = \sum_{i=1}^{n-1} \frac{1}{2^i} \cdot \frac{1}{\Delta_n^2} \leq \frac{1}{\Delta_n^2}.$$

$\square$

## C. Batched Explore-then-Commit Algorithm

In this section, we propose the Batched Explore-then-Commit (Batched-ETC) algorithm. Following a similar regret-balancing idea as BARB, Algorithm 4 operates in batches. Starting with an initial exploration length $T_1$ that is the algorithm's input, in each batch $k$ we keep a counter $N_k$ counting the number of overlapping confidence intervals during exploitation. When $N_k$ exceeds a certain threshold, it triggers a fresh batch $k+1$ where the exploration length $T_{k+1} = 2T_k$.

---

**Algorithm 4** Batched Explore-then-Commit (Batched-ETC)

---

**Input:** Time horizon $T$, preference profile $(P_a)_{a \in \mathcal{K}}$, context dimension $d$, initial exploration length $T_1$, ridge penalty parameter $\lambda$.

1: $t \leftarrow 1$.
2: **for** $k = 1, 2, \cdots$ **do**
3:      $\boldsymbol{V}_i^{(0)} \leftarrow \lambda \boldsymbol{I}_{d \times d}, S_i^{(0)} \leftarrow \boldsymbol{0}_d, \hat{\theta}_i^{(0)} \leftarrow \boldsymbol{0}_d, \forall i \in [N]$.
4:      $N_k \leftarrow 0$. {Number of rounds with overlapping CIs during exploitation.}
5:      **for** $t_k = 1, 2, \cdots, T_k$ **do** {Learning Step}
6:          Observe context $\mathrm{x}_j(t_k)$ for every arm $a_j, \forall j \in [K]$.
7:          Choose a matching $\mu_{t_k}$, match player $p_i$ to arm $a_{i_{t_k}}$ where $i_{t_k} = \mu_{t_k}(i)$.
8:          For every player $p_i$, collect the reward $y_i(t_k)$.
9:          $\boldsymbol{V}_i^{(t_k)} \leftarrow \boldsymbol{V}_i^{(t_k-1)} + \mathrm{x}_{i_{t_k}} \mathrm{x}_{i_{t_k}}^\top, S_i^{(t_k)} \leftarrow S_i^{(t_k-1)} + \mathrm{x}_{i_{t_k}} y_i(t_k)$
10:      **end for**
11:      $\hat{\theta}_i^{(T_k)} \leftarrow \left(\boldsymbol{V}_i^{(T_k)}\right)^{-1} S_i^{(T_k)}$.
12:      **for** $t_k = T_k + 1, \cdots$ **do** {Exploitation Step}
13:          $\hat{U}_{i,j}(t_k) = (\hat{\theta}_i^{(t_k)})^\top \mathrm{x}_j(t_k), F_i \leftarrow$ False.
14:          $\mu_{t_k}(i) \leftarrow \mu^{GS}(i), \forall i \in [N]$. (Input $\hat{U}(t_k)$ and $(P_a)_{a \in \mathcal{K}}$ for **Gale-Shapley**)
15:          $\Delta_k \leftarrow \sqrt{\frac{\log T}{T_k}}$
16:          **for** $i = 1, 2, \cdots, N$ **do**
17:              $\hat{U}_{i,\cdot}^{\text{sort}}(t_k) \leftarrow \text{Sort}\left(\hat{U}_{i,\cdot}(t_k), \text{decreasing}\right)$
18:              $\Delta_{i,\min} \leftarrow \min\left\{\hat{U}_{i,j}^{\text{sort}}(t_k) - \hat{U}_{i,j+1}^{\text{sort}}(t_k)\right\}, j \in [N]$
19:              **if** $\Delta_{i,\min} > 2\Delta_k$ **then**
20:                  $F_i \leftarrow$ True
21:              **end if**
22:          **end for**
23:          $t \leftarrow t + 1$.
24:          **if** $t > T$ **then**
25:              Stop and return.
26:          **end if**
27:          **if** $\exists i \in [N]$, s.t. $F_i ==$ False **then** {CIs are overlapped}
28:              $N_k \leftarrow N_k + 1$.
29:              **if** $N_k > \frac{3 \log T}{16 \Delta_k^2}$ **then** {Exploitation regret exceeds exploration regret}
30:                  Break the loop and enter the next batch, set $T_{k+1} \leftarrow 2T_k$,
31:              **end if**
32:          **end if**
33:      **end for**
34: **end for**

---

## C.1. Theoretical Analysis of Batched-ETC (Algorithm 4)

**Theorem C.1** (Regret Upper Bound for Batched-ETC with stochastic contexts). *Assume that the environment follows Assumption 1, 2 and 3. Further, assume that the context distribution follows Assumption 4, following the Batched-ETC algorithm, the player-optimal stable regret for $p_i \in \mathcal{N}$ is*

$$Reg_i(T) = \mathcal{O}\left(\frac{\log T}{\bar{\lambda}^2 \Delta_{\min}^2} + \frac{\log T}{\Delta_{\min}^2}\right). \tag{28}$$

*Proof.* Denote $k^*$ as the largest batch number of Algorithm 4. That is to say, the counter $N_k$ would not meet the pre-defined threshold in batch $k^*$. In batch $k$, given the exploration length $T_k$, define $\Delta_k = C\frac{\sqrt{\log T}}{\bar{\lambda}\sqrt{T_k}}$ as the width for the confidence interval constructed for the estimated utility in the exploitation phase, where $C$ is a constant specified as in Lemma C.1. Define $\mathcal{F}_k = \left\{\exists t_k > T_k, i \in [N], j \in [K] : |\hat{\boldsymbol{U}}_{i,j}(t) - \boldsymbol{U}_{i,j}(t)| > \Delta_k\right\}$ as the bad event that some preference of players over arms is not estimated well in the exploitation phase of batch $k$. When $k > k^*$, we define $\mathbb{P}(\mathcal{F}_k) = 0$. Let $\mathcal{F}_d^{(t)} := \left\{\hat{\boldsymbol{U}}_{i,j}^{\text{sort}}(t) - \hat{\boldsymbol{U}}_{i,j+1}(t) > 2\Delta_k, \forall j \in [N], \forall i \in [N]\right\} \cap \left\{\hat{\boldsymbol{U}}_{i,N}^{\text{sort}}(t) - \hat{\boldsymbol{U}}_{i,j}^{\text{sort}}(t) > 2\Delta_k, \forall j \in [N+1, K], \forall i \in [N]\right\}$ as the event that the first $(N+1)$-ranked arms have disjoint confidence intervals for any player $p_i$ at time $t$. Denote $T_k'$ as the total length of the $k$-th batch. Define $\mathcal{F}_{\Delta_{k^*}} := \left\{\Delta_{k^*} \geq \frac{\Delta_{\min}}{4\sqrt{2}}\right\}$ as the event that the algorithm stops at an early batch.

The optimal stable regret for each player $p_i$ by following Batched-ETC (Algorithm 4) satisfies

$$Reg_i(T) = \mathbb{E}\left[\sum_{t=1}^{T}\left(\boldsymbol{U}_i^*(t) - \boldsymbol{U}_{i,\mu_t(i)}(t)\right)\right] \tag{29}$$

$$= \mathbb{E}\left[\sum_{t=1}^{T}\left(\boldsymbol{U}_i^*(t) - \boldsymbol{U}_{i,\mu_t(i)}(t)\right)\mathbb{1}\{\forall k \in [T]: \neg\mathcal{F}_k\}\right] + \mathbb{E}\left[\sum_{t=1}^{T}\left(\boldsymbol{U}_i^*(t) - \boldsymbol{U}_{i,\mu_t(i)}(t)\right)\mathbb{1}\{\exists k \in [T]: \mathcal{F}_k\}\right]$$

$$\leq \mathbb{E}\left[\sum_{t=1}^{T}\left(\boldsymbol{U}_i^*(t) - \boldsymbol{U}_{i,\mu_t(i)}(t)\right)\mathbb{1}\{\forall k \in [T]: \neg\mathcal{F}_k\}\right] + B_y T\mathbb{P}\left(\exists k \in [T]: \mathcal{F}_k\right) \tag{30}$$

$$\leq \mathbb{E}\left[\sum_{t=1}^{T}\left(\boldsymbol{U}_i^*(t) - \boldsymbol{U}_{i,\mu_t(i)}(t)\right)\mathbb{1}\{\forall k \in [T]: \neg\mathcal{F}_k\}\right] + B_y T \cdot \frac{NK}{T} \tag{31}$$

$$= \mathbb{E}\left[\sum_{t=1}^{T}\left(\boldsymbol{U}_i^*(t) - \boldsymbol{U}_{i,\mu_t(i)}(t)\right)\mathbb{1}\{\{\forall k \in [k^*]: \neg\mathcal{F}_k\}\cap\mathcal{F}_{\Delta_{k^*}}\}\right]$$

$$+ \mathbb{E}\left[\sum_{t=1}^{T}\left(\boldsymbol{U}_i^*(t) - \boldsymbol{U}_{i,\mu_t(i)}(t)\right)\mathbb{1}\{\{\forall k \in [k^*]: \neg\mathcal{F}_k\}\cap\neg\mathcal{F}_{\Delta_{k^*}}\}\right] + B_y NK$$

$$\leq \mathbb{E}\left[\sum_{t=1}^{T}\left(\boldsymbol{U}_i^*(t) - \boldsymbol{U}_{i,\mu_t(i)}(t)\right)\mathbb{1}\{\{\forall k \in [k^*]: \neg\mathcal{F}_k\}\cap\mathcal{F}_{\Delta_{k^*}}\}\right] + B_y T\mathbb{P}\left(\neg\mathcal{F}_{\Delta_{k^*}}\right) + B_y NK \tag{32}$$

$$\leq \mathbb{E}\left[\sum_{k=1}^{k^*}\left[B_y T_k + \sum_{t_k=T_k+1}^{T_k'}\left(\boldsymbol{U}_i^*(t_k) - \boldsymbol{U}_{i,\mu_t(i)}(t_k)\right)\right]\mathbb{1}\{\{\forall k \in [k^*]: \neg\mathcal{F}_k\}\cap\mathcal{F}_{\Delta_{k^*}}\}\right] + B_y T \cdot \frac{1}{T} + B_y NK \tag{33}$$

$$\leq \mathbb{E}\left[\sum_{k=1}^{k^*}\left[B_y T_k\mathbb{1}\{\{\forall k \in [k^*]: \neg\mathcal{F}_k\}\cap\mathcal{F}_{\Delta_{k^*}}\}\right.\right.$$

$$+ \sum_{t_k=T_k+1}^{T_k'}\left[\left(\boldsymbol{U}_i^*(t_k) - \boldsymbol{U}_{i,\mu_t(i)}(t_k)\right)\mathbb{1}\left\{\{\forall k \in [k^*]: \neg\mathcal{F}_k\}\cap\mathcal{F}_{\Delta_{k^*}}\cap F_d^{(t_k)}\right\}\right.$$

$$\left.\left.+ \left(\boldsymbol{U}_i^*(t_k) - \boldsymbol{U}_{i,\mu_t(i)}(t_k)\right)\mathbb{1}\left\{\{\forall k \in [k^*]: \neg\mathcal{F}_k\}\cap\mathcal{F}_{\Delta_{k^*}}\cap\neg F_d^{(t_k)}\right\}\right]\right]\right] + B_y(1 + NK)$$

$$\leq \mathbb{E}\left[\sum_{k=1}^{k^*}\left[B_y T_k\mathbb{1}\{\{\forall k \in [k^*]: \neg\mathcal{F}_k\}\cap\mathcal{F}_{\Delta_{k^*}}\}\right.\right.$$

$$\left.\left.+ \sum_{t_k=T_k+1}^{T_k'}\left(\boldsymbol{U}_i^*(t_k) - \boldsymbol{U}_{i,\mu_t(i)}(t_k)\right)\mathbb{1}\left\{\{\forall k \in [k^*]: \neg\mathcal{F}_k\}\cap\mathcal{F}_{\Delta_{k^*}}\cap\neg F_d^{(t_k)}\right\}\right]\right] + B_y(1 + NK) \tag{34}$$

$$\leq B_y\mathbb{E}\left[\sum_{k=1}^{k^*}[T_k + N_k]\mathbb{1}\{\{\forall k \in [k^*]: \neg\mathcal{F}_k\}\cap\mathcal{F}_{\Delta_{k^*}}\}\right] + B_y(1 + NK) \tag{35}$$

$$\leq B_y\mathbb{E}\left[\sum_{k=1}^{k^*}\left(\frac{C^2\log T}{\tilde{\lambda}^2\Delta_k^2} + \frac{3\log T}{16\Delta_k^2} + 3NK\right)\mathbb{1}\{\{\forall k \in [k^*]: \neg\mathcal{F}_k\}\cap\mathcal{F}_{\Delta_{k^*}}\}\right] + B_y(1 + NK) \tag{36}$$

$$\leq 2B_y\mathbb{E}\left[\left(\frac{C^2\log T}{\tilde{\lambda}^2\Delta_{k^*}^2} + \frac{3\log T}{16\Delta_{k^*}^2}\right)\mathbb{1}\{\{\forall k \in [k^*]: \neg\mathcal{F}_k\}\cap\mathcal{F}_{\Delta_{k^*}}\}\right] + 3NK \cdot 2\log_2\left(\frac{4\sqrt{2}\Delta_1}{\Delta_{\min}}\right) + B_y(1 + NK) \tag{37}$$

$$\leq 64B_y\left(\frac{C^2\log T}{\tilde{\lambda}^2\Delta_{\min}^2} + \frac{3\log T}{16\Delta_{\min}^2}\right) + 6NK\log_2\left(\frac{4\sqrt{2}\Delta_1}{\Delta_{\min}}\right) + B_y(1 + NK) = \mathcal{O}\left(\frac{\log T}{\tilde{\lambda}^2\Delta_{\min}^2} + \frac{\log T}{\Delta_{\min}^2}\right).$$

Eq.(29) holds according to the assumption that the noises are i.i.d. with mean 0. Eq.(30) and (32) come from the fact that the expected per-round regret is $\mathbb{E}\left[\boldsymbol{U}_i^*(t) - y_i(t)\right] = \boldsymbol{U}_i^*(t) - \boldsymbol{U}_{i,\mu_t(i)}(t)$ and both terms lie in $[-B_\theta B_x, B_\theta B_x]$ under Assumptions 1 and 3, and hence the expected regret per round is bounded by $2B_\theta B_x = B_y$. Eq.(31) relies on Lemma C.1. Eq.(33) follows from the same reasoning as we used in Lemma B.2. Eq.(34) holds because, with Lemma B.3, under the good event $\neg \mathcal{F}_k$ (ensuring all utilities are well-estimated) and the event $\mathcal{F}_d^{(t_k)}$ (ensuring the top $(N+1)$-ranked confidence intervals are disjoint at time $t_k$), the Gale-Shapley subroutine invoked by Algorithm 4 is guaranteed to output the OSS for that round. Eq.(35) follows from the algorithm setting that $N_k$ counts the number of times during exploitation that the top $(N+1)$-ranked CIs are overlapped. Eq.(36) holds according to the definition of $\Delta_k$ and that the algorithm would move to the next batch whenever $N_k > \frac{3\log T}{16\Delta_k^2}$. Eq.(37) follows from the definition of $\Delta_k$ and that $T_{k+1} = 2T_k$, we have $\Delta_{k+1} = \frac{\Delta_k}{\sqrt{2}}$, and hence the regret from the $k$-th batch would dominate the cumulative regret from the first $(k-1)$-th batches as shown by Lemma B.5.

$\square$

**Lemma C.1.** *Under Assumptions 1, 3 and 4, we have that for any given time $t_k$ in batch $k$, for any fixed player $p_i$, arm $a_j$,*

$$\mathbb{P}\left(|\hat{\boldsymbol{U}}_{i,j}(t) - \boldsymbol{U}_{i,j}(t)| > \frac{B_x}{\tilde{\lambda} + \frac{T_k\lambda}{2}}\left(B_x B_\epsilon \sqrt{2T_k \log(4T^3)} + \lambda B_\theta\right)\right) \leq \frac{1}{T^3}.$$

*Furthermore, for some constant $C$ such that $C\frac{\sqrt{\log T}}{\tilde{\lambda}\sqrt{T_k}} \geq \frac{B_x}{\tilde{\lambda} + \frac{T_k\lambda}{2}}\left(B_x B_\epsilon \sqrt{2T_k \log(4T^3)} + \lambda B_\theta\right)$, we have*

$$\mathbb{P}\left(|\hat{\boldsymbol{U}}_{i,j}(t) - \boldsymbol{U}_{i,j}(t)| > C\frac{\sqrt{\log T}}{\tilde{\lambda}\sqrt{T_k}}\right) \leq \frac{1}{T^3}.$$

*And hence,*

$$\mathbb{P}\left(\exists k \in [T] : \mathcal{F}_k\right) \leq \frac{NK}{T}.$$

*Proof.* Let $X = \left(X_1^\top, \cdots, X_K^\top\right)^\top$ be the contexts for all $K$ arms, let $\Sigma = \mathbb{E}\left[XX^\top \mid X \in \mathcal{D}_x\right]$, $\Sigma_i = \mathbb{E}\left[X_i X_i^\top \mid X_i \in \mathcal{D}_{x_i}\right]$. Under Assumption 4, $\lambda_{\min}(\Sigma) \geq \tilde{\lambda}$. By Cauchy's eigenvalue interlacing theorem (Lemma J.4), we have that $\lambda_{\min}(\Sigma_i) \geq \lambda_{\min}(\Sigma) \geq \tilde{\lambda} > 0$. Therefore, under Assumption 4, it suffices for any player $p_i$ to sample a fixed arm to learn the preference parameter $\theta_i$. Without loss of generality, we assume that the algorithm matches $p_i$ to $a_i$ during the exploration phase.

In the $k$-th batch, let $X_i^{(k)} \in \mathbb{R}^{T_k \times d}$ be the contexts of arm $a_i$ during the exploration phase, without ambiguity, we use $X_i$ to replace $X_i^{(k)}$. For player $p_i$, in the exploitation phase, we have

$$
\begin{aligned}
\|\hat{\theta}_i^{(k)} - \theta_i\|_2 &= \|\left(X_i^\top X_i + \lambda \boldsymbol{I}_d\right)^{-1} X_i^\top \left(X_i \theta_i + \epsilon\right) - \theta_i\|_2 \\
&= \|\left(X_i^\top X_i + \lambda \boldsymbol{I}_d\right)^{-1}\left(X_i^\top X_i + \lambda \boldsymbol{I}_d - \lambda \boldsymbol{I}_d\right)\theta_i + \left(X_i^\top X_i + \lambda \boldsymbol{I}_d\right)^{-1} X_i^\top \epsilon - \theta\|_2 \\
&= \|\theta_i - \lambda \left(X_i^\top X_i + \lambda \boldsymbol{I}_d\right)^{-1}\theta_i + \left(X_i^\top X_i + \lambda \boldsymbol{I}_d\right)^{-1} X_i^\top \epsilon - \theta\|_2 \\
&= \|\left(X_i^\top X_i + \lambda \boldsymbol{I}_d\right)^{-1}\left(X_i^\top \epsilon - \lambda \theta_i\right)\|_2 \\
&\leq \frac{1}{\lambda_{\min}\left(X_i^\top X_i + \lambda \boldsymbol{I}_d\right)} \|X_i^\top \epsilon - \lambda \theta_i\|_2 \\
&\leq \frac{1}{\lambda_{\min}\left(X_i^\top X_i + \lambda \boldsymbol{I}_d\right)}\left(\|X_i^\top \epsilon\| + \lambda B_\theta\right).
\end{aligned}
$$

By matrix Bernstein inequality (Lemma J.5), we have that with probability at least $1 - o(1/T^3)$, $\lambda_{\min}\left(X_i^\top X_i\right) > \frac{T_k}{2}\lambda_{\min}(\Sigma_i) \geq \frac{T_k\tilde{\lambda}}{2}$, and hence with probability at least $1 - o(1/T^3)$, we have

$$\frac{1}{\lambda_{\min}\left(X_i^\top X_i + \lambda \boldsymbol{I}_d\right)} \leq \frac{1}{\lambda + \frac{T_k\tilde{\lambda}}{2}}. \tag{38}$$

On the other hand, $X_i^\top \epsilon = \sum_{t=1}^{T_k} \epsilon_t X_{i,t}$ is the sum of independent random vectors. By Hoeffding's inequality, for any $s > 0$

$$\mathbb{P}\left(\|\sum_{t=1}^{T_k} \epsilon_t X_{i,t}\|_2 \geq s\right) \leq 2\exp\left(-\frac{s^2}{2B_\epsilon^2 \sum_{t=1}^{T_k}\|X_{i,t}\|_2^2}\right),$$

where $\sum_{t=1}^{T_k}\|X_{i,t}\|_2^2 \leq T_k B_x^2$. Let the RHS of the above inequality be $\frac{1}{2T^3}$, then we have

$$\mathbb{P}\left(\|X_i^\top \epsilon\| \geq B_x B_\epsilon \sqrt{2T_k \log(4T^3)}\right) \leq \frac{1}{2T^3}. \tag{39}$$

Combining Eq.(38) and Eq.(39) with union bound, we have that

$$\mathbb{P}\left(\|\hat{\theta}_i^{(k)} - \theta_i\|_2 > \frac{1}{\lambda + \frac{T_k \tilde{\lambda}}{2}}\left(B_x B_\epsilon \sqrt{2T_k \log(4T^3)} + \lambda B_\theta\right)\right) \leq \frac{1}{T^3}. \tag{40}$$

Finally, as $|\hat{U}_{i,j}(t) - U_{i,j}(t)| = |\left(\hat{\theta}_i - \theta_i\right)^\top x_j(t)| \leq \|\hat{\theta}_i - \theta_i\|_2 \|x_j(t)\|_2 \leq B_x \|\hat{\theta}_i - \theta_i\|_2$, we reach the first part of the conclusion. While for the last part, we have

$$\mathbb{P}\left(\exists k \in [T] : \mathcal{F}_k\right) = \mathbb{P}\left(\exists k \in [T], \exists t_k \in [T_k + 1, T_k'], i \in [N], j \in [K] : |\hat{U}_{i,j}(t_k) - U_{i,j}(t_k)| > \Delta_k\right)$$

$$\leq \sum_{k=1}^{T} \sum_{t_k=T_k+1}^{T_k'} \sum_{i=1}^{N} \sum_{j=1}^{K} \mathbb{P}\left(|\hat{U}_{i,j}(t_k) - U_{i,j}(t_k)| > \Delta_k\right)$$

$$\leq T^2 NK \frac{1}{T^3} = \frac{NK}{T}.$$

$\square$

# D. Proof of Theorem 3.2

*Proof.* When the context distribution satisfies Assumption 5, there exists some threshold $\Delta_0$ and some constant $c$ such that the CDF of $\delta_{\min}$: $F(\Delta)$ satisfies $F(\Delta) \leq c\Delta$ for any $\Delta \leq \Delta_0$. Therefore,

$$\mathbb{P}\left(\delta_{\min} \geq \Delta\right) = 1 - F(\Delta) \geq 1 - c\Delta, \forall \Delta \leq \Delta_0$$

For the constant $c$ and the threshold $\Delta_0$ given above, there exists a $T_0 \geq e$ such that $1 - c\Delta_0 = 1 - \frac{\log T_0}{T_0 \Delta_0^2}$, and hence for $T \geq T_0$ sufficiently large, we have

$$1 - c\Delta_0 \leq 1 - \frac{\log T}{T\Delta_0^2}.$$

When $T \geq T_0$ is fixed, the LHS of the above inequality is a decreasing function of $\Delta_0$ while the RHS is an increasing function of $\Delta_0$. Furthermore, when $\Delta_0 \to 0$, $1 - c\Delta_0 \to 1$ while $1 - \frac{\log T}{T\Delta_0^2} \to -\infty$. Therefore, there exists a $\bar{\Delta} \leq \Delta_0$ such that

$$1 - F\left(\bar{\Delta}\right) \geq 1 - c\bar{\Delta} = 1 - \frac{\log T}{T\bar{\Delta}^2}. \tag{41}$$

Solve Eq.(41), we have $\bar{\Delta} = \left(\frac{\log T}{cT}\right)^{1/3} = \tilde{\Theta}\left(T^{-1/3}\right)$. And

$$\mathbb{P}\left(\delta_{\min} \geq \Delta\right) = 1 - F(\Delta) \geq 1 - \frac{\log T}{T\Delta^2}, \quad \forall \Delta \leq \bar{\Delta}.$$

From the minimum preference gap definition (Definition 2), we know that $\Delta_{\min}$ is the maximum $\Delta$ such that $\mathbb{P}\left(\delta_{\min} \geq \Delta\right) \geq 1 - \frac{\log T}{T\Delta^2}$. Therefore,

$$\Delta_{\min} \geq \bar{\Delta} \implies \Delta_{\min} = \tilde{\Omega}\left(T^{-1/3}\right).$$

Finally, as the regret upper bound for Algorithm 1 is $\mathcal{O}\left(\frac{\log^2 T}{\Delta_{\min}^2}\right)$ as shown in Theorem 3.1, we have the conclusion that

$$Reg_i(T) = \tilde{\mathcal{O}}\left(T^{2/3}\right), \quad \forall i \in [N] \quad \text{as} \quad T \to \infty.$$

$\square$

# E. Proof of Theorem 3.3

*Proof.* We prove Theorem 3.3 by contradiction. Fix a policy $\pi$, assume that this policy achieves $o\left(T^{2/3}\right)$ regret for each player in the market.

Let $\mathcal{N} = \{p_1, p_2, p_3\}, \mathcal{K} = \{a_1, a_2, a_3\}$, and $p_1 \succ p_2 \succ p_3$ for all arms. Throughout the proof, we assume that given the contexts and the preference parameters, all observations are Gaussian of unit variance, that is, when matching $p_i$ to $a_j$ at round $t$, we observe $y_i(t) \sim \mathcal{N}\left(\theta_i^\top x_j(t), 1\right)$, where $\theta_i$ is the preference parameter for player $p_i$ and $x_j(t)$ is the observed context for arm $a_j$ at time $t$. Let $\tau = T^{-1/3}, \varphi = \frac{1}{2}, \psi = \frac{1}{8}$. Consider two instances $\nu$ and $\nu'$ with the following preference parameters and context distributions, where the distribution of $x_2$ depends on the distribution of $x_1$, and the two instances only differ in the first entry of $\theta_1$:

$$
\theta_1 = \begin{bmatrix} \beta \\ 1 \\ 0 \\ 0 \end{bmatrix}, \theta_2 = \begin{bmatrix} 0 \\ 0 \\ 1 \\ 0 \end{bmatrix}, \theta_3 = \begin{bmatrix} 0 \\ 0 \\ 0 \\ 1 \end{bmatrix}
$$

$$
x_1 = \begin{bmatrix} U \\ 0 \\ 1 \\ \psi \end{bmatrix}, x_2 = \begin{bmatrix} 0 \\ 1 \\ 0 \\ f(U) \end{bmatrix}, x_3 = \begin{bmatrix} 0 \\ 0 \\ \psi \\ 0 \end{bmatrix},
$$

where $U$ is a uniform distribution on $[0,1]$, $f(U) = 1 \cdot \mathbb{1}\left\{U > \frac{1}{1+\tau}\right\} + \varphi \cdot \mathbb{1}\left\{U \le \frac{1}{1+\tau}\right\}$; $\beta = 1$ in instance $\nu$ and $\beta = 1 + \tau$ in instance $\nu'$. Therefore, we could write the utility matrix $U$ for $\nu$ and $U'$ for $\nu'$ as follows:

$$
U = \begin{bmatrix} U & 1 & 0 \\ 1 & 0 & \psi \\ \psi & f(U) & 0 \end{bmatrix}, \quad U' = \begin{bmatrix} (1+\tau) \cdot U & 1 & 0 \\ 1 & 0 & \psi \\ \psi & f(U) & 0 \end{bmatrix}.
$$

In Lemma E.2, we show that the above two instances are valid for proving a matching regret lower bound corresponding to the regret upper bound in Theorem 3.2, in the sense that the two instances satisfy Assumption 5.

Furthermore, since there are 3 players and 3 arms in the market, and the payoffs are non-negative for every player at every time, we can, without loss of generality, assume that every player gets matched to one arm every time. Denote the instantaneous regret for player $p_i$ as $U_i^*(t) - U_{p_i, \mu_t(p_i)}(t)$.

**Lemma E.1** (Properties of Instances $\nu$ and $\nu'$). *Based on the utility matrices $U$ and $U'$, we have the following properties of $\nu$ and $\nu'$:*

## 1. Benchmark Utilities

- *The benchmark utilities for the three players in instance $\nu$ are $U^* = (1, 1, 0)^\top$.*

- *The benchmark utilities for the three players in instance $\nu'$ are $U^* = (1, 1, 0)^\top$ when $U \le \frac{1}{1+\tau}$, and $U^* = ((1+\tau)U, \psi, 1)^\top$ when $U > \frac{1}{1+\tau}$.*

## 2. Regret under Consideration

- *In instance $\nu$, the regret for $p_2$, i.e., $Reg_{\nu, \pi}^{p_2}(T)$, under any context realizations would be non-negative.*

- *In instance $\nu'$, the social regret for $p_1 + p_2 + p_3$, i.e., $Reg_{\nu', \pi}^{p_1 + p_2 + p_3}(T)$, under any context realizations would be non-negative.*

The proof of Lemma E.1 is provided at the end of this section.

Let $\mathbb{P}_{\nu, \pi}$ be the joint probability measure over the history, and $\mathbb{E}_{\nu, \pi}$ be the expectation induced by instance $\nu$ and policy $\pi$, and $\mathbb{P}_{\nu', \pi}, \mathbb{E}_{\nu', \pi}$ be defined similarly. The two instances $\nu$ and $\nu'$ only differ in one entry of the utility matrix, i.e., the entry

corresponds to the pair $(p_1, a_1)$. From the divergence decomposition theorem (Lemma J.6), we know that

$$D\left(\mathbb{P}_{\boldsymbol{\nu},\pi}, \mathbb{P}_{\boldsymbol{\nu}',\pi}\right) = \sum_{i=1}^{N} \sum_{j=1}^{K} \mathbb{E}_{\boldsymbol{\nu},\pi} N_{ij}(T) \cdot D(\boldsymbol{\nu}_{ij}, \boldsymbol{\nu}_{ij}')$$

$$= \mathbb{E}_{\boldsymbol{\nu},\pi}\left[N_{11}(T)\right] \cdot D(\boldsymbol{\nu}_{11}, \boldsymbol{\nu}_{11}'),$$

where $\boldsymbol{\nu}_{ij}$ is the distribution of utilities obtained when player $p_i$ is matched to arm $a_j$ in the environment $\boldsymbol{\nu}$, $N_{ij}(T) \in \mathbb{N} \cup 0$ is the number of times player $p_i$ is matched to arm $a_j$, up to and including time $T$. By the chain rule of KL divergence (Lemma J.7), we have

$$D(\boldsymbol{\nu}_{11}, \boldsymbol{\nu}_{11}') = D(\mathbb{P}_{\boldsymbol{\nu},\pi}(U), \mathbb{P}_{\boldsymbol{\nu}',\pi}(U)) + D\left(\mathbb{P}_{\boldsymbol{\nu},\pi}(\boldsymbol{U}_{11}|U), \mathbb{P}_{\boldsymbol{\nu},\pi}(\boldsymbol{U}_{11}'|U)\right)$$

$$= 0 + \mathbb{E}_{u \sim U[0,1]} \frac{(\tau u)^2}{2}$$

$$\leq \frac{\tau^2}{2}.$$

Therefore, $D\left(\mathbb{P}_{\boldsymbol{\nu},\pi}, \mathbb{P}_{\boldsymbol{\nu}',\pi}\right) \leq \frac{\tau^2}{2} \cdot \mathbb{E}_{\boldsymbol{\nu},\pi}[N_{11}(T)]$. Now consider the event $A = \left\{N_{11}(T) \geq \frac{\tau T}{4}\right\}$ and its complement $A^c = \left\{N_{11}(T) < \frac{\tau T}{4}\right\}$, by Bretagnolle-Huber inequality (Lemma J.8), we have

$$\mathbb{P}_{\boldsymbol{\nu},\pi}(A) + \mathbb{P}_{\boldsymbol{\nu}',\pi}(A^c) \geq \frac{1}{2} \exp\left(-D\left(\mathbb{P}_{\boldsymbol{\nu},\pi}, \mathbb{P}_{\boldsymbol{\nu}',\pi}\right)\right) \geq \frac{1}{2} \exp\left(-\frac{\tau^2}{2} \cdot \mathbb{E}_{\boldsymbol{\nu},\pi}[N_{11}(T)]\right).$$

In instance $\boldsymbol{\nu}$, the stable matching benchmark for player $p_2$ is arm $a_1$ and it is the arm providing maximum utility for $p_2$, therefore, every time when we match $(p_1, a_1)$, $p_2$ could only choose between $a_2$ and $a_3$, which would incur positive regret and the regret could not be compensate by the rounds that $p_2$ is matched to $a_1$, this leads to $(1 - \psi) \cdot \mathbb{E}_{\boldsymbol{\nu},\pi}[N_{11}(T)] \leq Reg_{\boldsymbol{\nu},\pi}^{p_2}(T)$. From the assumption, we know that $\tau = T^{-1/3}$ and $Reg_{\boldsymbol{\nu},\pi}^{p_2}(T) = o(T^{2/3})$, i.e., $\limsup_{T \to \infty} \frac{Reg_{\boldsymbol{\nu},\pi}^{p_2}(T)}{T^{2/3}} = 0$ and hence there exist a constant $c_1$ such that $\mathbb{P}_{\boldsymbol{\nu},\pi}(A) + \mathbb{P}_{\boldsymbol{\nu}',\pi}(A^c) \geq c_1$, which in turns implies either $\mathbb{P}_{\boldsymbol{\nu},\pi}(A) \geq \frac{c_1}{2} := c_2$ or $\mathbb{P}_{\boldsymbol{\nu}',\pi}(A^c) \geq c_2$.

**Case I:** $\mathbb{P}_{\boldsymbol{\nu},\pi}(A) \geq c_2$.

In instance $\boldsymbol{\nu}$, when $p_1$ is matched to $a_1$, the best possible case for $p_2$ is that $p_2 - a_3$ such that $p_2$ collects $\psi$ expected reward in the corresponding round, and hence the regret in this round is $1 - \psi$. While $a_1$ is the arm with the maximum utility for $p_2$, this regret cannot be compensated in the rounds that $p_2$ is matched to $a_1$. Hence, we have

$$Reg_{\boldsymbol{\nu},\pi}^{p_2}(T) \geq (1 - \psi) \cdot \frac{\tau T}{4} \cdot \mathbb{P}_{\boldsymbol{\nu},\pi}(A).$$

If $\mathbb{P}_{\boldsymbol{\nu},\pi}(A) \geq c_2$, we have $Reg_{\boldsymbol{\nu},\pi}^{p_2}(T) \geq c_3 \tau T = \Omega\left(T^{2/3}\right)$ for some constant $c_3$, which is a contradiction.

**Case II:** $\mathbb{P}_{\boldsymbol{\nu}',\pi}(A^c) \geq c_2$.

In instance $\boldsymbol{\nu}'$, let $N_u(T)$ be the number of times such that $U > \frac{1}{1+\tau}$, as $U$ follows a uniform distribution on $[0, 1]$, we have $N_u(T) \sim Binomial\left(T, \frac{\tau}{1+\tau}\right)$. Define the event $B = \left\{N_u(T) \geq \frac{\tau T}{2}\right\}$ and its complement $B^c = \left\{N_u(T) < \frac{\tau T}{2}\right\}$.

$$\mathbb{P}_{\boldsymbol{\nu},\pi}(A^c) = \mathbb{P}_{\boldsymbol{\nu}',\pi}(A^c \cap B) + \mathbb{P}_{\boldsymbol{\nu}',\pi}(A^c \cap B^c)$$

$$\leq \mathbb{P}_{\boldsymbol{\nu}',\pi}(A^c \cap B) + \mathbb{P}_{\boldsymbol{\nu}',\pi}\left(N_u < \frac{\tau T}{2}\right)$$

$$\leq \mathbb{P}_{\boldsymbol{\nu}',\pi}(A^c \cap B) + \exp\left(-\frac{T\tau(1-\tau)^2}{8(1+\tau)}\right),$$

where the last inequality follows from Lemma E.3. When $\tau = T^{-1/3}$, we have that the second term of the last inequality being $o\left(\exp\left(-T^{1/3}\right)\right) \leq c_4$ for some constant $c_4 < c_2$ when $T$ is sufficiently large, leading to the requirement that $\mathbb{P}_{\boldsymbol{\nu}',\pi}\left(A^c \cap B\right) \geq c_2 - c_4 := c_5 > 0$.

Define $N_{ij}^+(T)$ and $N_{ij}^-(T)$ as the number of times such that player $p_i$ is matched to arm $a_j$ in the rounds that $U > \frac{1}{1+\tau}$ and $U \leq \frac{1}{1+\tau}$, respectively. Therefore, we have $N_{ij}(T) = N_{ij}^+(T) + N_{ij}^-(T)$, and $\left\{ N_{11}(T) < \frac{\tau T}{4} \right\} \subseteq \left\{ N_{11}^+(T) < \frac{\tau T}{4} \right\}$, which further implies

$$\mathbb{P}_{\boldsymbol{\nu'},\pi} \left( N_{11}^+(T) < \frac{\tau T}{4}, N_u(T) \geq \frac{\tau T}{2} \right) \geq \mathbb{P}_{\boldsymbol{\nu'},\pi}(A^c \cap B) \geq c_5.$$

We then consider the event $\left\{ N_{11}^+(T) < \frac{\tau T}{4}, N_u(T) \geq \frac{\tau T}{2} \right\}$. We further decompose the event as

$$\left\{ N_{11}^+(T) < \frac{\tau T}{4}, N_u(T) \geq \frac{\tau T}{2} \right\}$$
$$= \left\{ N_{11}^+(T) < \frac{\tau T}{4}, N_u(T) \geq \frac{\tau T}{2}, N_{13}^+(T) > \frac{\tau T}{8} \right\} \cup \left\{ N_{11}^+(T) < \frac{\tau T}{4}, N_u(T) \geq \frac{\tau T}{2}, N_{13}^+(T) \leq \frac{\tau T}{8} \right\},$$

then we have

$$\mathbb{P}_{\boldsymbol{\nu'},\pi} \left( N_{11}^+(T) < \frac{\tau T}{4}, N_u(T) \geq \frac{\tau T}{2}, N_{13}^+(T) > \frac{\tau T}{8} \right) + \mathbb{P}_{\boldsymbol{\nu'},\pi} \left( N_{11}^+(T) < \frac{\tau T}{4}, N_u(T) \geq \frac{\tau T}{2}, N_{13}^+(T) \leq \frac{\tau T}{8} \right)$$

$$\geq \mathbb{P}_{\boldsymbol{\nu'},\pi} \left( N_{11}^+(T) < \frac{\tau T}{4}, N_u(T) \geq \frac{\tau T}{2} \right) \geq \mathbb{P}_{\boldsymbol{\nu'},\pi}(A^c \cap B)$$

$$\geq c_5,$$

which implies either $\mathbb{P}_{\boldsymbol{\nu'},\pi} \left( N_{11}^+(T) < \frac{\tau T}{4}, N_u(T) \geq \frac{\tau T}{2}, N_{13}^+(T) > \frac{\tau T}{8} \right) \geq \frac{c_5}{2} := c_6$ or $\mathbb{P}_{\boldsymbol{\nu'},\pi} \left( N_{11}^+(T) < \frac{\tau T}{4}, N_u(T) \geq \frac{\tau T}{2}, N_{13}^+(T) \leq \frac{\tau T}{8} \right) \geq c_6$.

**Case II.a:** $\mathbb{P}_{\boldsymbol{\nu'},\pi} \left( N_{11}^+(T) < \frac{\tau T}{4}, N_u(T) \geq \frac{\tau T}{2}, N_{13}^+(T) > \frac{\tau T}{8} \right) \geq c_6$.

As the benchmark for player $p_1$ is always the one with the largest expected utility, the instantaneous regret for $p_1$ is always non-negative. Furthermore, every time when we match $p_1$ to $a_3$, the instantaneous regret for $p_1$ is no smaller than 1. Combining the above two points, we have

$$Reg_{\boldsymbol{\nu'},\pi}^{p_1} \geq \frac{\tau T}{8} \mathbb{P} \left( N_{13}^+(T) > \frac{\tau T}{8} \right)$$
$$\geq \frac{\tau T}{8} \mathbb{P}_{\boldsymbol{\nu'},\pi} \left( N_{11}^+(T) < \frac{\tau T}{4}, N_u(T) \geq \frac{\tau T}{2}, N_{13}^+(T) > \frac{\tau T}{8} \right)$$
$$\geq \frac{\tau T}{8} \cdot c_6 = \Omega \left( T^{2/3} \right),$$

since $\tau = T^{-1/3}$, which is a contradiction.

**Case II.b:** $\mathbb{P}_{\boldsymbol{\nu'},\pi} \left( N_{11}^+(T) < \frac{\tau T}{4}, N_u(T) \geq \frac{\tau T}{2}, N_{13}^+(T) \leq \frac{\tau T}{8} \right) \geq c_6$.

If $N_{11}^+(T) < \frac{\tau T}{4}$ and $N_{13}^+(T) \leq \frac{\tau T}{8}$, we have $N_{32}^+(T) \leq N_{11}^+(T) + N_{13}^+(T) \leq \frac{3\tau T}{8}$. This is because, without loss of generality, we assume that every player gets matched to one arm every time, and so when arm $a_2$ is occupied by player $p_3$, player $p_1$ is matched to either $a_1$ or $a_3$. Therefore,

$$\mathbb{P}_{\boldsymbol{\nu'},\pi} \left( N_{32}^+(T) \leq \frac{3\tau T}{8}, N_u(T) \geq \frac{\tau T}{2} \right) \geq c_6.$$

When $U > \frac{1}{1+\tau}$, the benchmark utility for $p_3$ is 1, which is the largest utility provided by arm $a_2$, and hence the instantaneous regret for $p_3$ is always non-negative during these rounds. We have

$$Reg_{\boldsymbol{\nu'},\pi}^{p_3}(T) = \mathbb{E} \left[ \sum_{t=1}^T \left[ \boldsymbol{U}_3^*(t) - \boldsymbol{U}_{p_3,\mu_t(p_3)}(t) \right] \right]$$
$$= \mathbb{E} \left[ \sum_{t=1}^T \left( \boldsymbol{U}_3^*(t) - \boldsymbol{U}_{p_3,\mu_t(p_3)}(t) \right) \mathbb{1} \left\{ U_t > \frac{1}{1+\tau} \right\} \right] + \mathbb{E} \left[ \sum_{t=1}^T \left( \boldsymbol{U}_3^*(t) - \boldsymbol{U}_{p_3,\mu_t(p_3)}(t) \right) \mathbb{1} \left\{ U_t \leq \frac{1}{1+\tau} \right\} \right],$$

where

$$
\mathbb{E}\left[\sum_{t=1}^{T}\left(\boldsymbol{U}_3^*(t) - \boldsymbol{U}_{p_3,\mu_t(p_3)}(t)\right)\mathbb{1}\left\{U_t > \frac{1}{1+\tau}\right\}\right]
$$

$$
\geq \mathbb{E}\left[\sum_{t=1}^{T}\left(\boldsymbol{U}_3^*(t) - \boldsymbol{U}_{p_3,\mu_t(p_3)}(t)\right)\mathbb{1}\left\{\left\{U_t > \frac{1}{1+\tau}\right\}\cap\left\{N_{32}^+(T) \leq \frac{3\tau T}{8}\right\}\cap\left\{N_u(T) \geq \frac{\tau T}{2}\right\}\right\}\right],
$$

$$
\geq (1-\psi)\frac{\tau T}{8}\cdot c_6
$$

$$
= \Omega\left(T^{\frac{2}{3}}\right), \tag{42}
$$

where the last inequality comes from the fact that every time when $U_t > \frac{1}{1+\tau}$ while $p_3$ is not matched to $a_2$, the instantaneous regret for $p_3$ is at least $1 - \psi$. And under the event $\left\{N_{32}^+(T) \leq \frac{3\tau T}{8}\right\}\cap\left\{N_u(T) \geq \frac{\tau T}{2}\right\}$, the number of times that $p_3$ is not matched to $a_2$ is at least $\frac{\tau T}{2} - \frac{3\tau T}{8} = \frac{\tau T}{8}$.

To ensure that $Reg_{\boldsymbol{\nu}',\pi}^{p_3}(T) = o\left(T^{2/3}\right)$, we need to compensate $p_3$ during the rounds when $U \leq \frac{1}{1+\tau}$ so that the regret incurred from Eq.(42) does not cause a contradiction to the assumption. While the benchmark utility for $p_3$ is 0, provided by arm $a_3$ in these rounds, the instantaneous regret for $p_3$ during these rounds is always non-positive, and hence there exists some constant $c_7$, say $c_7 := c_6/2$, such that

$$
\sum_{t=1}^{T}\mathbb{E}\left[\left(\boldsymbol{U}_3^*(t) - \boldsymbol{U}_{p_3,\mu_t(p_3)}(t)\right)\mathbb{1}\left\{U_t \leq \frac{1}{1+\tau}\right\}\right] = \mathbb{E}_{\boldsymbol{\nu}',\pi}\left[-\psi\cdot N_{31}^-(T) - \varphi\cdot N_{32}^-(T)\right]
$$

$$
\leq -c_7\frac{(1-\psi)\tau T}{8}.
$$

As $\psi < \varphi$ in our assumption, we have that under the policy $\pi$,

$$
\mathbb{E}_{\boldsymbol{\nu}',\pi}\left[\varphi(N_{31}^-(T) + N_{32}^-(T))\right] \geq c_7\frac{(1-\psi)\tau T}{8} \implies \mathbb{E}_{\boldsymbol{\nu}',\pi}\left[N_{31}^-(T) + N_{32}^-(T)\right] \geq c_7\frac{(1-\psi)\tau T}{8\varphi}.
$$

When $U \leq \frac{1}{1+\tau}$, the benchmark stable matching is $\{p_1 - a_2, p_2 - a_1, p_3 - a_3\}$ with the social benchmark utility of 2. While by matching $p_3$ to $a_1$ or $a_2$, the total collected rewards for the three players would be at most $\max\{(1+\tau)\cdot U + \psi + \varphi, 1 + \varphi, \psi, 1 + 2\psi\} \leq 1 + \psi + \varphi$. Therefore, when $U \leq \frac{1}{1+\tau}$, matching $p_3$ to $a_1$ or $a_2$ incurs the instantaneous social regret for the three players no less than $1 - \varphi - \psi$.

Finally, from Lemma E.1, we know that the instananeous social regret for $p_1 + p_2 + p_3$ is always non-negative, regardless of the value of $\boldsymbol{U}$ or the chosen matching, we have

$$
Reg_{\boldsymbol{\nu}',\pi}^{p_1+p_2+p_3}(T) = \mathbb{E}\left[\sum_{t=1}^{T}\left[\sum_{i=1}^{3}\left(\boldsymbol{U}_i^*(t) - \boldsymbol{U}_{p_i,\mu_t(p_i)}(t)\right)\right]\right]
$$

$$
\geq \mathbb{E}\left[\sum_{t=1}^{T}\left[\sum_{i=1}^{3}\left(\boldsymbol{U}_i^*(t) - \boldsymbol{U}_{p_i,\mu_t(p_i)}(t)\right)\mathbb{1}\left\{U_t \leq \frac{1}{1+\tau}, p_3 \text{ matches to } a_1\right\}\right]\right]
$$

$$
+ \mathbb{E}\left[\sum_{t=1}^{T}\left[\sum_{i=1}^{3}\left(\boldsymbol{U}_i^*(t) - \boldsymbol{U}_{p_i,\mu_t(p_i)}(t)\right)\mathbb{1}\left\{U_t \leq \frac{1}{1+\tau}, p_3 \text{ matches to } a_2\right\}\right]\right]
$$

$$
\geq (1 - \varphi - \psi)\mathbb{E}_{\boldsymbol{\nu}',\pi}\left[N_{31}^-(T) + N_{32}^-(T)\right]
$$

$$
\geq (1 - \varphi - \psi)\cdot c_7\frac{(1-\psi)\tau T}{8\varphi}
$$

$$
:= c_8\tau T = \Omega\left(T^{\frac{2}{3}}\right)
$$

for some constant $c_8$, which is a contradiction.

Therefore, there does not exist a policy $\pi$ such that the regret for every player is $o(T^{2/3})$, which confirms the theorem. $\square$

**Lemma E.2.** *For instances $\nu$ and $\nu'$, there exists a constant $c = 3$ such that $\mathbb{P}\left(\delta_{\min} \leq \Delta\right) = F(\Delta) \leq c\Delta$ for any $\Delta \leq \frac{1}{16}$.*

*Proof.* **In instance $\nu$:** For the three players, we have

$$\delta_{\min}^{(p_1)} = \min\{1 - U, U\}, \quad \delta_{\min}^{(p_2)} = \frac{1}{2}, \quad \delta_{\min}^{(p_3)} = \frac{1}{8}.$$

Therefore, when $\Delta \leq \frac{1}{16}$, we have

$$\mathbb{P}\left(\delta_{\min} \leq \Delta\right) \leq \mathbb{P}\left(U \geq 1 - \Delta\right) + \mathbb{P}\left(U \leq \Delta\right) = 2\Delta.$$

**In instance $\nu'$:** For the three players, we have

$$\delta_{\min}^{(p_1)} = \min\{(1 + \tau)U, |(1 + \tau)U - 1|\}, \quad \delta_{\min}^{(p_2)} = \frac{1}{2}, \quad \delta_{\min}^{(p_3)} = \frac{1}{8}.$$

Therefore, when $\Delta \leq \frac{1}{16}$, we have

$$\mathbb{P}\left(\delta_{\min} \leq \Delta\right) \leq \mathbb{P}\left((1 + \tau)U \leq \Delta\right) + \mathbb{P}\left(1 - \Delta \leq (1 + \tau)U \leq 1 + \Delta\right) \leq \frac{3}{1 + \tau}\Delta.$$

Combining the two instances, we have that for any $\Delta \leq \frac{1}{16}$, we have

$$\mathbb{P}\left(\delta_{\min} \leq \Delta\right) \leq \max\left\{\frac{3}{1 + \tau}, 2\right\}\Delta \leq 3\Delta.$$

$\square$

**Lemma E.3.** *If $N_u \sim Binomial\left(T, \frac{\tau}{1+\tau}\right)$, we have*

$$\mathbb{P}\left(N_u < \frac{\tau T}{2}\right) \leq \exp\left(-\frac{T\tau(1 - \tau)^2}{8(1 + \tau)}\right).$$

*Proof.* $N_u$ is the sum of independent Bernoulli trials, and $\mathbb{E}[N_u] = \frac{\tau T}{1+\tau}$. Let $\rho$ solving the equality $(1 - \rho)\mathbb{E}[N_u] = \frac{\tau T}{2}$, we have $\rho = \frac{1-\tau}{2}$. From Lemma J.9, we know that

$$\begin{aligned}
\mathbb{P}\left(N_u < \frac{\tau T}{2}\right) &= \mathbb{P}\left(N_u \leq (1 - \rho)\mu\right) \\
&\leq \exp\left(-\frac{\mathbb{E}[N_u]\rho^2}{2}\right) \\
&= \exp\left(-\frac{T\tau(1 - \tau)^2}{8(1 + \tau)}\right).
\end{aligned}$$

$\square$

*Proof of Lemma E.1.* We prove the properties as follows.

**1. Benchmark Utilities**

- Under instance $\nu$, the preference ranking of player $p_1$ over the arms is $a_2 \succ a_1 \succ a_3$ almost surely, while the preference ranking of player $p_2$ is $a_1 \succ a_3 \succ a_2$, and for $p_3$ it is $a_2 \succ a_1 \succ a_3$, and hence the stable matching benchmark is $\mu^* = \{p_1 - a_2, p_2 - a_1, p_3 - a_3\}$, which corresponds to $\boldsymbol{U}^* = (1, 1, 0)^\top$.

- Under instance $\nu'$, when $U \leq \frac{1}{1+\tau}$, we have that $\boldsymbol{U}'_{11} \leq \boldsymbol{U}'_{12}$ almost surely, so that the preference ranking of players over arms would be the same as that in instance $\nu$ and $\boldsymbol{U}^* = (1, 1, 0)^\top$. When $U > \frac{1}{1+\tau}$, the preference ranking of player $p_1$ over the arms would be altered to $a_1 \succ a_2 \succ a_3$ and hence the stable matching benchmark shifts to $\mu^* = \{p_1 - a_1, p_2 - a_3, p_3 - a_2\}$, which corresponds to $\boldsymbol{U}^* = ((1 + \tau)U, \psi, 1)^\top$.

## 2. Regret under Consideration

- In instance $\boldsymbol{\nu}$, for player $p_2$, the maximum utility is from the matching $(p_2, a_1)$, which has a utility of 1. This pair is also the stable matching pair, and hence the regret for $p_2$ is always non-negative.

- In instance $\boldsymbol{\nu'}$, when $U \leq \frac{1}{1+\tau}$, the maximum weight matching is $\{(p_1 - a_2), (p_2 - a_1), (p_3 - a_3)\}$, providing a social welfare of 2; when $U > \frac{1}{1+\tau}$, the maximum weight matching is $\{(p_1 - a_1), (p_2 - a_3), (p_3 - a_2)\}$, providing a social welfare of $\frac{9}{8} + (1+\tau)U > 2$. Therefore, in either case, the stable matching is the same as the maximum weight matching, and hence the social regret for $p_1 + p_2 + p_3$ would be non-negative.

$\square$

# F. Adaptive Explore-Choose Oracle Algorithm

The full version of Adative Explore-Choose Oracle (AdECO) algorithm is presented in Algorithm 5.

---

**Algorithm 5** Adaptive Explore-Choose Oracle (AdECO)

---

**Input:** Time horizon $T$, preference profile $(P_a)_{a \in \mathcal{K}}$, context dimension $d$, parameter $\Delta$, $\varepsilon$, and $\eta$, ridge penalty parameter $\lambda$.

1: $\xi \leftarrow \frac{\Delta - \varepsilon}{4\eta}$.
2: $\boldsymbol{V}_i^{(1)} \leftarrow \lambda \boldsymbol{I}_{d \times d}, \hat{\theta}_i^{(1)} \leftarrow \boldsymbol{0}_d, \forall i \in [N]$.
3: **for** $t = 1, 2, \cdots, T$ **do**
4:      Observe context $\mathrm{x}_j(t)$ for every arm $a_j, \forall j \in [K]$.
5:      **if** $\exists i \in [N], j \in [K]$, s.t. $\|\mathrm{x}_j(t)\|_{(\boldsymbol{V}_i^{(t)})^{-1}} > \xi$ **then** {Exploration}
6:          Collect all $(i,j)$ such that $\|\mathrm{x}_j(t)\|_{(\boldsymbol{V}_i^{(t)})^{-1}} > \xi$, form a bipartite graph $G_t$.
7:          Find a **maximum cardinality matching** $\mu_c$ on $G_t$, let $M^c := \{i \in [N] : i$ is matched in $\mu^c\}$, match

$$\mu_t(i) \leftarrow \begin{cases} \mu^c(i) & \text{if } i \in M^c, \\ \emptyset & \text{otherwise.} \end{cases}$$

8:          Update $\boldsymbol{V}_i^{(t+1)}$ and $\hat{\theta}_i^{(t+1)}, \forall i \in M^c$ according to Eq.(8).
9:          $\boldsymbol{V}_i^{(t)} \leftarrow \boldsymbol{V}_i^{(t-1)}$ and $\hat{\theta}_i^{(t)} \leftarrow \hat{\theta}_i^{(t-1)}, \forall i \in [N] \setminus M^c$.
10:      **else**{Exploitation}
11:          $\hat{\theta}_i^{(t)} \leftarrow \hat{\theta}_i^{(t-1)}, \forall i \in [N]$.
12:          $\hat{\boldsymbol{U}}_{i,j}(t) = \left(\hat{\theta}_i^{(t)}\right)^\top \mathrm{x}_j(t), F_i \leftarrow$ False.
13:          **for** $i = 1, 2, \cdots, N$ **do**
14:              $\hat{\boldsymbol{U}}_{i,\cdot}^{\text{sort}}(t) \leftarrow \text{Sort}\left(\hat{\boldsymbol{U}}_{i,\cdot}(t), \text{decreasing}\right)$
15:              $\Delta_{i,\min} \leftarrow \min\left\{\hat{\boldsymbol{U}}_{i,j}^{\text{sort}}(t) - \hat{\boldsymbol{U}}_{i,j+1}^{\text{sort}}(t)\right\}, j \in [K-1]$
16:              **if** $\Delta_{i,\min} > \frac{\Delta + \varepsilon}{2}$ **then**
17:                  $F_i \leftarrow$ True
18:              **end if**
19:          **end for**
20:          **if** $\forall i \in [N], F_i ==$ True **then** {CIs are well-separated}
21:              $\mu_t(i) \leftarrow \mu^{GS}(i), \forall i \in [N]$. (Input $\hat{\boldsymbol{U}}(t)$ and $(P_a)_{a \in \mathcal{K}}$ for **Gale-Shapley**)
22:          **else**
23:              $\mu_t(i) \leftarrow \mu^{approx}(i), \forall i \in [N]$. (Input $\hat{\boldsymbol{U}}(t)$ and $(P_a)_{a \in \mathcal{K}}$ for $\alpha$-**approximation oracle** with instability tolerance $\frac{\Delta + \varepsilon}{2}$)
24:          **end if**
25:      **end if**
26: **end for**

---

When the confidence intervals for the utilities overlap, we cannot distinguish the strict preference ranking for players over arms. Moreover, when there are ties in the preference profile, the optimal stable share of each player might be achieved from different stable matchings. In this scenario, outputting a deterministic matching with the estimated utility matrix may benefit some players while incurring large regret for others. Therefore, we should consider distributions over matchings and approximation. Here, we assume access to an offline oracle that, given a utility matrix $\boldsymbol{U}$ and the uncertainty set with confidence radius $\gamma$, outputs a randomized matching guaranteeing each player $p_i$ an $\alpha$ of $\boldsymbol{U}_i^\varepsilon(p_i)$ in expectation, with additional error $2\gamma + \varepsilon$.

Before we formally present the definition of the approximation oracle, we first define the $\varepsilon$-optimal stable share within an uncertainty set. Given a utility matrix $\boldsymbol{U}$ and the arms' preference profile $(P_a)_{a \in \mathcal{K}}$, define the set of $\varepsilon$-stable matchings as

$$\mathcal{S}_{\boldsymbol{U}}^\varepsilon := \{\mu : \mu \text{ is an } \varepsilon - \text{stable matching w.r.t. } \boldsymbol{U} \text{ and } (P_a)_{a \in \mathcal{K}}\},$$

and the $\varepsilon$-optimal stable share with respect to $\boldsymbol{U}$ is

$$\boldsymbol{U}_\varepsilon^*(p_i) := \max_{\mu \in \mathcal{S}_{\boldsymbol{U}}^{\bar{\varepsilon}}} \boldsymbol{U}(p_i, \mu(p_i)), \quad \forall i \in [N].$$

Given an uncertainty set $\mathcal{U}$ of utility matrices, we define the $\varepsilon$-optimal stable share within $\mathcal{U}$ as

$$\mathcal{U}_\varepsilon^*(p_i) := \sup_{\boldsymbol{U} \in \mathcal{U}} \max_{\mu \in \mathcal{S}_{\boldsymbol{U}}^{\bar{\varepsilon}}} \boldsymbol{U}(p_i, \mu(p_i)), \quad \forall i \in [N]. \tag{43}$$

The $(\boldsymbol{\alpha}, \gamma, \varepsilon)$-approximation oracle is defined as follows.

**Definition 4** $((\boldsymbol{\alpha}, \gamma, \varepsilon)$-Approximation Oracle). An $(\boldsymbol{\alpha}, \gamma, \varepsilon)$-approximation oracle takes a rectangular uncertainty set $\mathcal{U}$ with center $\boldsymbol{U}$, confidence radius $\gamma$ as input and returns a (randomized) matching $\tilde{\mu}$ satisfying: $\mathbb{E}\left[\boldsymbol{U}_{\tilde{\mu}}(p_i)\right] \geq \boldsymbol{\alpha}^{\mathcal{U}}(p_i) \cdot \mathcal{U}_\varepsilon^*(p_i) - (2\gamma + \varepsilon)$ for every player $p_i$, where $\boldsymbol{\alpha}^{\mathcal{U}} \in (0, 1]^N$ is a player-specific approximation ratio vector (often simplified to $\boldsymbol{\alpha}$). If $\boldsymbol{\alpha}^{\mathcal{U}}(p_i) = \alpha$ is uniform across players and independent of $\mathcal{U}$, we call it an $(\alpha, \gamma, \varepsilon)$-approximation oracle, and without ambiguity, we simply denote it as $\alpha$-approximation oracle.

For example, Algorithm 6 (Lin et al., 2026) guarantees that for any input utility matrix $\boldsymbol{U}$ with instability tolerance $2\gamma + \varepsilon$, let $\mathcal{U}$ be the rectangular uncertainty set with center $\boldsymbol{U}$ and confidence radius $\gamma$, we have $\boldsymbol{\alpha}^{\boldsymbol{U}}(p_i) \geq 1/\lfloor \log_2 N + 2 \rfloor$. We restate the algorithm as follows, while omitting the detailed analysis.

---

**Algorithm 6** $\varepsilon$-Oracle for Approximated Player Optimal Stable Matching

---

**Input:** $N$ players, $K$ arms, Utility matrix $\boldsymbol{U}$ that encodes the preference of players over arms, strict preference profile $P_a$ of arms, an integer $m \geq 1$, and the instability tolerance $\varepsilon \geq 0$.
1: For each arm $a \in \mathcal{A}$, duplicate it $m$ times and denote the $i$-th copy as $a^{(i)}$.
2: Each replica $a^{(i)}$ shares the same preference $P_a$ as the original arm $a$.
3: For every player $p$ and job $a^{(i)}$, define the utility

$$\boldsymbol{U}(p, a^{(i)}) := \boldsymbol{U}(p, a) - (i-1)\varepsilon$$

and use it to generate the players' preference profile $P_w$ (breaking ties in favor of lower indices).
4: Run Gale-Shapley algorithm on $P_w$ and $P_a$ to compute a player-optimal stable matching $\tilde{\mu}$.
5: For each $i \in [m]$, build a matching $\tilde{\mu}_i$, which matches each arm $a$ with $\tilde{\mu}_i(a) := \tilde{\mu}(a^{(i)})$.
**Output:** The distribution $D$ which selects each matching $\tilde{\mu}_i$ with probability $1/m$.

---

**Computational Complexity** Algorithm 6 replicates each arm $\lfloor \log_2 N + 2 \rfloor$ times, assigns utilities accordingly, and then invokes the Gale-Shapley algorithm to compute a matching based on the resulting preference lists. Consequently, the computational complexity is determined by running Gale-Shapley with $N$ players and $K\lfloor \log_2 N + 2 \rfloor$ arms, yielding $\mathcal{O}\left(NK \log_2 N\right)$.

# G. Proof of Theorem 4.1

*Proof.* Given $\Delta$ and $\varepsilon$, we define $\gamma = \frac{\Delta - \varepsilon}{4}$, $\xi = \frac{\Delta - \varepsilon}{4\eta}$, for the exploration phase, denote the estimation indices for player $p_i$ as $\mathcal{G}^{(i)} = \left\{ t_1^{(i)}, \cdots, t_{|\mathcal{G}^{(i)}|}^{(i)} \right\}$. Notice that for $t \in \mathcal{G}^{(i)}$, we have $\|x_j(t)\|_{\left( V_i^{(t)} \right)^{-1}} > \xi$ for $j = \mu_t(i)$, and we only use the contexts and matching rewards from $\mathcal{G}^{(i)}$ to update the Gram matrix $V_i$ and the estimation of the preference parameter $\hat{\theta}_i$. Take $\mathcal{G}$ as the set of rounds in which the market explores, we have $\mathcal{G} = \cup_{i \in [N]} \mathcal{G}^{(i)}$. Define $\mathcal{F} = \left\{ \exists t \notin \mathcal{G}, i \in [N], j \in [K] : |\hat{U}_{i,j}(t) - U_{i,j}(t)| > \gamma \right\}$ as the bad event that some preference of players over arms is not estimated well during the exploitation phase. Using $\gamma$ as the width for the confidence interval constructed for the estimated utility, let $\mathcal{F}_d^{(t)} := \left\{ \hat{U}_{i,j}^{\text{sort}}(t) - \hat{U}_{i,j+1}^{\text{sort}}(t) > 2\gamma + \varepsilon = \frac{\Delta + \varepsilon}{2}, \forall j \in [K-1], \forall i \in [N] \right\}$ as the event that all arms have well-separated confidence intervals for any player $p_i$ at time $t$ in the sense that the distance between any two confidence intervals for the arms is at least $\varepsilon$. Define the instantaneous regret for a player $p_i$ at time $t$ as $(U_i^*(t) \mathbb{1}\{\delta_{\min}(t) > \Delta\} + \alpha U_i^\varepsilon(t) \mathbb{1}\{\delta_{\min}(t) \leq \Delta\}) - U_{i,\mu_t(i)}(t)$, where $\varepsilon < \Delta$ and is typically chosen as $\Delta/2$.

The optimal stable regret for each player $p_i$ by following AdECO (Algorithm 2) satisfies

$$Reg_i^{\alpha,\Delta}(T) = \mathbb{E}\left[\sum_{t=1}^{T}\left[(\boldsymbol{U}_i^*(t)\mathbb{1}\{\delta_{\min}(t) > \Delta\} + \alpha\boldsymbol{U}_i^\varepsilon(t)\mathbb{1}\{\delta_{\min}(t) \leq \Delta\}) - \boldsymbol{U}_{i,\mu_t(i)}(t)\right]\right] \tag{44}$$

$$= \mathbb{E}\left[\sum_{t=1}^{T}\left[(\boldsymbol{U}_i^*(t)\mathbb{1}\{\delta_{\min}(t) > \Delta\} + \alpha\boldsymbol{U}_i^\varepsilon(t)\mathbb{1}\{\delta_{\min}(t) \leq \Delta\}) - \boldsymbol{U}_{i,\mu_t(i)}(t)\right]\mathbb{1}\{\neg\mathcal{F}\}\right]$$

$$+ \mathbb{E}\left[\sum_{t=1}^{T}\left[(\boldsymbol{U}_i^*(t)\mathbb{1}\{\delta_{\min}(t) > \Delta\} + \alpha\boldsymbol{U}_i^\varepsilon(t)\mathbb{1}\{\delta_{\min}(t) \leq \Delta\}) - \boldsymbol{U}_{i,\mu_t(i)}(t)\right]\mathbb{1}\{\mathcal{F}\}\right]$$

$$\leq \mathbb{E}\left[\sum_{t=1}^{T}\left[(\boldsymbol{U}_i^*(t)\mathbb{1}\{\delta_{\min}(t) > \Delta\} + \alpha\boldsymbol{U}_i^\varepsilon(t)\mathbb{1}\{\delta_{\min}(t) \leq \Delta\}) - \boldsymbol{U}_{i,\mu_t(i)}(t)\right]\mathbb{1}\{\neg\mathcal{F}\}\right] + B_y T\mathbb{P}(\mathcal{F}) \tag{45}$$

$$\leq \mathbb{E}\left[\sum_{t=1}^{T}\left[(\boldsymbol{U}_i^*(t)\mathbb{1}\{\delta_{\min}(t) > \Delta\} + \alpha\boldsymbol{U}_i^\varepsilon(t)\mathbb{1}\{\delta_{\min}(t) \leq \Delta\}) - \boldsymbol{U}_{i,\mu_t(i)}(t)\right]\mathbb{1}\{\neg\mathcal{F}\}\right] + B_y T \cdot \frac{N}{T} \tag{46}$$

$$\leq \mathbb{E}\left[\sum_{t\in\mathcal{G}}\left[(\boldsymbol{U}_i^*(t)\mathbb{1}\{\delta_{\min}(t) > \Delta\} + \alpha\boldsymbol{U}_i^\varepsilon(t)\mathbb{1}\{\delta_{\min}(t) \leq \Delta\}) - \boldsymbol{U}_{i,\mu_t(i)}(t)\right]\mathbb{1}\{\neg\mathcal{F}\}\right]$$

$$+ \mathbb{E}\left[\sum_{t\notin\mathcal{G}}\left[(\boldsymbol{U}_i^*(t)\mathbb{1}\{\delta_{\min}(t) > \Delta\} + \alpha\boldsymbol{U}_i^\varepsilon(t)\mathbb{1}\{\delta_{\min}(t) \leq \Delta\}) - \boldsymbol{U}_{i,\mu_t(i)}(t)\right]\mathbb{1}\{\neg\mathcal{F}\}\right] + B_y N$$

$$\leq \mathbb{E}\left[B_y|\mathcal{G}|\mathbb{1}\{\neg\mathcal{F}\}\right] \tag{47}$$

$$+ \mathbb{E}\left[\sum_{t\notin\mathcal{G}}\left[(\boldsymbol{U}_i^*(t)\mathbb{1}\{\delta_{\min}(t) > \Delta\} + \alpha\boldsymbol{U}_i^\varepsilon(t)\mathbb{1}\{\delta_{\min}(t) \leq \Delta\}) - \boldsymbol{U}_{i,\mu_t(i)}(t)\right]\mathbb{1}\left\{\neg\mathcal{F}\cap\mathcal{F}_d^{(t)}\right\}\right]$$

$$+ \mathbb{E}\left[\sum_{t\notin\mathcal{G}}\left[(\boldsymbol{U}_i^*(t)\mathbb{1}\{\delta_{\min}(t) > \Delta\} + \alpha\boldsymbol{U}_i^\varepsilon(t)\mathbb{1}\{\delta_{\min}(t) \leq \Delta\}) - \boldsymbol{U}_{i,\mu_t(i)}(t)\right]\mathbb{1}\left\{\neg\mathcal{F}\cap\neg\mathcal{F}_d^{(t)}\right\}\right] + B_y N$$

$$\leq \mathbb{E}\left[B_y|\mathcal{G}|\mathbb{1}\{\neg\mathcal{F}\}\right] \tag{48}$$

$$+ \mathbb{E}\left[\sum_{t\notin\mathcal{G}}\left[(\boldsymbol{U}_i^*(t)\mathbb{1}\{\delta_{\min}(t) > \Delta\} + \alpha\boldsymbol{U}_i^\varepsilon(t)\mathbb{1}\{\delta_{\min}(t) \leq \Delta\}) - \boldsymbol{U}_{i,\mu_t(i)}(t)\right]\mathbb{1}\left\{\neg\mathcal{F}\cap\neg\mathcal{F}_d^{(t)}\right\}\right] + B_y N$$

$$\leq \mathbb{E}\left[\left(B_y|G| + \frac{\Delta+\varepsilon}{2}\cdot T\right)\mathbb{1}\{\neg\mathcal{F}\}\right] + B_y N \tag{49}$$

$$\leq B_y \frac{\eta^2 Nd\log\left(\frac{T+d\lambda}{\lambda}\right)}{\gamma^2} + \frac{\Delta+\varepsilon}{2}\cdot T + B_y N \tag{50}$$

$$= \mathcal{O}\left(\frac{Nd\log^2 T}{(\Delta-\varepsilon)^2} + \frac{\Delta+\varepsilon}{2}\cdot T\right).$$

Eq.(44) according to the assumption that the noises are i.i.d. with mean 0. Eq.(45) and (47) come from the fact that the expected per-round regret is $(\boldsymbol{U}_i^*(t)\mathbb{1}\{\delta_{\min}(t) > \Delta\} + \alpha\boldsymbol{U}_i^\varepsilon(t)\mathbb{1}\{\delta_{\min}(t) \leq \Delta\}) - \boldsymbol{U}_{i,\mu_t(i)}(t)$ and all terms lie in $[-B_\theta B_x, B_\theta B_x]$ under Assumptions 1 and 3, and hence the expected regret per round is bounded by $2B_\theta B_x = B_y$. Eq.(46) holds based on Lemma G.1. Eq.(48) holds according to Lemma G.2. Eq.(49) follows from Lemma G.3. Eq.(50) holds based on Lemma B.4 by setting the parameters accordingly.

$\square$

**Lemma G.1.** *Let* $\delta = T^{-1}, \eta = R\sqrt{d\log\left(\frac{1+TB_x^2/\lambda}{\delta}\right)} + \lambda^{1/2}B_\theta$, *we have* $\mathbb{P}(\mathcal{F}) \leq \frac{N}{T}$.

*Proof.*

$$\mathbb{P}\left(\mathcal{F}\right) = \mathbb{P}\left(\exists t \notin \mathcal{G}, i \in [N], j \in [K] : |\hat{\boldsymbol{U}}_{i,j}(t) - \boldsymbol{U}_{i,j}(t)| > \gamma\right)$$

$$\leq \sum_{i=1}^{N} \mathbb{P}\left(\exists j \in [K], \exists t \notin \mathcal{G} : |\hat{\boldsymbol{U}}_{i,j}(t) - \boldsymbol{U}_{i,j}(t)| > \gamma\right) \tag{51}$$

$$\leq \sum_{i=1}^{N} \mathbb{P}\left(\exists j \in [K], \exists t \notin \mathcal{G} : \|\hat{\theta}_i - \theta_i\|_{\boldsymbol{V}_i^{(t)}} \|\mathrm{x}_j(t)\|_{\left(\boldsymbol{V}_i^{(t)}\right)^{-1}} > \gamma\right) \tag{52}$$

$$\leq \sum_{i=1}^{N} \mathbb{P}\left(\exists t \notin \mathcal{G} : \|\hat{\theta}_i - \theta_i\|_{\boldsymbol{V}_i^{(t)}} > \eta\right) \tag{53}$$

$$\leq N\delta = \frac{N}{T}. \tag{54}$$

Eq.(51) follows from the union bound. Eq.(52) holds according to Cauchy-Schwarz inequality. Eq.(53) comes from the fact that for any $t \notin \mathcal{G}$, we have $\|\mathrm{x}_j(t)\|_{\left(\boldsymbol{V}_i^{(t)}\right)^{-1}} \leq \xi = \frac{\gamma}{\eta}$ in Algorithm 2. Eq.(54) holds according to Lemma J.2. $\qquad\square$

**Lemma G.2.** *At time $t$, under the event $\neg\mathcal{F} \cap \mathcal{F}_d^{(t)}$, the instantaneous regret of Algorithm 2 is non-positive, i.e.,*

$$\mathbb{E}\left[\left((\boldsymbol{U}_i^*(t)\mathbb{1}\{\delta_{\min}(t) > \Delta\} + \alpha\boldsymbol{U}_i^\varepsilon(t)\mathbb{1}\{\delta_{\min}(t) \leq \Delta\}) - \boldsymbol{U}_{i,\mu_t(i)}(t)\right)\mathbb{1}\left\{\neg\mathcal{F} \cap \mathcal{F}_d^{(t)}\right\}\right] \leq 0.$$

*Proof.* When $\delta_{\min}(t) > \Delta$, the Gale-Shapley subroutine called by Algorithm 2 outputs the unique player-optimal stable matching with respect to the ground-truth utility matrix $\boldsymbol{U}$, as the preference rankings of players over arms induced from $\hat{\boldsymbol{U}}$ would be the same as those from $\boldsymbol{U}$ under the event $\neg\mathcal{F} \cap \mathcal{F}_d^{(t)}$. Specifically, for any player $p_i$, any two arms $a_j$ and $a_{j'}$ such that $\hat{\boldsymbol{U}}_{i,j}(t) > \hat{\boldsymbol{U}}_{i,j'}(t)$, under the event $\mathcal{F}_d^{(t)}$, we have $\hat{\boldsymbol{U}}_{i,j}(t) - \hat{\boldsymbol{U}}_{i,j'}(t) \geq 2\gamma + \varepsilon$, so that $\boldsymbol{U}_{i,j}(t) - \boldsymbol{U}_{i,j'}(t) \geq \hat{\boldsymbol{U}}_{i,j}(t) - \gamma - (\hat{\boldsymbol{U}}_{i,j'}(t) + \gamma) \geq \varepsilon$. Therefore, every player received the expected reward equal to their corresponding OSS in time $t$, leading to the instantaneous regret being 0.

When $\delta_{\min}(t) \leq \Delta$, from Lemma G.4, we know that if we call the Gale-Shapley oracle with the estimated utility matrix $\hat{\boldsymbol{U}}$ and the preference profile $P_a$, the resulting matching gives every player the expected reward equal to $\boldsymbol{U}_i^\varepsilon(t)$ with respect to the ground-truth utility matrix $\boldsymbol{U}$ and $P_a$. This is because under the event $\neg\mathcal{F} \cap \mathcal{F}_d^{(t)}$, the minimum difference between any two arms is no less than $\varepsilon$. And since $\alpha \in (0, 1]$, we have that $\boldsymbol{U}_i^*(t) = \boldsymbol{U}_i^\varepsilon(t) \geq \alpha\boldsymbol{U}_i^\varepsilon(t)$, and hence the instantaneous regret is non-positive. $\qquad\square$

**Lemma G.3.** *At time $t$, under the event $\neg\mathcal{F} \cap \neg\mathcal{F}_d^{(t)}$, the instantaneous regret of Algorithm 2 is no larger than $\frac{\Delta+\varepsilon}{2}$, i.e.,*

$$\mathbb{E}\left[\left[(\boldsymbol{U}_i^*(t)\mathbb{1}\{\delta_{\min}(t) > \Delta\} + \alpha\boldsymbol{U}_i^\varepsilon(t)\mathbb{1}\{\delta_{\min}(t) \leq \Delta\}) - \boldsymbol{U}_{i,\mu_t(i)}(t)\right]\mathbb{1}\left\{\neg\mathcal{F} \cap \neg\mathcal{F}_d^{(t)}\right\}\right] \leq \frac{\Delta+\varepsilon}{2}.$$

*Proof.* At time $t$, under the event $\neg\mathcal{F} \cap \neg\mathcal{F}_d^{(t)}$, there exists a player $p_i$, two arms $a_j$ and $a_{j'}$ such that $\hat{\boldsymbol{U}}_{i,j}(t) - \hat{\boldsymbol{U}}_{i,j'}(t) \leq 2\gamma + \varepsilon$ while $|\hat{\boldsymbol{U}}_{i,j}(t) - \boldsymbol{U}_{i,j}(t)| \leq \gamma$ and $|\hat{\boldsymbol{U}}_{i,j'}(t) - \boldsymbol{U}_{i,j'}(t)| \leq \gamma$. Therefore, $|\boldsymbol{U}_{i,j}(t) - \boldsymbol{U}_{i,j'}(t)| \leq 4\gamma + \varepsilon$. Since $\gamma = \frac{\Delta-\varepsilon}{4}$, we have $\delta_{\min}(t) \leq 4\gamma + \varepsilon = \Delta$, which implies that the benchmark utility would be $\alpha\boldsymbol{U}_i^\varepsilon(t)$ in this case.

Under the event $\neg\mathcal{F}$, the ground-truth utility matrix $\boldsymbol{U}$ lies in the uncertainty set $\mathcal{U}$ with center $\hat{\boldsymbol{U}}$ and radius $\gamma$, and hence $\boldsymbol{U}_i^\varepsilon \leq \mathcal{U}_\varepsilon^*(p_i)$ for any $i \in [N]$ according to the definition of $\mathcal{U}_\varepsilon^*(p_i)$ (Eq.(43)). From Definition 4, we know that when calling the $(\alpha, \gamma, \varepsilon)$-approximation oracle with input $\hat{\boldsymbol{U}}$ and confidence radius $\gamma$, the receiving expected utility satisfies

$$\mathbb{E}[\boldsymbol{U}_{i,\mu_t(i)}(t)] \geq (\alpha\mathcal{U}_\varepsilon^*(p_i))(t) - (2\gamma + \varepsilon) \geq \alpha\boldsymbol{U}_i^\varepsilon(t) - (2\gamma + \varepsilon)$$

Specifically, we prove the corresponding property for Algorithm 6 in Lemma G.5.

Therefore, when setting the approximation ratio $\alpha = 1/\lfloor\log_2 N + 2\rfloor$, we have the instantaneous regret

$$\mathbb{E}\left[\left(\alpha\boldsymbol{U}_i^\varepsilon(t) - \boldsymbol{U}_{i,\mu_t(i)}(t)\right)\mathbb{1}\left\{\neg\mathcal{F} \cap \neg\mathcal{F}_d^{(t)}\right\}\right] \leq 2\gamma + \varepsilon = \frac{\Delta+\varepsilon}{2}.$$

$\qquad\square$

**Lemma G.4.** *Given a utility matrix $U$ and the preference profile $P_a$ with $N$ players and $K$ arms, define the minimum difference between any two utilities for player $p_i$ as*

$$\delta_{\min}^{(i)} = \min_{j,j' \in [K]} |U_{i,j} - U_{i,j'}|,$$

*and $\delta_{\min} = \min_{i \in [N]} \delta_{\min}^{(i)}$. If $\delta_{\min} \geq \varepsilon$, the set of $\varepsilon$-stable matchings $\mathcal{S}^\varepsilon$ is the same as the set of stable matching $\mathcal{S}$ with respect to $U$ and $P_a$, and hence for any player $p_i$, $U_i^\varepsilon = U_i^*$.*

*Proof.* From the definition of the set of stable matchings $\mathcal{S}^\varepsilon$, we know that $\mathcal{S} \subseteq \mathcal{S}^\varepsilon$. If there is a matching $\mu \in \mathcal{S}^\varepsilon$ such that $\mu \notin \mathcal{S}$, there exists a blocking pair $(p_i, a_j)$ such that $p_i \succ_{a_j} \mu(a_j)$ and $U_{i,j} > U_{i,\mu(p_i)}$. Since $\delta_{\min} \geq \varepsilon$, we also have $U_{i,j} \geq U_{i,\mu(p_i)} + \varepsilon$, which implies $(p_i, a_j)$ is also an $\varepsilon$-blocking pair, violating the $\varepsilon$-stability of $\mu$, which is a contradiction. Therefore, $\mathcal{S} = \mathcal{S}^\varepsilon$ and hence $U_i^\varepsilon = U_i^*$. $\square$

**Lemma G.5.** *Given an uncertainty set $\mathcal{U}$, define the center $\hat{U}$ of the set $\mathcal{U}$, and the uncertainty parameter as*

$$\hat{U}(p, a) = \frac{\inf_{U \in \mathcal{U}} U(p, a) + \sup_{U \in \mathcal{U}} U(p, a)}{2}, \quad \gamma = \frac{1}{2} \sup_{U_1, U_2 \in \mathcal{U}} \|U_1 - U_2\|_{\max}.$$

*Run the $\alpha$-approximation oracle defined in Algorithm 6 with input $\hat{U}$, $m = \lfloor \log_2 N + 2 \rfloor$ and instability tolerance $2\gamma + \varepsilon$, we could generate a distribution $D \in \Delta(\mathcal{I})$ such that*

$$U_D(p_i) \geq \frac{\mathcal{U}_\varepsilon^*(p_i)}{m} - (2\gamma + \epsilon), \quad \forall i \in [N].$$

*Proof.* By running the $\alpha$-approximation oracle with $\hat{U}$, $2\gamma + \varepsilon$, $\lfloor \log_2 N + 2 \rfloor$, from Lemma J.10, we have

$$U_D(p_i) \geq \frac{\hat{U}_{2\gamma+\varepsilon}^*(p_i)}{m} - (2\gamma + \varepsilon), \quad \forall i \in [N]. \tag{55}$$

By construction, any utility matrix $U \in \mathcal{U}$ satisfies $\|U - \hat{U}\|_{\max} \leq \gamma$. From Lemma G.6, we know that for any matching $\mu \in \mathcal{S}_U^\varepsilon, \forall U \in \mathcal{U}$, we have $\mu \in \mathcal{S}_{\hat{U}}^{2\gamma+\varepsilon}$, that is $\bigcup_{U \in \mathcal{U}} \mathcal{S}_U^\varepsilon \subseteq \mathcal{S}_{\hat{U}}^{2\gamma+\varepsilon}$. Therefore, for the $\varepsilon$-optimal stable share, we have

$$\mathcal{U}_\varepsilon^*(p_i) \leq \hat{U}_{2\gamma+\varepsilon}^*(p_i), \quad \forall i \in [N]. \tag{56}$$

Combining Eq.(55) and (56), the conclusion holds. $\square$

**Lemma G.6.** *Fix the preferences of arms over players. Given two utility matrix $U_1$ and $U_2$ such that $\|U_1 - U_2\|_{\max} < \gamma$, then any $\varepsilon$-stable matching $\mu$ for $U_1$, is also $(2\gamma + \varepsilon)$-stable with respect to $U_2$.*

*Proof.* If a matching $\mu$ is $\varepsilon$-stable with respect to $U_1$, then for any $(p, a)$ pair such that $p \succ_a \mu(a)$, we must have

$$U_1(p, a) \leq U(p, \mu(p)) + \varepsilon, \tag{57}$$

from the definition of $\varepsilon$-stable matching.

Since $\|U_1 - U_2\|_{\max} \leq \gamma$, we have that for any $(p, a)$ pair, $|U_1(p, a) - U(p, a)| \leq \gamma$. Therefore,

$$U_2(w, a) \leq U_1(w, a) + \gamma \leq U_1(w, \mu(w)) + \gamma + \epsilon \leq U_2(w, \mu(w)) + 2\gamma + \epsilon, \tag{58}$$

where the first and the last inequality come from $\|U_1 - U_2\|_{\max} \leq \gamma$, while the second inequality holds according to Eq.(57). Therefore, combining $p \succ_a \mu(a)$ and Eq.(58), we can conlude that matching $\mu$ is $(2\gamma + \varepsilon)$-stable with respect to $U_2$. $\square$

*Remark G.1.* Interestingly, we can show that when the top-$N$ arms are sufficiently well-separated – specifically, when there is a gap of at least $\varepsilon$ between the confidence intervals of any two such arms for any player $p_i$ – choosing the Gale-Shapley oracle yields the same regret guarantee as the original selection rule. Concretely, when taking $j \in [N]$ in Line 15 of Algorithm 5, we arrive at an identical result. This equivalence holds because, under such separation, we have $U_i^*(t) = U_i^\varepsilon(t)$ for every player $p_i$. For brevity, however, we omit a detailed proof of this case.

# H. Additional Numerical Experiments

In this section, we provide additional numerical experiment results.

For the stochastic contexts case, we evaluate player-optimal stable regret in a market with $N = 4$ players and $K = 4$ arms. Contexts have dimension $d = 3$. Each arm's preference over players is a random permutation of $\{1, \cdots, N\}$. Each entry of the preference parameter $\theta_i$ is uniformly randomly sampled from $[0, 1]$; while for the contexts, we firstly sample a Gaussian distribution with mean 10 and variance 1 for each entry and then normalize the vector so that $\|x_j(t)\|_2 = 1$.

The ridge penalty parameter is set to $\lambda = 1$ for all algorithms under comparison. For ETC, the exploration length is $h = 5000$; for Batched-ETC, the initial exploration length is $T_1 = 100$; and for BARB, the initial candidate gap is $\Delta_1 = 0.5$. In short, the algorithm parameter settings are the same as those in the example presented in the main text, while this example is in good condition in the sense that the minimum eigenvalue of the context covariance matrix would be large compared to the first example.

Figure 2 shows the player-optimal stable regret for each player. Due to batched explorations, the performances of BARB and Batched-ETC are slightly worse than that of ETC. However, when the minimum eigenvalue of the context covariance matrix vanishes, the performance of ETC and Batched-ETC may significantly deteriorate, as shown in Figure 1, which implies BARB should be a more robust choice.

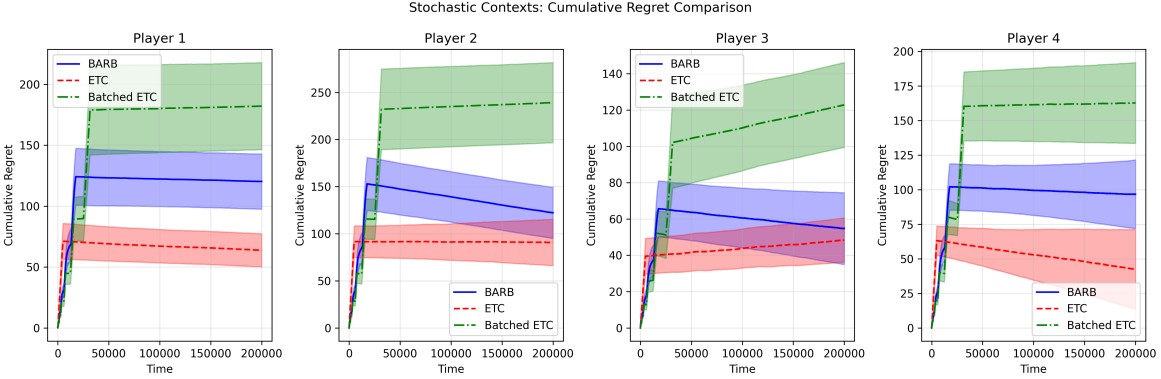

*Figure 2.* Experimental Comparison of BARB, Batched-ETC, and ETC in a market with 4 players and 4 arms.

Moreover, we compare the performance of BARB, ETC, and Batched-ETC in a larger market with 25 players and 25 arms. The dimension of the contexts is 3. The preferences of players over arms and the contexts are generated with the same idea as in the example presented in the main text, such that the minimum eigenvalue of the context covariance matrix is small. For the ETC algorithm, the exploration length is set to $h = 10,000$, while all other algorithmic parameters remain the same as in the example in the main text. Figure 3 shows the maximum player-optimal stable regret for each algorithm. In this large market setting, BARB achieves the smallest maximum cumulative regret, followed by ETC, and Batched-ETC performs the worst. This conclusion is aligned with that in the small market setting.

For generality, we vary the market size $N = K \in \{3, 6, 9, 12\}$ to show the performance of the algorithm BARB. In Figure 4, the result shows that the maximum cumulative regret over players increases as the market size becomes larger. This dependency is consistent with the theoretical results.

Furthermore, to assess the robustness of the algorithm's performance with respect to the choice of the initial candidate gap $\Delta_1$, we conduct a sensitivity analysis by varying $\Delta_1$ over the set $0.4, 0.6, 0.8, 1.0$. The results, presented in Figure 5, show that the maximum cumulative regrets over players under different choices exhibit little variation, indicating that the algorithm is robust to this parameter selection.

For the adversarial contexts, we consider a market with $N = 4$ players and $K = 4$ arms. Contexts have dimension $d = 3$. Each arm's preference over players is a random permutation of $\{1, \cdots, N\}$. To confuse the algorithm, we intentionally alternate between generating contexts, thereby producing a utility matrix characterized by a mix of large and small gaps.

Given that the $\varepsilon$-optimal stable share is typically NP-hard to compute, we compare the rewards collected by each player under two conditions: (i) using the Gale-Shapley or approximation oracle with a known utility matrix, and (ii) following the

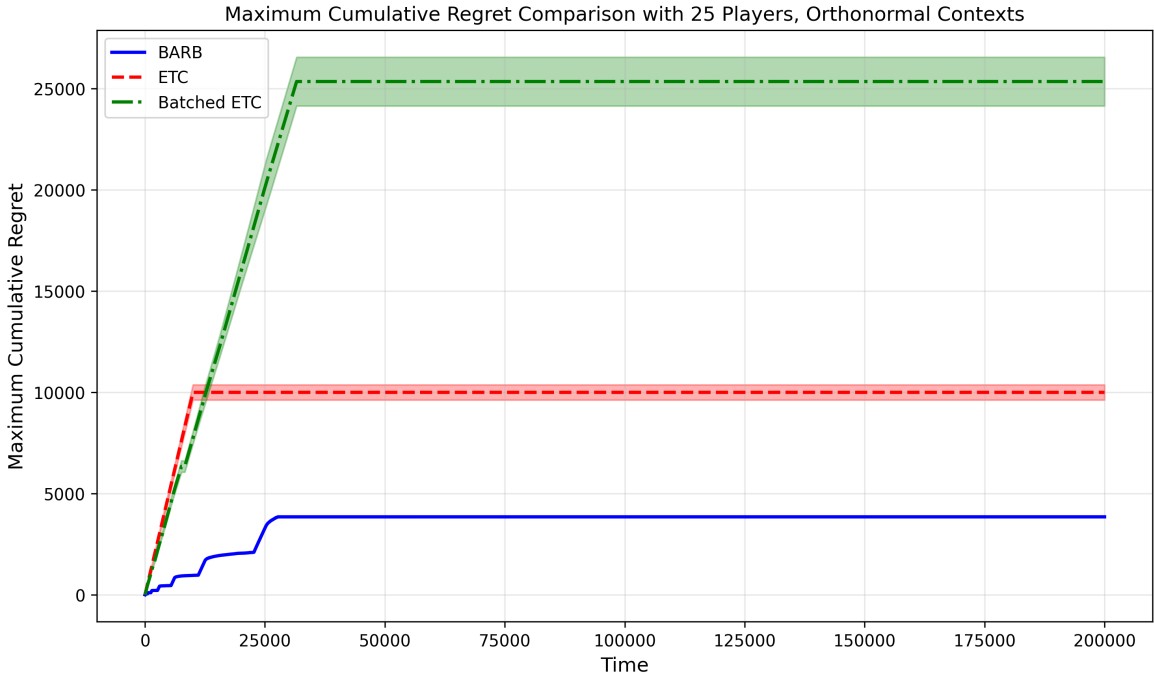

*Figure 3.* Experimental Comparison of BARB, Batched-ETC, and ETC with 25 players and 25 arms in a market with small minimum preference gap.

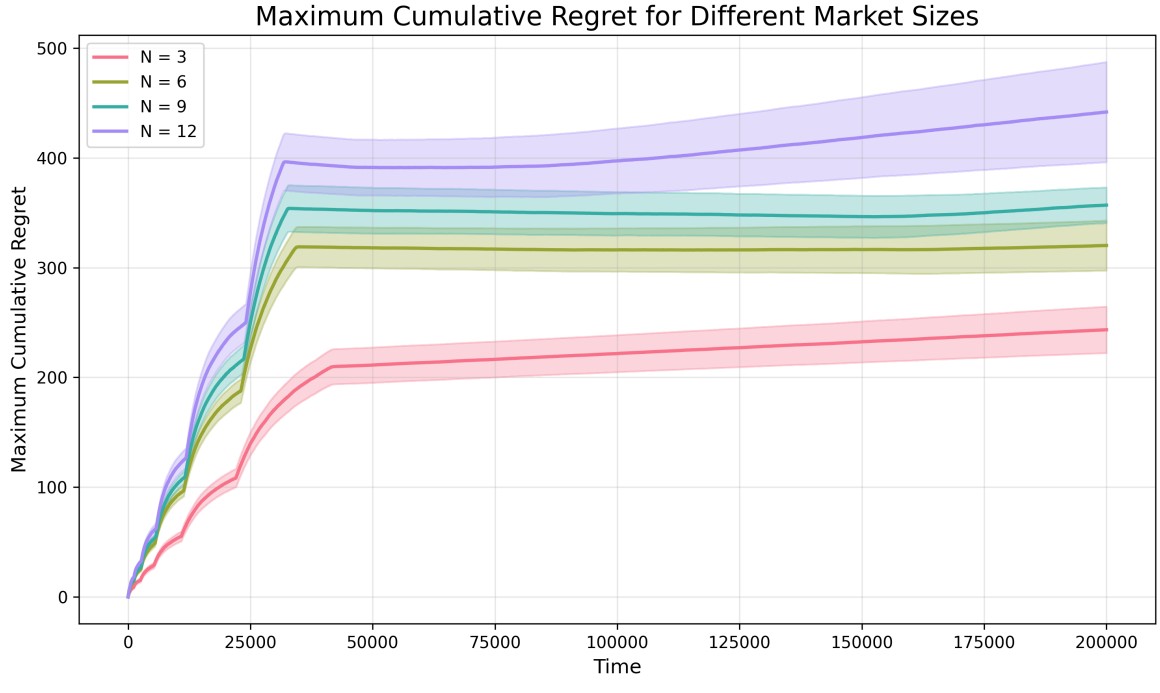

*Figure 4.* Experimental comparison of BARB with different market sizes.

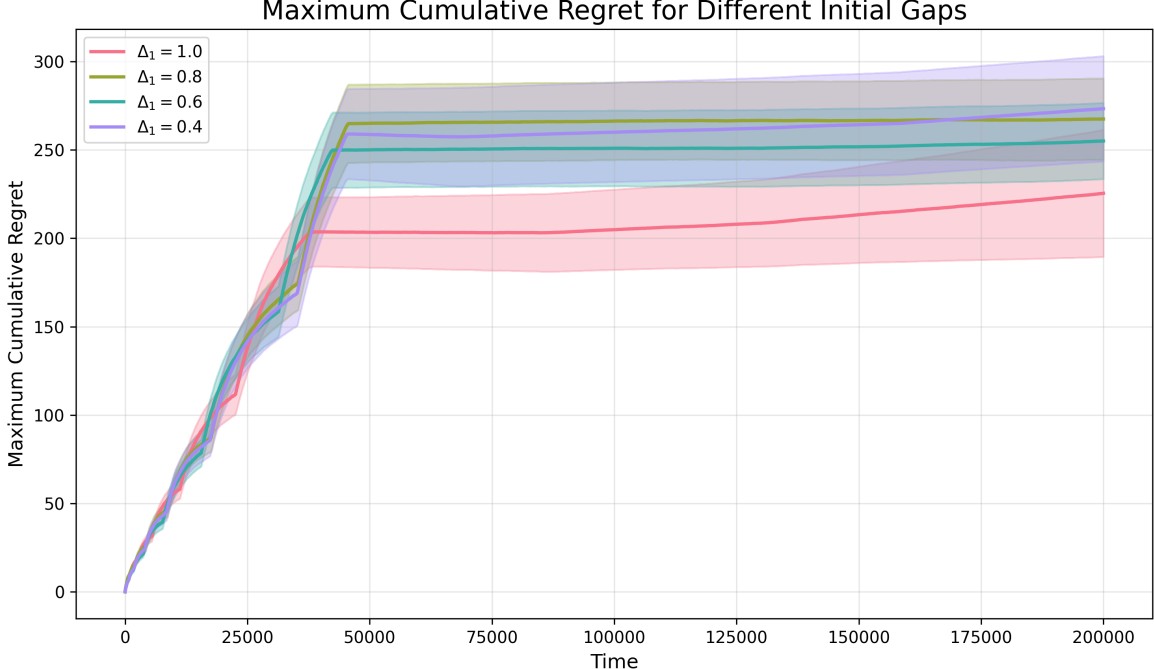

*Figure 5.* Experimental comparison of BARB with different initial candidate gaps $\Delta_1$ in a market with $N = 4$ players and $K = 4$ arms.

AdECO algorithm (Algorithm 2).

Figure 6 shows the numerical results for each player in the adversarial setting. From Figure 6, we find that the collected reward curve of AdECO closely follows that of the oracle, demonstrating that its adaptive exploration strategy provides a good estimation.

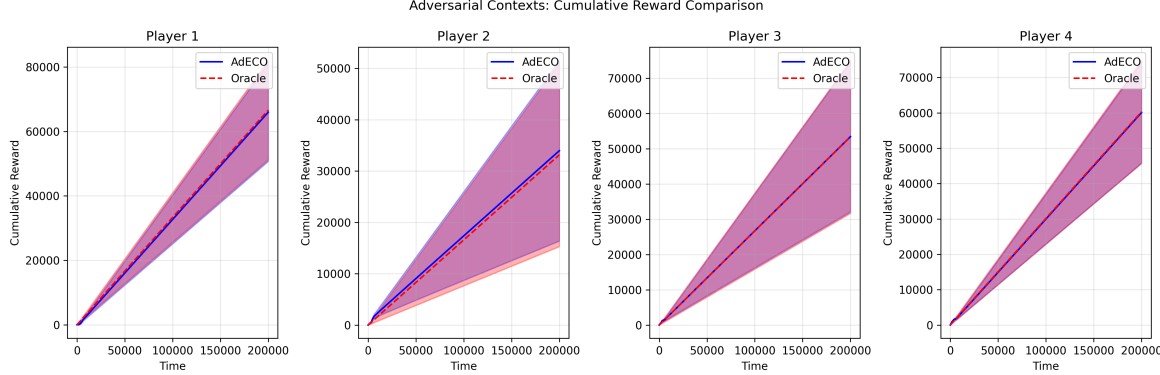

*Figure 6.* Experimental Comparison of AdECO and the offline oracle with 4 players and 4 arms.

Furthermore, we consider a more challenging scenario where, in each round, the adversary randomly selects the market to be in either the large-gap or small-gap region. We vary the probability of choosing the small-gap region as $p \in \{0.1, 0.5, 0.9\}$ and report the difference between the reward collected by the oracle—an algorithm that exactly knows the ground-truth utility and selects the offline oracle accordingly—and that collected by AdECO. The maximum difference across players is shown in Figure 7. As the probability of the small-gap regime increases, the reward difference between the two algorithms decreases. This trend may be attributed to the choice of performance metric in each regime: in the small-gap regime, we use approximation regret as the metric, whereas in the large-gap regime, we use standard regret. The difference in scale between these two metrics likely accounts for the observed results.

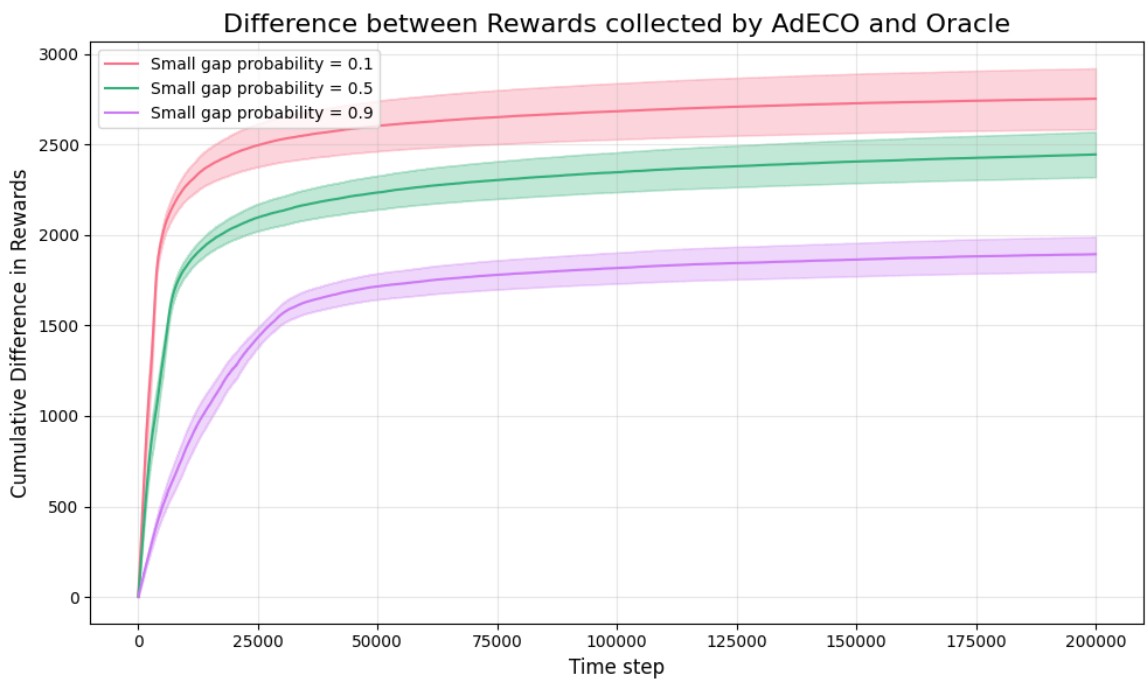

*Figure 7.* Adversarial contexts with varying probability of choosing the small-gap regime.

# I. Illustration of the Minimum Difference $\delta_{\min}$

In this section, we provide an illustrative example on the minimum difference $\delta_{\min}$.

For simplicity, we consider a stochastic contextual market with one player and three arms and the dimension of the context is $d = 1$. The latent preference parameter $\theta_1 = 1$, and the context distribution for the three arms are $Unif[0, \frac{1}{2}]$, $Unif[\frac{1}{4}, \frac{3}{4}]$ and $Unif[\frac{1}{2}, 1]$, respectively. Therefore, the distributions of the matching utilities are $U_1 \sim Unif[0, \frac{1}{2}]$, $U_2 \sim Unif[\frac{1}{4}, \frac{3}{4}]$ and $U_3 \sim Unif[\frac{1}{2}, 1]$, respectively [4].

According to the definition of minimum preference, we have

$$\delta_{\min} = \min\left\{|U_1 - U_2|, |U_2 - U_3|, |U_3 - U_1|\right\}.$$

To derive the CDF $F_{\delta_{\min}}(x)$ of the minimum difference, we have that

$$F_{\delta_{\min}}(x) = \mathbb{P}\left(\delta_{\min} \le x\right) = 1 - \mathbb{P}\left(\delta_{\min} > x\right).$$

Let $S(x) := \mathbb{P}\left(\delta_{\min} > x\right)$, given that the utilities all follow uniform distribution, we have that the joint density is constant on the cube

$$\Omega = \left[0, \frac{1}{2}\right] \times \left[\frac{1}{4}, \frac{3}{4}\right] \times \left[\frac{1}{2}, 1\right], \quad Vol\left(\Omega\right) = \frac{1}{8}.$$

Therefore,

$$S(x) = 8 \cdot Vol\left(\{(U_1, U_2, U_3) \in \Omega : |U_i - U_j| > x, \forall i < j\}\right).$$

We partition the cube $\Omega$ into regions where the order of $(U_1, U_2, U_3)$ is fixed. Only the following orderings have positive volume:

(A) $U_1 \le U_2 \le U_3$.

(B) $U_2 \le U_1 \le U_3$.

(C) $U_1 \le U_3 \le U_2$.

Denote the corresponding volumes as $V_A$, $V_B$ and $V_C$, we have that

$$S(x) = 8 \cdot (V_A(x) + V_B(x) + V_C(x)).$$

We work with different regions separately.

**Region A** : $U_1 \le U_2 \le \mu_3$.

Here, $\delta_{\min} = \min\{U_2 - U_1, U_3 - U_2\}$, thus we have $U_2 - U_1 > x$ and $U_3 - U_2 > x$ in region A.

Therefore, given $U_2$, we have

$$U_1 \in \left[0, \min\left\{\frac{1}{2}, U_2 - x\right\}\right], \quad U_3 \in \left[\max\left\{\frac{1}{2}, U_2 + x\right\}, 1\right],$$

and hence

$$V_A(x) = \int_{U_2} \max\left\{0, \min\left\{\frac{1}{2}, U_2 - x\right\}\right\} \cdot \max\left\{0, 1 - \max\left\{\frac{1}{2}, U_2 + x\right\}\right\} dU_2,$$

which makes

$$V_A(x) = \begin{cases} \frac{4}{3}x^3 - \frac{1}{2}x^2 - \frac{1}{4}x + \frac{3}{32}, & 0 \le x \le \frac{1}{4}, \\ -\frac{4}{3}x^3 + 2x^2 - x + \frac{1}{6}, & \frac{1}{4} \le x \le \frac{1}{2}. \end{cases}$$

---

[4] Without causing ambiguity, we drop the index of the player and focus on the index of the arm.

**Region B**  : $U_2 \leq U_1 \leq U_3$.

Here, $\delta_{\min} = \min \{U_1 - U_2, U_3 - U_1\}$, then given $U_1$, we have

$$U_2 \in \left[\frac{1}{4}, \min\left\{\frac{3}{4}, U_1 - x\right\}\right], \quad U_3 \in \left[\max\left\{\frac{1}{2}, U_1 + x\right\}, 1\right].$$

With a similar derivation to that in region A, we have that

$$V_B(x) = \begin{cases} \frac{2}{3}x^3 + \frac{1}{8}x^2 - \frac{1}{8}x + \frac{1}{64}, & 0 \leq x \leq \frac{1}{8} \\ -\frac{2}{3}x^3 + \frac{5}{8}x^2 - \frac{3}{16}x + \frac{7}{384}, & \frac{1}{8} \leq x \leq \frac{1}{4}. \end{cases}$$

**Region C**  : $U_1 \leq U_3 \leq U_2$.

Here, $\delta_{\min} = \min \{U_3 - U_1, U_2 - U_3\}$, then given $U_3$, we have that

$$U_1 \in \left[0, \min\left\{\frac{1}{2}, U_3 - x\right\}\right], \quad U_2 \in \left[\max\left\{\frac{1}{4}, U_3 + x\right\}, \frac{3}{4}\right].$$

Similarly, we have

$$V_C(x) = \begin{cases} \frac{2}{3}x^3 + \frac{1}{8}x^2 - \frac{1}{8}x + \frac{1}{64}, & 0 \leq x \leq \frac{1}{8} \\ -\frac{2}{3}x^3 + \frac{5}{8}x^2 - \frac{3}{16}x + \frac{7}{384}, & \frac{1}{8} \leq x \leq \frac{1}{4}. \end{cases}$$

Combining the above three regions, we have the final CDF of $\delta_{\min}$ as $F_{\delta_{\min}}(x) = 1 - S(x)$, and

$$F_{\delta_{\min}}(x) = \begin{cases} 0, & x < 0 \\ 1 - 8 \cdot \left(\frac{8}{3}x^3 - \frac{1}{4}x^2 - \frac{1}{2}x + \frac{1}{8}\right), & 0 \leq x \leq \frac{1}{8}, \\ 1 - 8 \cdot \left(\frac{3}{4}x^2 - \frac{5}{8}x + \frac{25}{192}\right), & \frac{1}{8} \leq x \leq \frac{1}{4} \\ 1 - 8 \cdot \left(-\frac{4}{3}x^3 + 2x^2 - x + \frac{1}{6}\right), & \frac{1}{4} \leq x \leq \frac{1}{2} \\ 1, & x \geq \frac{1}{2}. \end{cases}$$

The cumulative distribution function (CDF) of $\delta_{\min}$ for the above example is depicted in Figure 8. We also plot the function $\frac{\log T}{T x^2}$ for three distinct time horizons: 1000, 10000, and 100000. In this plot, the crossing points correspond to the minimum preference gap $\Delta_{\min}$ (Definition 2) for each respective $T$.

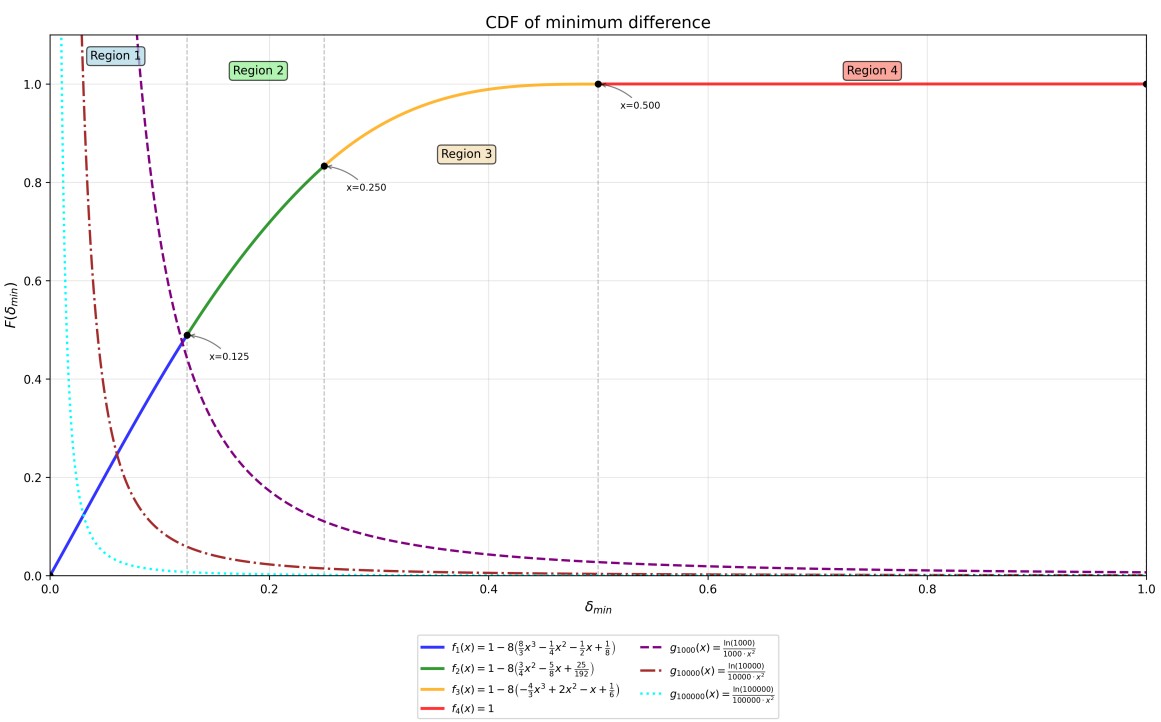

*Figure 8.* CDF of $\delta_{\min}$ and the minimum preference gap.

# J. Technical Lemmas

**Lemma J.1** (Elliptical Potential Lemma, Proposition 1 (Carpentier et al., 2020)). *Let $x_1, \cdots, x_T$ be a sequence of arbitrary vectors in $\mathbb{R}^d$ such that $\|x_i\|_2 \leq 1$. For any $1 \leq t \leq T$, define $\boldsymbol{V}_t = \sum_{s=1}^{t-1} x_s x_s^\top + \lambda \boldsymbol{I}$. The elliptical potential satisfies*

$$\sum_{t=1}^T \|x_t\|_{\boldsymbol{V}_{t+1}^{-1}} = \sum_{t=1}^T \sqrt{x_t^\top \boldsymbol{V}_{t+1}^{-1} x_t} \leq \sqrt{Td \log\left(\frac{T + d\lambda}{d\lambda}\right)},$$

*and for any $p \in \mathbb{R}^+, \lambda \geq 1$, we have*

$$\sum_{t=1}^T \|x_t\|_{\boldsymbol{V}_{t+1}^{-p}} \leq \sum_{t=1}^T \|x_t\|_{\boldsymbol{V}_t^{-p}} \leq 2^{\frac{p}{2}} \sum_{t=1}^T \|x_t\|_{\boldsymbol{V}_{t+1}^{-p}}.$$

**Lemma J.2** (Confidence Ellipsoid, Theorem 2 (Abbasi-Yadkori et al., 2011)). *Let $\mathrm{x}_1, \cdots, \mathrm{x}_T$ be a sequence of vectors in $\mathbb{R}^d$ such that for all $t \geq 1$, $\|\mathrm{x}_t\|_2 \leq B_x$. Let $\theta \in \mathbb{R}^d$ satisfy $\|\theta\|_2 \leq B_\theta$. Let $\{F_t\}_{t=0}^\infty$ be a filtration such that $\mathrm{x}_t$ is $F_{t-1}$-measurable, let $\{\epsilon_t\}$ be a real-valued stochastic process such that $\epsilon_t$ is $F_t$-measurable and $\epsilon_t$ is conditionally $R$-subgaussian for some $R \geq 0$, i.e.*

$$\forall \lambda \in \mathbb{R}, \quad \mathbb{E}\left[e^{\lambda \epsilon_t} \mid F_{t-1}\right] \leq \exp\left(\frac{\lambda^2 R^2}{2}\right).$$

*Let $y_t = \theta^\top \mathrm{x}_t + \epsilon_t$ such that $y_t$ is $F_t$-measurable. Let $\boldsymbol{V} = \lambda \boldsymbol{I}$ for $\lambda > 0$, define*

$$\boldsymbol{V}_t = \boldsymbol{V} + \sum_{s=1}^t \mathrm{x}_s \mathrm{x}_s^\top, \quad \boldsymbol{S}_t = \sum_{s=1}^t y_s \mathrm{x}_s,$$

*and let $\hat{\theta}_t$ be the $\ell^2$-regularized least square estimate of $\theta$ with regularization parameter $\lambda$, i.e.,*

$$\hat{\theta}_t = \boldsymbol{V}_t^{-1} S_t.$$

*Then, for any $\delta > 0$, with probability at least $1 - \delta$, for all $t \geq 0$, $\theta$ lies in the set*

$$\mathcal{C}_t = \left\{\theta \in \mathbb{R}^d : \|\hat{\theta}_t - \theta\|_{\boldsymbol{V}_t} \leq R\sqrt{d \log\left(\frac{1 + tB_x^2/\lambda}{\delta}\right)} + \lambda^{1/2} B_\theta\right\}.$$

**Lemma J.3** (Multiplicative Chernoff Bound, Deviation Above the Mean, Theorem 4.4 (Mitzenmacher & Upfal, 2017)). *Let $X_1, \cdots, X_n$ be independent Bernoulli trials such that $\mathbb{P}(X_i) = p_i$, Let $X = \sum_{i=1}^n X_i$ and $\mu = \mathbb{E}[X]$. Then for any $\rho \geq 0$, we have*

$$\mathbb{P}(X \geq (1 + \rho)\mu) \leq \exp\left(-\frac{\rho^2 \mu}{2 + \rho}\right).$$

**Lemma J.4** (Cauchy's Eigenvalue Interlacing Theorem, Theorem 4.3.17 (Horn & Johnson, 2012)). *Suppose $A \in \mathbb{R}^{n \times n}$ is symmetric. Let $B \in \mathbb{R}^{m \times m}$ with $m < n$ be a principal submatrix (obtained by deleting both $i$-th row and $i$-th column for some values of $i$. Suppose $A$ has eigenvalues $\lambda_1 \leq \cdots \leq \lambda_n$ and $B$ has eigenvalues $\beta_1 \leq \cdots \leq \beta_m$. Then*

$$\lambda_k \leq \beta_k \leq \lambda_{k+n-m} \quad for \quad k = 1, \cdots, m.$$

**Lemma J.5** (Matrix Bernstein Inequality, Theorem 1.6.2 (Tropp, 2015)). *Let $S_1, \cdots, S_n$ be independent, random matrices with common dimensions $d \times d$, and assume that $\|S_i\|_2 \leq M$. Let $\mathbb{E}[S_i] = \Sigma$, $V = \|\sum_{i=1}^n \mathbb{E}\left[(S_i - \Sigma)^2\right]\| \leq n\|\Sigma\|_2^2$. Then for any $t \geq 0$,*

$$\mathbb{P}\left(\|\frac{1}{n}\sum_{i=1}^n S_i - \Sigma\|_2 \geq t\right) \leq 2d \cdot \exp\left(-\frac{nt^2/2}{V/n + Mt/3}\right) \lesssim 2d \exp\left(-C_1 n\right),$$

*where $C_1$ is a constant.*

**Lemma J.6** (Divergence Decomposition, Lemma 15.1 (Lattimore & Szepesvári, 2020)). *For two bandit instances $\boldsymbol{\nu} = \{\nu_{ij} : i \in [N], j \in [K]\}$, and $\boldsymbol{\nu'} = \{\nu'_{ij} : i \in [N], j \in [K]\}$, fix some policy $\pi$ and let $\mathbb{P}_{\boldsymbol{\nu},\pi}$ and $\mathbb{P}_{\boldsymbol{\nu'},\pi}$ be the probability measures induced by the $T$-round interconnection of $\pi$ and $\boldsymbol{\nu}$ (respectively, $\pi$ and $\boldsymbol{\nu'}$), the following divergence decomposition holds,*

$$D\left(\mathbb{P}_{\boldsymbol{\nu},\pi}, \mathbb{P}_{\boldsymbol{\nu'},\pi}\right) = \sum_{i=1}^{N} \sum_{j=1}^{K} \mathbb{E}_{\boldsymbol{\nu},\pi}\left[N_{ij}(T)\right] \cdot D\left(\nu_{ij}, \nu'_{ij}\right).$$

**Lemma J.7** (Chain Rule of KL Divergence, Theorem 2.5.3 (Cover, 1999)). *Consider two random variables $X$ and $Y$ with two possible joint probability mass functions $p(x, y)$ and $q(x, y)$. The KL divergence between $p(x, y)$ and $q(x, y)$ is*

$$D\left(p(x,y)\|q(x,y)\right) = D\left(p(x)\|q(x)\right) + D\left(p(y|x)\|q(y|x)\right).$$

**Lemma J.8** (Bretagnolle-Huber Inequality, Theorem 14.2 (Lattimore & Szepesvári, 2020)). *Let $P$ and $Q$ be probability measures on the same measurable space $(\Omega, \mathcal{F})$, and let $A \in \mathcal{F}$ be an arbitrary event. Then,*

$$P(A) + Q(A^c) \geq \frac{1}{2}\exp\left(-D(P, Q)\right),$$

*where $A^c = \Omega \backslash A$ is the complement of $A$, $D(\cdot, \cdot)$ is the KL divergence.*

**Lemma J.9** (Multiplicative Chernoff Bound, Deviation Below the Mean, Theorem 4.5 (Mitzenmacher & Upfal, 2017)). *Let $X_1, \cdots, X_n$ be independent Bernoulli trials such that $\mathbb{P}(X_i) = p_i$. Let $X = \sum_{i=1}^{n} X_i$ and $\mu = \mathbb{E}[X]$. Then for any $0 < \rho < 1$, we have*

$$\mathbb{P}\left(X \leq (1 - \rho)\mu\right) \leq \exp\left(-\frac{\mu\rho^2}{2}\right).$$

**Lemma J.10** (Approximate Optimal Stable Share Guarantee of Algorithm 6, Theorem 5 (Lin et al., 2026)). *Given any utility matrix $\boldsymbol{U}$, parameter $m = \lfloor \log_2 N + 2 \rfloor$, and the instability tolerance $\varepsilon \geq 0$, Algorithm 6 computes a distribution $D \in \Delta(\mathcal{I})$, where $\mathcal{I}$ is the set of internally stable matchings, such that $\boldsymbol{U}_D(p) \geq \frac{\boldsymbol{U}^*_\varepsilon(p)}{m} - \varepsilon, \forall p \in \mathcal{N}$.*

