# OpenReview forum: "Adaptive Bandit Algorithms for Contextual Matching Markets"
_ICML.cc/2026/Conference — ICML 2026 regular_

### Official Review · Reviewer_z32p · 2026-02-25

**Soundness:** 3
**Presentation:** 2
**Significance:** 3
**Originality:** 3
**Overall Recommendation:** 5
**Confidence:** 2

**Summary:**

This paper studies online learning in contextual matching markets where the arm/job side has a fixed, known strict preference ranking over players, while each player’s preference over arms is induced by an unknown linear contextual utility observed through bandit feedback. The goal is to minimize each player’s regret relative to the player-optimal stable matching at each round. For stochastic contexts, the authors introduce a “minimum preference gap” notion and propose BARB, a fully adaptive batched regret-balancing algorithm that does not require prior knowledge of the gap (or context covariance structure), achieving an instance-dependent guarantee of order $\tilde{O}(\log^2T/\Delta_{\min}^2)$.

For adversarial contexts, they argue standard stable regret can become vacuous when utility differences are arbitrarily small, and propose a refined regret notion that interpolates between exact stability in large-gap rounds and $\varepsilon$-stable approximate benchmarks in small-gap rounds, leading to $\tilde{O}(T^{2/3})$-type guarantees under an offline approximation oracle. Overall, the work provides a principled treatment of stability-aware learning under both stochastic and adversarial contextual dynamics, with clear algorithmic ideas and accompanying theory.

**Compliance With Llm Reviewing Policy:**

Affirmed.

**Final Justification:**

This paper is technically solid and well-presented. I will keep my score.

**Key Questions For Authors:**

In the adversarial setting, the regret definition (Eq. 7) depends on the approximation factor $\alpha$, but the main upper bound in Theorem 4.1 does not show explicit dependence on it. Could the authors clarify how the choice/guarantee of $\alpha$ affects the interpretability/strength of $\mathrm{Reg}^{\alpha,\Delta}$ in practice?

**Limitations:**

Yes

**Strengths And Weaknesses:**

**Strengths**

1.  Compared to prior work that typically relies on a *static/uniform* preference-gap condition, this paper introduces a distributional “minimum preference gap” for stochastic contexts, which relaxes the worst-case requirement and better captures scenarios where small-gap contexts occur infrequently.
2. The algorithm BARB is fully adaptive and does not require any prior information about the gap $\Delta$.
3. The theoretical result is comprehensive, providing both instance-dependent and instance-independent regret bounds, accompanied by matching lower bounds under mild assumptions. It further extends the framework to adversarial contexts by proposing a tractable regret notion and the corresponding algorithm AdECO with worst-case guarantees.
4. Numerical results showed that their algorithms outperforms the baseline when eigenvalues $\lambda_{\min}$ is small.



**Weakness**

1. While the paper advocates a relaxed, distributional difficulty measure, Assumption 4 and 5 (eigenvalue lower bound for refined rates and the CDF condition near zero to obtain $\widetilde O(T^{2/3})$ remain somewhat analysis-driven and lack actionable guidance on how to interpret/verify them in practice.
2. The experiment is run over small scales with $N=K=4$. What's the algorithm performance in larger scale settings?

---

> ### Author Rebuttal · Authors · 2026-03-31
>
> We thank reviewer z32p for the valuable comments. We provide our response point by point below.
>
> ### **R1.Interpretability of Technical Assumptions**
> Assumption 4 ensures that, under the distribution $\mathcal{D}_x$, the data spans all dimensions of the feature space, thereby avoiding multicollinearity or near-degeneracy. We acknowledged that this assumption is somewhat unrealistic.
> However, the main result of [1] relies on this assumption and requires knowledge of the constant (the eigenvalue lower bound) for algorithm implementation. Our key point is that our gap definition and algorithm adapt to this assumption without needing to know this constant.
>
> Assumption 5 requires that the probability that the minimum utility difference falls below $t$ grows at most linearly in $t$, ensuring that we do not have excessively high probability for small gaps -- a regime that would force linear regret. Verification of this assumption is natural within the algorithm implementation. BARB maintains a candidate gap and a counter to track rounds with gaps smaller than this candidate during the exploration phase of each batch, shrinking the candidate gap once the counter reaches a threshold. Thus, once BARB stops exploring and keeps exploiting in a batch, we can infer with reasonably high probability that the minimum preference gap is large and Assumption 5 holds.
> This can also be used to verify Assumption 4 during the implementation of Batched-ETC: when the algorithm stops exploration and keeps exploitation in a batch, it implies that, with high probability, $\theta$ is well-estimated in every direction under the data distribution, which further indicates that the contexts have provided sufficient information to cover the feature space.
>
> To further aid interpretability and illustrate the role of the minimum preference gap, we will include an illustrative example in the final version (see [anonymous visualization](https://ibb.co/RkWR9W7S)).
> In this example, there is one player and three arms, with preference parameter $\theta = 1$ and context distributions for the three arms given by $U[0, 0.5]$, $U[0.25, 0.75]$, and $U[0.5, 1]$. The CDF of the minimum utility difference and the preference gap could be visualized from the figure.
>
> ### **R2.Scalability of the Experiments**
> We conducted additional experiments and will include the results in the final version. First, we vary the market size $N = K \in \{3, 6, 9, 12\}$ and observe that the maximum cumulative regret over players of BARB increases as the market size grows, which is consistent with our theoretical results (see [anonymous visualization](https://ibb.co/pS1dyPn)). Second, we compare the performance of BARB, ETC, and Batched-ETC in a larger market with $25$ players using orthonormal contexts. In this setting, BARB achieves smallest maximum cumulative regret, followed by ETC, and Batched-ETC performs the worst, which is aligned with the conclusion in the small market (see [anonymous visualization](https://ibb.co/jZTLrvLf)).
>
> ### **R3.Absence of $\alpha$ in the Regret Upper Bound in Theorem 4.1**
> The regret analysis in Theorem 4.1 decomposes into two parts: exploration and exploitation. In the exploration phase, when $\delta_{\min}$ is small, the benchmark is $\alpha U_i^{\varepsilon}$, and the regret scales linearly with $\alpha$ -- each exploration round incurs at most $\alpha B_y$ approximation regret, where $B_y$ is a uniform bound on the observed reward. Since $\alpha \leq 1$, we directly adopt the looser upper bound $B_y$. In the exploitation phase, the approximation oracle's guarantee (Definition 4) ensures that the per-round loss in the small-gap regime is bounded by $(\Delta + \epsilon)/2$, which is independent of $\alpha$.
>
> ---
>
> [1] Dynamic Matching Bandit for Two-Sided Online Markets. Li, Y., Wang, C.-h., Cheng, G., and Sun, W. W., 2022.

---

> > ### Author Rebuttal · Reviewer_z32p · 2026-04-02
> >
> > We thank the authors for the detailed response. I would like to keep my score. Good luck!

---

### Official Review · Reviewer_w8Cm · 2026-03-11

**Soundness:** 3
**Presentation:** 3
**Significance:** 2
**Originality:** 2
**Overall Recommendation:** 4
**Confidence:** 3

**Summary:**

This paper studies contextual bandit learning in two-sided matching markets where players are matched to arms under stability constraints. The setting assumes that arm preferences over players are known and fixed, while player preferences over arms are modeled through a linear contextual utility function. The learning objective is to minimize the player-optimal stable regret, defined relative to the stable matching benchmark induced by the true utilities.
The paper considers two scenarios for context generation: stochastic contexts drawn from an unknown distribution and adversarial contexts that may be arbitrarily chosen. For the stochastic case, the authors introduce a minimum preference gap notion that probabilistically characterizes the difficulty of distinguishing utilities under contextual variation. They propose a batched adaptive exploration-exploitation procedure that estimates player preferences and performs stable matching when confidence intervals are sufficiently separated. Theoretical analysis shows instance-dependent improved regret bounds with improved bounds under additional covariance assumptions.

For adversarial contexts, the paper introduces a relaxed regret notion based on approximate stability when utility gaps become arbitrarily small. The proposed AdECO algorithm combines adaptive exploration with either Gale-Shapley matching or an approximation oracle depending on the uncertainty in the utility estimates. Experiments on synthetic matching markets illustrate the empirical behavior of the algorithms and compare them with baseline explore-then-commit strategies.

**Compliance With Llm Reviewing Policy:**

Affirmed.

**Final Justification:**

The rebuttal improves my understanding of several aspects of the paper, particularly regarding the lower bound and the coordination challenges in multi-player settings. I upgrade my score from Weak Reject to Weak Accept.

**Key Questions For Authors:**

How realistic is the assumption that arm preferences over players are fixed and known? Could the framework be extended to settings where both sides learn preferences?

**Limitations:**

The limitations discussion seems insufficient. The proposed framework assumes linear contextual utilities and bounded noise, which may not capture complex preferences in real matching markets. Additionally, the requirement that arm-side preferences are fixed and known simplifies the problem but limits applicability in environments where both sides have uncertain or evolving preferences. The algorithms also rely on theoretical assumptions about context distributions or adversarial thresholds that may not hold in practice. Finally, the empirical validation is limited to small synthetic experiments, leaving open questions about scalability and robustness in large-scale markets.

**Strengths And Weaknesses:**

**Strength:**

The paper studies contextual bandits in two-sided matching markets with stability constraints, which is a challenging and relatively underexplored intersection of bandits and market design
The paper proposes algorithms and regret guarantees for both settings, which broadens the scope of existing work that often focuses only on stochastic models
The introduction of the minimum preference gap provides a mechanism to characterize learning difficulty in contextual matching markets and enables regret bounds that scale with problem hardness

**Weaknesses:**

Many algorithmic components (ridge regression confidence sets, elliptical potential arguments, exploration thresholds) are standard in linear contextual bandits. The main novelty lies in adapting them to the matching-market setting rather than introducing fundamentally new learning principles
The model assumes known and fixed arm-side preferences over players, which may be unrealistic in many real markets where both sides learn preferences

The regret definitions, particularly in the adversarial setting with approximate stability and oracle access, become somewhat convoluted and may reduce interpretability.

Certain parts of the paper (particularly regret definitions and algorithm descriptions) are dense and require careful reading to follow.

---

> ### Author Rebuttal · Authors · 2026-03-31
>
> We thank reviewer w8Cm for the valuable comments. We provide our response point by point below.
>
> ### **R1.Novelty of the Algorithmic Components**
> We agree that some of the building blocks we employ (ridge regression and elliptical potentials) are commonly used in contextual bandits literature. However, adapting them to the matching market is non-trivial.
>
> First, stable regret is defined for each player individually; achieving sublinear regret for all players poses a multi-objective problem for the platform, which significantly increases the difficulty compared to the single-player contextual bandit problem.
> The difficulty is especially evident when the minimum utility difference is small or approaches zero, as a small estimation error by one player can propagate to others, resulting in substantial loss. This phenomenon is absent in standard contextual bandits, requiring more careful management of the exploration-exploitation trade-off. This motivates our batch-based design of shrinking the candidate gap to adaptively search for the correct minimum preference gap.
>
> Second, in our setting, the algorithm must select a matching that coordinates the actions of different players, meaning that player-arm pairs cannot be explored independently. Our use of maximum cardinality matching enables efficient coordination of exploration.
>
> The main novelty of our paper lies in introducing the probabilistic minimum preference gap and the adversarial regret definition, which together provide a proper metric for quantifying problem difficulty. Moreover, our $\Omega(T^{2/3})$ lower bound is specific to matching markets and confirms the tightness of our analysis -- a result that has no analogue in standard contextual bandits.
>
> ### **R2.Fixed and Known Arm-side Preferences over Players**
> We acknowledge that assuming fixed, known arm-side preferences is a simplification, though this is standard in the literature [1-5] and realistic in several practical domains. For instance, in online labor markets, employers' preferences over employees are learned, while employees' preferences are based on known utilities such as wages. Similarly, in ride-hailing, customers rank drivers by proximity and rating, whereas drivers' preferences over trips are learned.
>
> Nevertheless, our framework can be extended to the case where both sides learn, but this introduces additional complexity. First, due to the lattice structure of stable matchings, a player-optimal matching is arm-pessimal, suggesting a benchmark that considers the optimal stable matching for players and the pessimal one for arms [6]. Second, we need to impose a utility model for the arm side, with contexts observed for both sides each round. Exploration would then be triggered whenever any player-arm or arm-player pair has $||x||_{V^{-1}}$ exceeding a threshold, ensuring exploitation occurs only when all participants have sufficiently accurate preference estimates.
>
> ### **R3.Interpretability of the Approximation Regret**
> Please see R2 to Reviewer VCyL.
>
> ### **R4.Discussion of Limitations**
> Thank you for your suggestions. We will include thorough discussions of the limitations in the final version.
>
> Regarding the theoretical assumptions, we would like to emphasize that, compared to existing work on contextual matching markets [4, 5], our framework requires minimal assumptions on the context distribution. Specifically, BARB only requires that contexts are generated stochastically, whereas the aformentioned references impose additional spectral assumptions (i.e., a minimum eigenvalue condition).
> For more details, please also see R1 to Reviewer z32p.
>
> For AdECO in the adversarial context setting, the threshold $\Delta$ is an input to the algorithm rather than an assumption. Theorem 4.1 guarantees that for any given $\Delta$, AdECO achieves the corresponding regret bound $Reg^{\alpha, \Delta}$.
> For empirical validation, we will include additional experimental results on the scalability of our algorithm and its performance comparison with baselines in larger markets in the final version (please see R2 to Reviewer z32p).
>
> ---
>
> [1] Competing Bandits in Matching Markets. Liu et al., 2020.
>
> [2] Player-optimal Stable Regret for Bandit Learning in Matching Markets. Kong and Li, 2023.
>
> [3] Stable Matching with Ties: Approximation Ratios and Learning. Lin et al., 2025
>
> [4] Dynamic Matching Bandit for Two-Sided Online Markets. Li et al., 2022.
>
> [5] Compteing Bandits in Decentralized Large Contextual Matching Markets. Parikh et al., 2024.
>
> [6] Decentralized Two-sided Bandit Learning in Matching Market. Zhang and Fang, 2024.

---

> > ### Author Rebuttal · Reviewer_w8Cm · 2026-04-03
> >
> > Thank you to the authors for their detailed and timely rebuttal. I have carefully read their responses and provide my updated assessment below.
> >
> > **On Algorithmic Novelty (R1)**
> > I appreciate the clarification, and I agree these contributions have value. That said, my concern about the incremental nature of the algorithmic contribution is not fully resolved.
> >
> > **On Fixed and Known Arm-Side Preferences (R2)**
> > The authors acknowledge this limitation and provide motivating examples (online labor markets, ride-hailing) where the assumption is realistic. They also sketch a possible extension to the bilateral learning setting. I find the examples convincing.
> >
> > **On the Interpretability of the Approximation Regret (R3)**
> > The authors clarify that $\Delta$ is an algorithmic input rather than an environmental assumption, and that the regret bound holds for any fixed $\Delta$. This partially addresses my concern. However, the fact that the benchmark switches between $U_i$(t) and $αU^\varepsilon_i(t)$ depending on $\delta_{min}(t)$ round by round still makes the regret definition somewhat difficult to interpret in practice.
> >
> > **On the Discussion of Limitations and Empirical Validation (R4)**
> > I appreciate the authors' commitment to expanding the limitations section and providing additional experiments in the final version.
> >
> > **Summary**
> > The rebuttal improves my understanding of several aspects of the paper, particularly regarding the lower bound and the coordination challenges in multi-player settings. I upgrade my score from Weak Reject to Weak Accept.

---

> > > ### Author Response · Authors · 2026-04-05
> > >
> > > We thank Reviewer w8Cm for their careful reading and positive feedback on our rebuttal.
> > >
> > > Regarding the novelty of the algorithm components, we would like to further clarify that our confidence interval (CI) overlap counter and the multiplicative Chernoff bound-calibrated threshold for batch transitions in BARB (for the stochastic context case) are novel compared to other bandit algorithms in the literature.
> > >
> > > Previous work uses CI overlaps at a per-round level either for action elimination or for one-shot exploration–exploitation switching. For example, [1] eliminates suboptimal arms when their CIs are disjoint from the arm with the highest estimated utility; [2] switches to and remains in the exploitation phase once all CIs are disjoint, such that the correct preference ranking can be inferred with high probability. In contrast, we aggregate these overlaps to construct a global statistical test statistic. By tracking the frequency of overlaps during exploitation rounds and comparing it to a concentration-calibrated threshold derived from the multiplicative Chernoff bound,
> > > we can determine with high probability whether the current candidate gap is smaller than the environment's minimum preference gap. Consequently, our algorithm transitions to the next batch in a data-driven manner, which differs from the doubling trick for predefined time horizons used in prior work [3]. The batch design also allows us to adaptively estimate the true minimum preference gap on the fly, yielding an algorithm that organically adapts to instance-specific complexity without requiring the minimum preference gap as an input.
> > >
> > > [1] Action Elimination and Stopping Conditions for the Multi-Armed Bandit and Reinforcement Learning Problems, Even-Dar et al., 2006.
> > >
> > > [2] Player-optimal Stable Regret for Bandit Learning in Matching Markets, Kong & Li, 2023.
> > >
> > > [3] Gambling in a Rigged Casino: The Adversarial Multi-Armed Bandit Problem, Auer et al., 1995.

---

### Official Review · Reviewer_B1xT · 2026-03-12

**Soundness:** 3
**Presentation:** 3
**Significance:** 3
**Originality:** 3
**Overall Recommendation:** 5
**Confidence:** 4

**Summary:**

This paper investigates bandit learning in two-sided matching markets where players' unknown preferences over arms are modeled as linear functions of time-varying contexts. To address the fragility of stable matchings—where minor context shifts can drastically alter the optimal assignments and cause regret spikes—the authors analyze both stochastic and adversarial context environments. They introduce a novel probabilistic "minimum preference gap" for stochastic settings and a tractable $\alpha$-approximate regret metric for adversarial settings, allowing them to bypass traditional impossibility results in markets with vanishing utility gaps. Finally, they propose two fully adaptive algorithms (BARB and AdECO) that dynamically alternate between exploration and exploitation, achieving instance-dependent poly-logarithmic regret bounds and a tight $\mathcal{O}(T^{2/3})$ instance-independent bound without requiring prior knowledge of the environment's hardness.

**Compliance With Llm Reviewing Policy:**

Affirmed.

**Final Justification:**

I thank the authors for the insightful rebuttal and comments. All my concerns are addressed including the lack of decentralization (in the followup comment). I will increase my score for this paper.

**Key Questions For Authors:**

Please place the paper properly in the literature by comparing pros and cons of closely related works.

**Limitations:**

This is a theoretical work with hard to anticipate societal impact.

**Strengths And Weaknesses:**

Pros:
- The authors study a challenging problem of centralized linear contextual two-sided matching market.  The problem is studied under both stochastic and adversarial contexts, and regret bounds are developed.
-  The authors provide "Minimum Preference Gap" condition which provide $log^2(T)$ regret in stochastic contexts setting. They further provide conditions on gap CDF that enables $O(T^{2/3})$ regret.
- The authors also study adversarial context setting, and provide an $\alpha$-approximate regret guarantee that scales as $O(log^2(T)/(Delta- \epsilon) + (Delta + \epsilon) T)$

Cons:
- Assumption 2 implies zero mean Gaussian noise is not supported. Why do we need this restriction while standard linear contextual bandits can support Gaussian noise? This seems to narrow the applicability of this work.
- In Theorem 4.1, there is no $\alpha$ dependence. Is this omission expected?
- Abrupt shifts in contexts may change the stable matching which can only be learnt in multiple rounds in a decentralized setting, while the system may keep changing between rounds. Can we extend the current results to the decentralized setting?
- The comparison with Parikh et al. is lacking. The authors should compare the gap condition introduced in the current paper with the gap condition introduced in the paper by Parikh et al. The latter seems to be more general. More importantly the challenges of decentralized two-sided markets is harder (the previous point) which is not emphasized in this paper.

Comments/Questions:
- Are the regret bounds independent of the number of arms $K$?
- Are there existing algorithms that can be used for $\alpha$-approximation oracles for certain values of $\alpha$?
- One recent work "Competing Bandits in Matching Markets via Super Stability" Basu et al. proposes a decentralization scheme for a centralized algorithm for the non-contextual two-sided matching market. Can we adapt ideas from there for the current paper for decentralization? Just curious.

---

> ### Author Rebuttal · Authors · 2026-03-31
>
> We thank reviewer B1xT for the valuable comments. We provide our response point by point below.
> ### **R1.Bounded Noise Assumption**
> We adopted the bounded noise assumption for simplicity, as it allows directly bounding the observed reward and per-round regret by $B_y$, simplifying the handling of bad events and exploration. However, a closer look at the definition reveals that the expected per-round regret is $\mathbb{E}[U_i(t) - y_i(t)] = U_i(t) - U_{i,\mu_t(i)}(t)$, where both terms lie in $[-B_\theta B_x, B_\theta B_x]$ (by Assumptions 1 and 3). Hence, the expected per-round regret is bounded by $2B_\theta B_x$, and the cumulative expected regret from exploration rounds is at most $2B_\theta B_x |\mathcal{G}_k|$, removing the need for an explicit noise bound.
> Moreover, the confidence set for $\hat{\theta}$ can be adapted to sub-Gaussian noise by replacing $B_y$ with the sub-Gaussian parameter [1]. Therefore, extending the results to sub-Gaussian noise only requires adjusting the per-round regret constant and the confidence radius; the rest of the analysis -- including the elliptical potential lemma, exploration threshold, and regret decomposition -- remains unchanged. Neither the algorithm structure nor the regret order is affected. We thank the reviewer for this observation and will revise the paper accordingly.
> ### **R2.Absence of $\alpha$ in the Regret Upper Bound in Theorem 4.1**
> Please see R3 to Reviewer z32p.
>
> ### **R3.Extend the Current Results to Decentralized Setting**
> Our current results do not directly extend to the decentralized setting, as we consider markets where preferences change each round and the benchmark is round-specific -- determined by the current preference structure. In a decentralized setting, a player can infer her own preference profile through accurate estimation, but without knowledge of other players' preferences, she cannot derive the player-optimal stable matching for the current round. [2] addresses the static market setting, where the benchmark is fixed, enabling convergence via shared communication flags. In the dynamic case, such decomposition appears infeasible. [3] studies decentralized contextual matching markets, but it imposes stringent assumptions: the number of possible preference structures is finite and known, and each player's preference profile is distinct across environments to enable identification. In contrast, we allow for an unknown number of environments and arbitrary preference structures at any round -- the total number of possible environments is exponential in the number of players and arms. This makes designing decentralized algorithms a significant challenge, and whether it is possible and the minimum communication required remain open.
>
> ### **R4.Gap Condition Compared to Parikh et al. (2024)**
> The gap conditions in [3] are deterministic lower bounds: $\Delta_{min}$ is the smallest preference difference for every player in every round, $\Delta_{minrank}$ is the smallest gap among arms whose relative order changes between the current environment and any other environment, and $\Delta_p$ ensures that a ``large'' gap appears at least every $P_e$ rounds. For any finite horizon $T$, these deterministic constraints restrict the environment's behavior, precluding worst-case realizations altogether. In contrast, our probabilistic minimum preference gap captures the intrinsic difficulty of the distribution, allows for occasional ties, and yields stronger guarantees when worst case scenarios occur with small probability. We will add a clear discussion of these differences in the final version.
>
> ### **R5.Regret Bound with Respect to the Number of Arms $K$**
> Our current regret bound depends on $K$ (see Eq. (20) in Appendix B), but this dependence arises solely from the union bound applied to the bad event. Upon closer inspection, this dependence can be removed entirely. Specifically, we only exploit when the contexts from all arms satisfy $||x_j(t)||_{(V_i^{(t)})^{-1}} < \Delta_k / \eta$, which allows us to drop the union bound over arms. Meanwhile, the terms involving the time horizon $T$ do not depend on $K$. This is because each arm provides a context, enabling us to learn the utilities for all arms jointly per player, thereby replacing the typical dependence on $K$ in stochastic bandits with dependence on $d$, the context dimension. We thank the reviewer for pointing this out and will make the corresponding revision in the final version.
>
> ### **R6.Existing Approximation Oracle**
> [4] proposed an approximation oracle with approximation ratio $1/\lfloor\log_2 N+2\rfloor$, which we restated in Algorithm 6 in Appendix F.
>
> ---
>
> [1] Improved Algorithms for Linear Stochastic Bandits, Abbasi-Yadkori et al., 2011.
>
> [2] Competing Bandits in Matching Markets via Super Stability, Basu, 2025.
>
> [3] Competing Bandits in Decentralized Contextual Matching Markets, Parikh et al., 2024.
>
> [4] Stable Matching with Ties: Approximation Ratios and Learning, Lin et al., 2025

---

> > ### Author Rebuttal · Reviewer_B1xT · 2026-04-03
> >
> > I thank the authors for a detailed response. All my concerns about the work are resolved except the lack of a clear pathway to decentralized learning in this setup. I acknowledge that the system is challenging, and is not easily amenable to decentralized learning. I appreciate the authors adding their thought around this in the paper, however I will maintain my score.

---

> > > ### Author Response · Authors · 2026-04-08
> > >
> > > We thank Reviewer B1xT for their careful reading of our rebuttal.
> > >
> > > To enable decentralization within our current algorithmic framework, we introduce the following assumption for environment identification:
> > >
> > > **Assumption 1**. Let $\mathcal{E}$ denote the set of environments. For any two distinct environments $e, e' \in \mathcal{E}$, the top-$N$ ranked arms for each player $p_i$ are distinct; that is, the ranking vectors $\nu_i^e$ and $\nu_i^{e'}$ differ in at least one of the first $N$ entries.
> > >
> > > Here, an environment is defined by the preference ranking: if two contexts yield the same preference ranking, they are considered the same environment.  This assumption ensures that environments can be distinguished using only the preference rankings of each individual player, without requiring direct communication. Notably, this assumption is also used in [1], but our requirement is weaker: [1] requires each player to know the exact number of possible environments, whereas we allow that number to be unknown. Moreover, [1] demonstrates the necessity of this assumption by providing an example in which sublinear regret cannot be achieved without it.
> > >
> > > Additionally, we assume that every player $p_i$ maintains two flags that can be shared within the market.
> > > - $F_i$ indicates whether the Mahalanobis norm $||x_j||_{V_i^{-1}}$ for any arm $a_j$ is smaller than a given threshold.
> > > - $F_i'$ indicates whether the constructed confidence intervals (CIs) are disjoint in the current exploitation round.
> > >
> > > For simplicity, we discuss only the case with stochastic contexts. When multiple players propose to the same arm in a round, only the player with the highest ranking in that arm's preference profile receives a reward; the others receive zero.
> > >
> > > The algorithm is sketched as follows from player $p_i$'s perspective.
> > >
> > > The candidate gap $\Delta_k$ is public; thus $\xi_k$ can be inferred locally. At round $t_k$, after observing the contexts for every arm $a_j$, $p_i$ computes $||x_j(t_k)||_{V_i^{-1}}$ and compares it to $\xi_k$. If there exists an $x_j$ such that the Mahalanobis norm exceeds the threshold, $F_i$ is set to *True*.
> > >
> > > 1. **If there exists any player with $F_i = \text{True}$**, the system explores. Player $p_i$ proposes to the arm with the largest Mahalanobis norm among those exceeding $\xi_k$. If no such arm exists, $p_i$ proposes nothing in that round. The updates for the Gram matrix $V_i$ and the preference parameter estimate $\hat{\theta}_i$ follow Eq. (5) in the paper.
> > >
> > > 2. **If every player has $F_i = \text{False}$**, the system exploits. Player $p_i$ first computes the estimated utility and constructs confidence intervals for every arm. If all CIs are disjoint, set $F_i' = \text{False}$; otherwise, set $F_i' = \text{True}$.
> > >
> > >    - **(a)** If there exists any player with $F_i' = \text{True}$, the counter $N_{i,k} \leftarrow N_{i,k} + 1$. This counter is maintained locally but takes the same value across all players. The player proposes to an arbitrary arm. If $N_{i,k}$ exceeds the threshold defined in BARB, the algorithm moves to the next batch, and $\Delta_k$ is shrunk by a factor of $\sqrt{2}$.
> > >
> > >    - **(b)** If all players have $F_i' = \text{False}$, the player identifies the environment from the current estimated utilities and proposes to the highest-ranked arm that has not yet rejected $p_i$ in that environment.
> > >
> > > With the above implementation, the exploration regret remains the same as in BARB, while the exploitation regret incurs an additional term of $N^2 E$ compared to BARB, where $E$ is the number of environments that occur during the horizon. Note that the algorithm does not require knowledge of $E$; this quantity is used only for theoretical analysis. Moreover, this additional term is independent of $T$, and therefore the order of the regret bound remains unchanged.
> > >
> > > We hope this clarification addresses the reviewer's concern.
> > >
> > > [1] Parikh et al., "Competing Bandits in Decentralized Contextual Matching Markets," 2024.

---

### Official Review · Reviewer_VCyL · 2026-03-15

**Soundness:** 2
**Presentation:** 2
**Significance:** 2
**Originality:** 2
**Overall Recommendation:** 5
**Confidence:** 4

**Summary:**

The paper addresses bandit learning in two-sided matching markets where player utilities follow a linear contextual model. In each round, arms arrive with observable feature vectors (context), and the platform must match arms to players while minimizing each player's regret against a stable matching benchmark (player optimal regret). The authors obtain results for both stochastic and adversarial context settings,— BARB (Batched Adaptive Regret-Balancing) for stochastic contexts and AdECO (Adaptive Explore-Choose Oracle) for adversarial contexts. The authors proceed to explore a general aspect of instance-dependent poly-logarithmic regret bounds, and prove tight $T^{2/3}$ instance-independent bounds for the stochastic setting.

**Compliance With Llm Reviewing Policy:**

Affirmed.

**Final Justification:**

Good Job on the decentralized regret. Please add this to the main paper. I increase my score to 5.

**Key Questions For Authors:**

See weaknesses

**Limitations:**

yes

**Strengths And Weaknesses:**

Strengths:

1) Algorithmic Contributions. The paper delivers meaningful algorithms for matching markets. The novel minimum preference gap definition is a probabilistic relaxation of prior deterministic gap assumptions. The matching upper and lower bounds of $T^{2/3}$ are a novel contribution, as prior work on static preferences could only achieve instance-independent guarantees.

Both BARB and AdECO are well-motivated. The adaptive phase transition between exploration and exploitation is elegant, and the batched structure of BARB allows operation without prior knowledge of the gap or covariance structure .

2) Empirical Validation. The experiments are reasonably designed. The comparison against ETC and Batched-ETC across two covariance regimes effectively illustrates BARB's robustness advantage, and the AdECO results in the adversarial setting are encouraging.

Weaknesses

1) Centralization Assumption. The main criticism of the proposed algorithm is the availability of centralized platform with full visibility over all player-arm utilities and contexts. Although resent works have (like Parikh et al) studied the decentralized version, it is not clear why authors choose to study this framework. Also it is not practical.

2) New Regret Definition and Approximation Oracle: The AdECO algorithm relies on access to an offline approximation oracle used in one recent work. More discussions are required on the practical aspect (computational complexity) of such an oracle. Also motivation is needed for the new regret definition for the adversarial setting.

3) Regret guarantee for adversarial setting: It seems that if the gap is constant,  AdECO incurs linear regret, which is quite different from previous works. Either motivate why is this the case or provide some hard instances where this is unavoidable.

4) Experiment Scope. Experiments are limited to $N=K=4$ with $T=200k$. Real-world matching markets are often considerably larger, and scalability results are absent. The adversarial experiment also lacks a direct regret comparison, instead showing cumulative rewards relative to an oracle.

5) Presentation:  Some key definitions (particularly Definition 2 for the minimum preference gap) require considerable unpacking and could benefit from more intuitive exposition. The connection between the stochastic and adversarial settings in Remark 4.1 is somewhat informal.

6) Missing Baselines. The paper lacks comparison against the contextual matching work of Li et al. (2022) in the main experiments, which is arguably the closest prior work.

---

> ### Author Rebuttal · Authors · 2026-03-31
>
> We thank reviewer VCyL for the valuable comments. We provide our response point by point below.
>
> ### **R1.Centralized Assumption**
> While we agree that decentralized settings are practically relevant, we argue that the centralized assumption is both a natural starting point and a realistic model for real-world applications. For instance, in ride-hailing, the platform centrally matches drivers with customers using real-time contextual data (location, time, traffic) and can learn their preferences over time. Studying the centralized setting allows us to isolate the core challenge of minimizing stable regret adaptively without the additional complexity of communication and coordination overhead.
>
> For the comparison with [1], please see R3 to Reviewer B1xT.
>
> ### **R2.Motivation for the Adversarial Regret Definition and the Computational Complexity of the Approximation Oracle**
> When contexts are generated adversarially, we may encounter infinitely many rounds with arbitrarily small preference gaps, making meaningful regret guarantees impossible under the standard stable regret definition. This issue also appears in static settings when ground-truth utilities exhibit small gaps [3]. Consider a static market with 2 players and 2 arms, where both arms prefer $p_1$ to $p_2$, with utilities $[1, 1+\Delta]$ for $p_1$ and $[1, 0]$ for $p_2$. The stable matching benchmark assigns $p_1$ to $a_2$ and $p_2$ to $a_1$, yielding utilities $1+\Delta$ and $1$, respectively. Correctly ranking the arms for $p_1$ requires $\Omega(1/\Delta^2)$ exploration rounds, which incurs $\Omega(1/\Delta^2)$ regret for $p_2$; when $\Delta = O(1/\sqrt{T})$, $p_2$ suffers linear regret. This example extends to contextual bandits, where contexts can induce arbitrarily small $\Delta$, demonstrating that adversarial contexts necessitate an appropriate regret definition that clips small gaps.
> In combinatorial bandits, approximation regret typically addresses computational limitations. Here, however, the approximation ratio handles a statistical limitation, and can also be interpreted through fairness: when it is impossible to guarantee every player her optimal stable share in a market with small gaps, we aim to ensure each player receives at least an $\alpha$-fraction of the optimal stable share (OSS).
>
> Regarding computational complexity, the offline approximation oracle in [3] replicates each arm $\lfloor \log_2 N + 2 \rfloor$ times, assigns utilities appropriately, and invokes Gale–Shapley, yielding $O(NK \log_2 N)$.
>
> ### **R3.Regret Guarantee for Adversarial Setting**
> The gap $\Delta$ is an algorithmic input rather than an environment-dependent quantity. Our regret bound holds uniformly for any choice of $\Delta$, so in practice one would not fix it to a constant.
> Furthermore, the term $\frac{\Delta + \varepsilon}{2}T$ captures the exploitation regret. Under the good concentration event, no regret is incurred when using the Gale–Shapley oracle; regret arises only in rounds where the approximation oracle is invoked, with benchmark $\alpha U_i^{\varepsilon}(t)$. When $\Delta$ and $\varepsilon$ are large, $U_i^{\varepsilon}(t)$ can be large and harder to approach, leading to increased exploitation regret.
>
> ### **R4.Experiment Scope**
> Regarding scalability results and comparison in larger markets, please see R2 to Reviewer z32p.
>
> In the adversarial case, computing the full set of $\varepsilon$-optimal stable matchings is NP-hard, making it computationally impossible to retrieve the $\varepsilon$-optimal stable share -- the benchmark in the small-gap regime. Therefore, we instead compare the cumulative reward of the proposed algorithm against that of an oracle with known utilities.
>
> ### **R5.Presentation**
> Thanks for your suggestion. We will include an illustrative example on the minimum preference gap in the final version (please see R1 to Reviewer z32p), and we will also further clarify the connection between the stochastic and the adversarial setting.
>
> ### **R6.Lacking Comparison against Li et al. (2022)**
> As it is unclear whether the comment refers to the stochastic or adversarial setting, we address both cases.
>
> In the stochastic setting, we indeed compared with the algorithm of [2], whose two-phase design—exploration followed by exploitation—aligns with the ETC baseline.
>
> In the adversarial setting, ETC’s front-loaded exploration makes it inherently fragile: an adversary can restrict early contexts to a narrow region and later shift to unexplored areas, leading to poor parameter estimation and large losses. Consequently, ETC is not an appropriate baseline here. As there is no prior work on adversarial contextual matching markets, we focus on evaluating our proposed algorithm.
>
> ---
>
> [1] Competing Bandits in Decentralized Contextual Matching Markets, Parikh et al., 2024.
>
> [2] Dynamic Matching Bandit for Two-Sided Online Markets. Li et al., 2022.
>
> [3] Stable Matching with Ties: Approximation Ratios and Learning. Lin et al., 2025

---

> > ### Author Rebuttal · Reviewer_VCyL · 2026-04-04
> >
> > The authors answered some of my concerns. I still have concerns about the centralized nature of the framework, which I believe should be addressed, given the state of the art papers in this field are all decentralized.

---

> > > ### Author Response · Authors · 2026-04-08
> > >
> > > We thank Reviewer VCyL for their careful reading of our rebuttal.
> > >
> > > To enable decentralization within our current algorithmic framework, we introduce the following assumption for environment identification:
> > >
> > > **Assumption 1**. Let $\mathcal{E}$ denote the set of environments. For any two distinct environments $e, e' \in \mathcal{E}$, the top-$N$ ranked arms for each player $p_i$ are distinct; that is, the ranking vectors $\nu_i^e$ and $\nu_i^{e'}$ differ in at least one of the first $N$ entries.
> > >
> > > Here, an environment is defined by the preference ranking: if two contexts yield the same preference ranking, they are considered the same environment.  This assumption ensures that environments can be distinguished using only the preference rankings of each individual player, without requiring direct communication. Notably, this assumption is also used in [1], but our requirement is weaker: [1] requires each player to know the exact number of possible environments, whereas we allow that number to be unknown. Moreover, [1] demonstrates the necessity of this assumption by providing an example in which sublinear regret cannot be achieved without it.
> > >
> > > Additionally, we assume that every player $p_i$ maintains two flags that can be shared within the market.
> > > - $F_i$ indicates whether the Mahalanobis norm $||x_j||_{V_i^{-1}}$ for any arm $a_j$ is smaller than a given threshold.
> > > - $F_i'$ indicates whether the constructed confidence intervals (CIs) are disjoint in the current exploitation round.
> > >
> > > For simplicity, we discuss only the case with stochastic contexts. When multiple players propose to the same arm in a round, only the player with the highest ranking in that arm's preference profile receives a reward; the others receive zero.
> > >
> > > The algorithm is sketched as follows from player $p_i$'s perspective.
> > >
> > > The candidate gap $\Delta_k$ is public; thus $\xi_k$ can be inferred locally. At round $t_k$, after observing the contexts for every arm $a_j$, $p_i$ computes $||x_j(t_k)||_{V_i^{-1}}$ and compares it to $\xi_k$. If there exists an $x_j$ such that the Mahalanobis norm exceeds the threshold, $F_i$ is set to *True*.
> > >
> > > 1. **If there exists any player with $F_i = \text{True}$**, the system explores. Player $p_i$ proposes to the arm with the largest Mahalanobis norm among those exceeding $\xi_k$. If no such arm exists, $p_i$ proposes nothing in that round. The updates for the Gram matrix $V_i$ and the preference parameter estimate $\hat{\theta}_i$ follow Eq. (5) in the paper.
> > >
> > > 2. **If every player has $F_i = \text{False}$**, the system exploits. Player $p_i$ first computes the estimated utility and constructs confidence intervals for every arm. If all CIs are disjoint, set $F_i' = \text{False}$; otherwise, set $F_i' = \text{True}$.
> > >
> > >    - **(a)** If there exists any player with $F_i' = \text{True}$, the counter $N_{i,k} \leftarrow N_{i,k} + 1$. This counter is maintained locally but takes the same value across all players. The player proposes to an arbitrary arm. If $N_{i,k}$ exceeds the threshold defined in BARB, the algorithm moves to the next batch, and $\Delta_k$ is shrunk by a factor of $\sqrt{2}$.
> > >
> > >    - **(b)** If all players have $F_i' = \text{False}$, the player identifies the environment from the current estimated utilities and proposes to the highest-ranked arm that has not yet rejected $p_i$ in that environment.
> > >
> > > With the above implementation, the exploration regret remains the same as in BARB, while the exploitation regret incurs an additional term of $N^2 E$ compared to BARB, where $E$ is the number of environments that occur during the horizon. Note that the algorithm does not require knowledge of $E$; this quantity is used only for theoretical analysis. Moreover, this additional term is independent of $T$, and therefore the order of the regret bound remains unchanged.
> > >
> > > We hope this clarification addresses the reviewer's concern.
> > >
> > > [1] Parikh et al., "Competing Bandits in Decentralized Contextual Matching Markets," 2024.

---

### Decision · Program_Chairs · 2026-04-30

**Decision:**

Accept (regular)

**Comment:**

The paper studies bandit learning in contextual matching markets under both stochastic and adversarial contexts, introducing a probabilistic minimum preference gap, fully adaptive algorithms, and matching instance-independent bounds. Two reviewers scored 5 nd two scored 4. One concern that came up during the reviews was the centralized assumption, but the authors provided a detailed decentralized extension during the discussion that reviewer VcyL found convincing enough to increase their score. The theoretical contributions are solid, and the authors aso addressed scalability concerns raised by the reviewers with additional experiments.